# Finding Dori 🐠: Memorization in Text-to-Image Diffusion Models Is Not Local

## Abstract

Text-to-image diffusion models (DMs) have achieved remarkable success in image generation. However, concerns about data privacy and intellectual property remain due to their potential to inadvertently memorize and replicate training data. Recent mitigation efforts have focused on identifying and pruning weights responsible for triggering verbatim training data replication, based on the assumption that memorization can be localized. We challenge this assumption and demonstrate that, even after such pruning, small perturbations to the text embeddings of previously mitigated prompts can re-trigger data replication, revealing the fragility of such methods. Our further analysis then provides multiple indications that memorization is indeed *not* inherently local: (1) replication triggers for memorized images are distributed throughout text embedding space; (2) embeddings yielding the same replicated image produce divergent model activations; and (3) different pruning methods identify inconsistent sets of memorization-related weights for the same image. Finally, we show that bypassing the locality assumption enables more robust mitigation through adversarial fine-tuning. These findings provide new insights into the nature of memorization in text-to-image DMs and inform the development of more reliable mitigations against DM memorization.

## 1 Introduction

Generating high-quality images with diffusion models (DMs) enjoys great popularity. However, undesired memorization and verbatim replication of training data in text-to-image DMs (Somepalli et al., 2023; Carlini et al., 2023) poses significant risks to privacy and intellectual property, as it can favor the unintended replication of sensitive or copyrighted data points during inference. In response, various detection and mitigation strategies have been proposed (Somepalli et al., 2023; Webster, 2023; Wen et al., 2024; Ren et al., 2024). Most existing mitigation techniques either aim to identify and filter out highly memorized samples during training (Somepalli et al., 2023; Ren et al., 2024) or modify inputs at inference time (Somepalli et al., 2023; Wen et al., 2024; Ren et al., 2024) to reduce memorization-induced data replication. While the training-based methods require computationally expensive retraining, the inference-time methods are limited to models behind APIs, as users of open-source models can easily disable these mechanisms by altering the source code.

To overcome both limitations, recent approaches (Hintersdorf et al., 2024; Chavhan et al., 2024) observe that the text prompts of memorized images elicit distinct activation patterns in the DMs. Based on these activations, the methods prune a small set of weights, effectively reducing the risk of verbatim data replication, while preserving overall image quality. However, since these methods work with a single prompt per memorized image, it remains an open question whether they prevent the replication of memorized images through different inputs. We search for *Diffusion Memorization* (**Dori** 🐠) beyond the prompt space by crafting *adversarial embeddings*, *i.e.,* text embeddings different from the memorized prompts, that trigger generation of memorized images. Such adversarial embeddings allow us to recover supposedly removed memorized data after pruning (see Fig. 1, left), revealing that pruning merely conceals verbatim memorization. Rather than being limited to a subset of individual weights, memorization appears to be distributed throughout the model. For a single memorized data point, multiple adversarial embeddings can trigger its replication, with the DM following different generation paths, see Fig. 1 (right). Similarly, the different activation patterns and memorization weights identified for the same memorized image vary across different inputs that trigger its replication, further undermining the notion of locality.

Figure 1: **Left:** ❶ *Without mitigation*, the DM closely replicates the training sample. ❷ *Mitigation strategies*, such as pruning memorization neurons with NeMo (Hinterdorf et al., 2024) or Wanda (Chavhan et al., 2024), prevent replication for the memorized prompt, thereby suggesting successful removal. Yet, ❸ *adversarial embeddings* 🐟 still trigger replication. **Right:** While pruning alters the generation trajectory for the original memorized prompt (blue), adversarial embeddings steer denoising along alternative paths (red) that still lead to the memorized content, unaffected by the pruning-based mitigation.

Abandoning the locality assumption, we develop the first memorization removal method effective against adversarial embeddings. We employ adversarial training (Szegedy et al., 2014; Goodfellow et al., 2014; Madry et al., 2018), which iteratively searches for adversarial embeddings that trigger replication, and pair it with full fine-tuning rather than pruning a subset of weights to achieve reliable removal of memorized data points.

In summary, we make the following contributions:

1. We reveal that existing weight-pruning methods merely conceal memorization in text-to-image DMs rather than truly erase memorized individual data points from a model.

2. We challenge the assumption that memorization is local, demonstrating that locality fails to hold across (1) the text embedding space, (2) a model's activations, and (3) its trained weights.

3. Finally, we introduce fine-tuning with adversarial text embeddings as a strong memorization removal method, demonstrating that memorization can be permanently mitigated in already trained DMs without relying on locality, paving the way for more refined methods.

## 2    BACKGROUND AND RELATED WORK

In this section, we present the core principles of text-to-image generation using DMs and introduce research focused on unintended memorization of individual training data points. We also discuss the fundamental differences between mitigating memorization and concept unlearning.

### 2.1    TEXT-TO-IMAGE GENERATION WITH DIFFUSION MODELS

Diffusion models (Song & Ermon, 2020; Ho et al., 2020) (DMs) are a class of generative models trained by gradually corrupting training images by adding Gaussian noise and training a model $\epsilon_{\boldsymbol{\theta}}$ to predict the noise that has been added. Once trained, DMs generate new images by starting from pure noise $\boldsymbol{x}_T \sim \mathcal{N}(\boldsymbol{0}, \boldsymbol{I})$ and progressively denoising it. At each time step $t = T, \ldots, 1$, the model $\epsilon_{\boldsymbol{\theta}}$ predicts the noise $\epsilon_{\boldsymbol{\theta}}(\boldsymbol{x}_t, t, \boldsymbol{y})$ needed for the denoising step. In the domain of text-to-image generation, the denoising process is guided by a text prompt $\boldsymbol{p}$, which is transformed into a text embedding $\boldsymbol{y}$ by a text encoder. We discuss more technical details on their training in Appx. E.1.

### 2.2    MEMORIZATION IN DIFFUSION MODELS

**Definition.** In the context of generative models, memorization (Feldman & Zhang, 2020; Feldman, 2020) can manifest as the model reproducing portions of its training data, such as closely replicating a particular individual training sample. Specifically, verbatim memorization (VM) describes cases when a training image is reliably generated by the model with almost a pixel-perfect match. Template memorization (TM) is a more relaxed notion, in which only parts of the image are closely replicated, such as the background of an image or a specific object (Webster, 2023). Especially, verbatim

generation of individual training data points has a detrimental effect on the trustworthy deployment of DMs, as it can lead to privacy leaks and copyright violations if sensitive and copyrighted data is included in the models' training set.

**Memorization in DMs.** Recent work has demonstrated that DMs, especially text-to-image models (Rombach et al., 2022; Saharia et al., 2022), are prone to unintended data point memorization (Somepalli et al., 2023; Carlini et al., 2023; Kadkhodaie et al., 2024; Gu et al., 2023; Chen et al., 2024b; Ma et al., 2024; Dar et al., 2023; Zhang et al., 2024a), raising concerns around privacy and intellectual property. Since then, multiple methods have been developed to detect data replication (Webster, 2023; Wen et al., 2024; Ren et al., 2024; Kriplani et al.). While many of these techniques rely on the availability of training prompts to identify memorized content, another line of research detects memorization even in the absence of training prompts, focusing instead on identifying specific memorized images (Ma et al., 2024; Jiang et al., 2025).

**Mitigation.** Memorization in DMs can either be prevented during training or by intervening in the generation process at inference time. Existing training-time mitigation techniques either adjust the training data by removing duplicates (Carlini et al., 2023; Somepalli et al., 2023) or reject training samples for which the model indicates signs of memorization (Wen et al., 2024; Ren et al., 2024; Chen et al., 2025). However, since re-training large DMs is expensive, inference-time mitigation strategies are crucial for already trained models. These mitigation strategies adjust the input tokens (Somepalli et al., 2023), update the text embeddings (Wen et al., 2024), change the cross-attention scores (Ren et al., 2024), or guide the noise prediction away from memorized content (Chen et al., 2024a). However, all these methods offer no permanent mitigation, increase the inference time, and can easily be turned off for locally deployed models.

**Local Pruning-Based Mitigation.** More permanent solutions have focused on identifying and removing the weights responsible for triggering data replication. Hintersdorf et al. (2024) developed *NeMo*, a localization algorithm to detect memorization neurons within the cross-attention value layers of DMs, of which all weights are pruned. More specifically, NeMo first conducts an out-of-distribution detection to identify neurons with high absolute activations under memorized prompts and reduces the set of identified neurons by checking their influence on data replication individually. Similarly, Chavhan et al. (2024) applied *Wanda* (Sun et al., 2024), a pruning technique originally developed for large language models, to locate and prune individual weights in the output fully-connected layers of transformer blocks responsible for memorization. Wanda identifies weights by their weight importance, computed as the product between the weights and the activation norm. The method then prunes the top $k\%$ of weights with the highest importance scores compared to scores computed on a null string. While both methods successfully avoid data replication triggered by memorized prompts, it remains open whether the memorized data points are successfully erased from the model.

**Memorization Mitigation vs. Concept Unlearning in DMs.** Apart from the localization-based memorization mitigation techniques, one of the few approaches that try to remove information from DMs' weights are *concept unlearning methods* (Gandikota et al., 2023; Kumari et al., 2023; Zhang et al., 2024b) that are used for content moderation. Although these methods bear some methodological similarity, they pursue fundamentally different objectives. Concept unlearning targets the suppression of broad, high-level concepts, such as nudity or specific objects (*e.g.,* cars), across all generations. In contrast, memorization mitigation (this work) seeks to remove the model's ability to reproduce *specific, individual* memorized training data points. For example, mitigating verbatim generation of a particular memorized image of a car to protect copyright prevents the model from generating *that exact image*, but does not affect its capacity to generate cars in general. Concept unlearning, on the other hand, eliminates the model's ability to generate *any* car, which is undesirable when we want to mitigate memorization of a specific data point but leave the model unchanged otherwise. Therefore, mitigating memorization, *i.e.,* (verbatim) training data replication, although deceptively similar to concept removal, needs a different approach. In the next sections, we validate this empirically by showing that concept removal methods are indeed not suitable for mitigating the memorization of individual data points, which calls for stronger, tailored tools.

## 3 BREAKING PRUNING-BASED MITIGATION METHODS

In this section, we highlight that pruning-based memorization mitigations only conceal memorization but fail to truly remove memorized images from DMs. Specifically, we show that even after applying

these mitigations, we can still trigger the generation of memorized images through carefully crafted adversarial text embedding. This reveals that, despite the apparent mitigation of memorization under standard textual prompts, the underlying memorized images persist in the model weights.

## 3.1 FINDING DORI 🐟 WITH ADVERSARIAL TEXT EMBEDDINGS

To demonstrate that pruning-based memorization mitigation strategies (NeMo and Wanda) fail to *truly remove* memorized images from DMs, we develop a novel approach for generating adversarial text embeddings that can still trigger the verbatim reproduction of supposedly removed memorized data points. Instead of relying on natural-language prompts, we use unconstrained continuous optimization in the text embedding space to uncover triggers capable of reconstructing memorized images, even after a mitigation has been applied. The existence of such adversarial triggers reveals that memorized content is still encoded in the model weights and can be verbatim extracted, which poses a significant privacy and copyright risk, especially for open-weight models.

Formally, let $\boldsymbol{x}_{mem}$ be a known memorized image and $\boldsymbol{y}_{mem}$ the text embedding for its prompt. After weight pruning using NeMo or Wanda, the model no longer replicates $\boldsymbol{x}_{mem}$ when conditioned on $\boldsymbol{y}_{mem}$, giving the impression that memorization has been successfully removed. To verify removal, we optimize an adversarial embedding $\boldsymbol{y}_{adv}$ initialized with $\boldsymbol{y}_{mem}$, using gradients of the standard diffusion loss $\mathcal{L}_{DM}$ (see Eq. (3)), with learning rate $\eta$ over multiple steps $i$ as:

$$\boldsymbol{y}_{adv}^{(i+1)} = \boldsymbol{y}_{adv}^{(i)} - \eta \nabla_{\boldsymbol{y}_{adv}^{(i)}} \mathcal{L}_{DM}(\boldsymbol{x}_{mem}, \boldsymbol{\epsilon}, \boldsymbol{y}_{adv}^{(i)}, t, \boldsymbol{\theta}_{NeMo/Wanda}), \tag{1}$$

where $\boldsymbol{\theta}_{NeMo/Wanda}$ are parameters of the DM after applying NeMo or Wanda to mitigate replication of $\boldsymbol{x}_{mem}$. Our goal is to find a final adversarial embedding $\boldsymbol{y}_{adv}$ that consistently triggers $\boldsymbol{x}_{mem}$, regardless of the initial noise, so we re-sample the noise $\boldsymbol{\epsilon} \sim \mathcal{N}(\boldsymbol{0}, \boldsymbol{I})$ at each optimization step. Similarly, we re-sample the timesteps $t \sim \mathcal{U}(1, T)$ to ensure that $\boldsymbol{y}_{adv}$ reliably triggers $\boldsymbol{x}_{mem}$ during the iterative denoising generation process of the DM. A detailed formulation of the optimization procedure is provided in Alg. 1 in Appx. F.

For comparison, we also evaluate UnlearnDiffAtk (Zhang et al., 2024c), a state-of-the-art method designed to re-trigger forgotten concepts in the context of concept unlearning. As established above, concept unlearning fundamentally differs from the challenge of removing individual memorized training examples, yet UnlearnDiffAtk represents the closest available baseline and is therefore included for completeness. Our results (see Appx. G.1) show that UnlearnDiffAtk fails to re-generate memorized images following pruning-based mitigations. This highlights both the particular challenge of showing the limits of pruning-based memorization techniques and the necessity of our adversarial optimization strategy, which enables analysis beyond what was possible with prior methods.

## 3.2 EXPERIMENTAL SETUP

We begin by defining the experimental setup used in this and the subsequent sections.

**Models and Datasets:** We focus our investigation on Stable Diffusion v1.4 (Rombach et al., 2022) and a set of 500 memorized prompts (Wen et al., 2024; Webster, 2023) from the LAION-5B (Schuhmann et al., 2022) training dataset, in line with previous research on memorization in text-to-image DMs (Wen et al., 2024; Ren et al., 2024; Hintersdorf et al., 2024; Chavhan et al., 2024). More recent DMs are trained on more carefully curated and deduplicated datasets, which reduces the amount of memorization, as discussed in previous work (Somepalli et al., 2023; Webster et al., 2023). Therefore, *Stable Diffusion v1.4 is the only existing model for which a known set of memorized prompts is publicly available*. As a result, it is currently *not possible* to conduct comparable memorization studies on other models. In the main paper, we focus on VM prompts as they represent the most critical form of memorization, but we additionally include results for TM prompts in Appx. H.

**Metrics:** Following prior work (Wen et al., 2024; Hintersdorf et al., 2024), we employ SSCD (Pizzi et al., 2022), a feature extractor commonly employed to detect and quantify copying behavior in DMs. To measure similarity between two images, we compute the cosine similarity between their SSCD feature embeddings. Higher values indicate a higher degree of content replication. All metrics are computed as the median of the maximum scores across ten generated images per memorized prompt or adversarial embedding. We vary the seeds for image generation and adversarial embedding optimization to avoid overfitting.

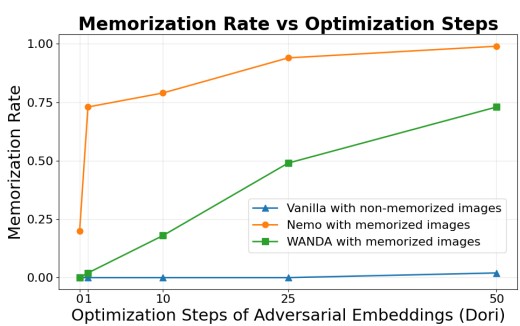

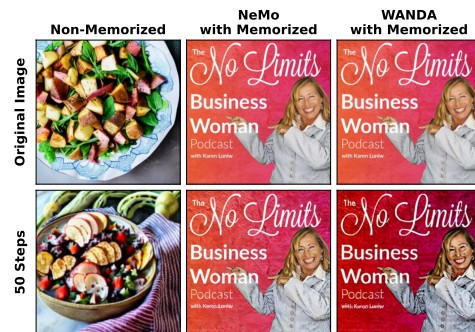

(a) Memorization Rate comparison between memorized and non-memorized images.

(b) Qualitative comparison between memorized and non-memorized images.

Figure 2: **Dori's 🐟 adversarial embeddings do not trigger replication of non-memorized images.** Memorization Rate (left) quickly spikes for memorized images, while remains at 0 for non-memorized images. Similarly, after 50 optimization steps, memorized images are replicated with almost pixel-perfect precision, while non-memorized images become only semantically similar (right).

To evaluate **replication**, we define $SSCD_{Orig}$ as the cosine similarity between generated images and their associated training image. Values above $0.7$ indicate that the memorized image is successfully generated (Wen et al., 2024). Additionally, we use the similarity between images generated before and after mitigation techniques are applied, denoted by $SSCD_{Gen}$, to assess the effects of mitigation. We expect lower $SSCD_{Gen}$ scores when mitigation is successful.

While SSCD-based metrics capture the overall trend of replication, we also quantify the number of images that remain memorized even after mitigation. To this end, we define the Memorization Rate (MR) as the ratio of memorized images that the model can still replicate, *i.e.,* those achieving an SSCD score above $0.7$ for at least one of the ten generations, sampled from different random seeds.

To assess **image quality**, we measure prompt alignment using CLIP (Radford et al., 2021) similarity $A_{CLIP}$, comparing each generated image with its corresponding textual prompt. Higher $A_{CLIP}$ scores indicate a stronger semantic alignment with the input prompt. We also compute the Fréchet Inception Distance (FID) (Heusel et al., 2017) and Kernel Inception Distance (KID (Binkowski et al., 2018), reported in the Appendix). Both these image quality metrics are evaluated on 30k prompts of the COCO dataset (Lin et al., 2014), a standard benchmark for image generation (Pavlov et al., 2023). Lower scores indicate improved image quality.

**Adversarial Embedding Optimization:** We initialize the adversarial embeddings with the original text embeddings of the prompts for the memorized images. We then optimize each embedding for $50$ steps with a learning rate of $0.1$ using Adam (Kingma & Ba, 2015) and a batch size of $8$. We perform $50$ update steps for Dori to ensure that only truly memorized content will be replicated through adversarial embeddings. In Fig. 2a, we confirm that indeed with that setting the model does not significantly generate non-memorized images, with an MR of only $0.02$. Conversely, memorized images are easily generated even after *one* optimization step for NeMo (MR $0.73$). Fig. 2b shows a significant qualitative difference in the behavior of Dori between memorized and non-memorized images. For memorized content, after ten embedding optimization steps, the content already closely resembles the original image. For non-memorized content, however, even after 50 steps, there are clear conceptual differences between the original and the generated images. We provide additional qualitative comparisons in Fig. 8. We extend our analysis and further motivate our selection of the threshold of $50$ steps in Appx. F, where we show that to reliably replicate an arbitrary (non-memorized) image, it is necessary to perform *over 500 steps* with Dori, *i.e.,* 10 times more than in our setting.

### 3.3 Pruning-Based Mitigation Conceals but Does Not Erase Memorization

Our analysis in Table 1 highlights that NeMo and Wanda prevent training data replication only in the text space, *i.e.,* breaking the mapping from the prompt to the corresponding memorized image, but do not remove the memorized images from the DM. As shown in the first row of Table 1, verbatim memorized prompts trigger the replication of the memorized training images in the original DM (No Mitigation). In contrast, image generations for non-memorized training prompts (2nd row) show no

Table 1: **Pruning-based mitigation of memorization is vulnerable to adversarial embeddings**. Without any mitigation technique applied (1st row), the generated images clearly indicate data replication. Searching for adversarial embeddings on non-memorized prompts (2nd row) does not lead to clear replication. After localizing and pruning weights with NeMo (3rd row) or Wanda (4th row), data replication appears effectively prevented. However, identifying adversarial embeddings with Dori (indicated by 🐟) reveals that embeddings capable of triggering data replication may persist, even after pruning.

| Setting | Memorization Type | ↓ SSCD$_{\text{Orig}}$ | ↓ SSCD$_{\text{Gen}}$ | ↑ $A_{\text{CLIP}}$ | ↓ MR | ↓ FID |
|---|---|---|---|---|---|---|
| No Mitigation | Verbatim | **0.90** $\pm$ 0.04 | N/A | 0.33 $\pm$ 0.01 | 0.98 | 14.44 |
| Non-Memorized Prompts | None | 0.17 $\pm$ 0.05 | N/A | 0.35 $\pm$ 0.02 | 0.00 | 14.44 |
| **Non-Memorized Prompts + 🐟** | None | **0.48** $\pm$ 0.06 | N/A | 0.32 $\pm$ 0.02 | **0.00** | |
| NeMo (Hintersdorf et al., 2024) | Verbatim | 0.33 $\pm$ 0.18 | 0.40 $\pm$ 0.21 | 0.34 $\pm$ 0.02 | 0.20 | 15.16 |
| **NeMo + 🐟** | Verbatim | **0.91** $\pm$ 0.03 | 0.97 $\pm$ 0.02 | 0.33 $\pm$ 0.02 | **0.99** | |
| Wanda (Chavhan et al., 2024) | Verbatim | 0.20 $\pm$ 0.08 | 0.21 $\pm$ 0.09 | 0.34 $\pm$ 0.02 | 0.00 | 16.86 |
| **Wanda + 🐟** | Verbatim | **0.76** $\pm$ 0.05 | 0.82 $\pm$ 0.05 | 0.32 $\pm$ 0.01 | **0.72** | |

signs of memorization under the SSCD score. Even when applying Dori, our adversarial embedding optimization, indicated by 🐟 in the table, the resulting metrics suggest no close data replication for non-memorized prompts. This finding is particularly important for the validity of our investigations, as it confirms that our adversarial embedding optimization method specifically targets memorized content and does not falsely report memorization for non-memorized data. We explore adversarial embeddings in the context of non-memorized data points more closely in Appx. F.1.

Applying NeMo (third row) or Wanda (fourth row), respectively, substantially reduces training data replication as reflected by lowered SSCD scores in contrast to the original DM. At first glance, both methods appear effective at mitigating the replication of memorized data points, as also visualized in Fig. 1 (❷). However, blocking replication from the original prompts does not imply that the memorized individual images have been removed from the model, as shown in rows marked with 🐟 and Fig. 1 (❸). These results suggest that pruning-based methods like NeMo and Wanda primarily conceal memorization rather than eliminate it. Also for TM results, reported in Appx. H.2, we observe increased replication of memorized training data, yet SSCD-based metrics fail to correctly quantify this type of replication due to their non-semantic variations in generated images. Overall, these results suggest that pruning-based memorization mitigations prevent the replication of memorized images via the text space but leave the memorized images intact internally in the DM.

**Ablations.** We also conduct a sensitivity analysis (Appx. H.2) on the number of *steps* required to yield adversarial embeddings, finding that in most cases, significantly fewer than 50 steps are already sufficient to identify embeddings that circumvent the mitigation methods. We also experiment with increasing the *strength of NeMo and Wanda* to test if they become resilient to Dori. For the former, we iteratively increase the set of pruned weights based on new adversarial embeddings (see Appx. H.6), and for the latter, we simply prune 10% of weights, instead of 1% (see Appx. H.5). In these settings, Wanda successfully removes memorization but at substantial damage to the DM's generative capabilities (see Appx. H.7), while NeMo remains non-robust to Dori.

## 3.4 GENERALIZATION TO OTHER MODELS

To verify the generalizability of our claims, we evaluate Stable Diffusion v2.0 with a subset of the prompts from Webster (2023), consistent with the setup in Ren et al. (2024). First, we perform filtering of the original images to identify clearly memorized image-text pairs (see Appx. H.8 for details). Then, we apply NeMo and Wanda to mitigate memorization, similarly as for SD-v1.4, and then evaluate the robustness of these pruned models against Dori using the same parameters as in Sec. 3.2. We observe similar behavior: pruning-based mitigation fails to fully remove memorized images from the model, as highlighted in Tab. 2. We extend our evaluation to other metrics in Appx. H.8.

Table 2: **Pruning-based methods fail for Stable Diffusion v2.0.**

| Setting | ↓ SSCD$_{\text{Orig}}$ | ↓MR |
|---|---|---|
| No Mitigation | 0.77 $\pm$ 0.04 | 1.00 |
| NeMo | 0.23 $\pm$ 0.07 | 0.06 |
| **NeMo + 🐟** | **0.87** $\pm$ 0.02 | **1.00** |
| Wanda | 0.09 $\pm$ 0.03 | 0.00 |
| **Wanda + 🐟** | **0.70** $\pm$ 0.04 | **0.53** |

## 4 THE ILLUSION OF MEMORIZATION LOCALITY

Our analysis in the previous section revealed that pruning-based memorization mitigation methods fail to remove memorized images from DMs, at least without compromising overall generation quality. This suggests that their underlying assumption of localized memorization is flawed. In this section, we provide multiple lines of evidence that *memorization in DMs is indeed not inherently local*. Specifically, in Section 4.1, we show that replication triggers for memorized images are widely distributed in the text embedding space, in Section 4.2, we demonstrate that distinct embeddings producing the same memorized training image can induce divergent model activations, and in Section 4.3, we find that different pruning methods identify inconsistent sets of memorization-related weights for the same image. Finally, in Section 4.4 we demonstrate that abandoning the locality assumption enables more robust mitigation through adversarial fine-tuning, paving the way towards novel methods that truly mitigate memorization of individual data points in DMs.

### 4.1 DATA REPLICATION TRIGGERS ARE NOT LOCALIZED IN TEXT EMBEDDING SPACE

In the previous section, we demonstrated that pruning-based memorization mitigations (NeMo and Wanda) can be broken by optimizing adversarial embeddings initialized *near the original training prompt*. Here, we show that triggers in the embedding space do not need to lie close to a memorized image's prompt: even adversarial embeddings initialized at *random, distant positions* in the text embedding space can still reliably trigger replication of memorized images.

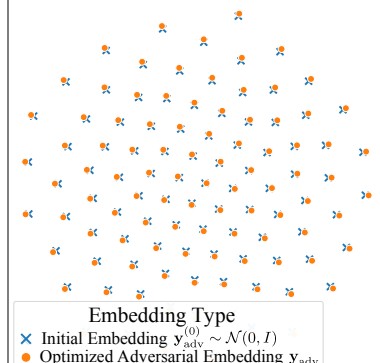

To illustrate this, we construct a set of 100 adversarial embeddings, $\boldsymbol{Y}_{adv}$, for a single memorized image, where each embedding is randomly initialized: $\boldsymbol{y}_{adv}^{(0)} \sim \mathcal{N}(\boldsymbol{0}, \boldsymbol{I})$. After optimizing each embedding for 50 steps in accordance with the procedure described in Section 3.1, we show that every run produces a generated image highly similar to the memorized sample (all with $\mathrm{SSCD}_{\mathrm{Orig}}$ scores clearly exceeding 0.7). Despite their consistent success at replication, the optimized adversarial embeddings remain widely dispersed in the embedding space and retain a distribution similar to their random initializations, as visualized by the t-SNE (van der Maaten & Hinton, 2008) plot in Fig. 3. This result further refutes the assumption of input space locality.

We repeat the experiment by initializing $\boldsymbol{y}_{adv}^{(0)}$ with embeddings of 100 randomly selected, *non-memorized* prompts. Results from this experiment, presented in Fig. 10 (see Appx. J.1), draw a similar picture of evenly distributed replication triggers. Both results clearly demonstrate that replication of memorized images can be triggered virtually from all over the embedding space, taking away the illusion of local memorization triggers.

Figure 3: **Data replication triggers are widely and uniformly scattered in the text embedding space.**

To quantify our observations further, we compute the pairwise distances both among all the initial random embeddings and among all the optimized adversarial embeddings. As shown in Fig. 11 (Appx. J.1), the optimized embeddings are in fact *even more* dispersed than their initializations, confirming that successful replication triggers are widely scattered rather than localized. These results demonstrate that the assumption of *input locality*, implicit in the pruning-based mitigation methods, is unfounded. Effective memorization mitigation must therefore address potential triggers distributed throughout the embedding space, not just those near the training prompt.

### 4.2 DIFFERENT TRIGGERS, DIFFERENT ACTIVATIONS, THE SAME IMAGE

Next, we examine internal model activations to assess whether replication triggers for memorized images are also dispersed at the level of network activity. For fixed input noise, we expect that activations should vary depending on the guiding text embedding. This analysis is essential, as pruning methods like Wanda and NeMo select candidate weights for removal based on activation patterns, treating these as per-weight importance metrics. If different adversarial embeddings that trigger the same image yield distinct activations, these methods may prune inconsistent sets of weights, undermining the assumption of locality in memorization further.

To quantify activation variability, we introduce a *discrepancy* metric, defined as the mean pairwise $\ell_2$-distance between activations in a given layer across different input embeddings during the initial denoising step. To ensure a fair comparison, the input noise is fixed while only the guiding adversarial embedding is varied. The exact formulation is provided in Eq. (6) (see Appx. J.2). We compute discrepancies for two embedding sets: $\mathbf{Y}_{adv}$, comprising 100 adversarial embeddings for a single memorized image (from Section 4.1), and $\mathbf{Y}_{mem}$, containing 100 text embeddings of randomly selected prompts associated with different memorized images. Since replicating different images should produce more varied activations, we expect higher discrepancies for $\mathbf{Y}_{mem}$ and lower, more consistent values for $\mathbf{Y}_{adv}$.

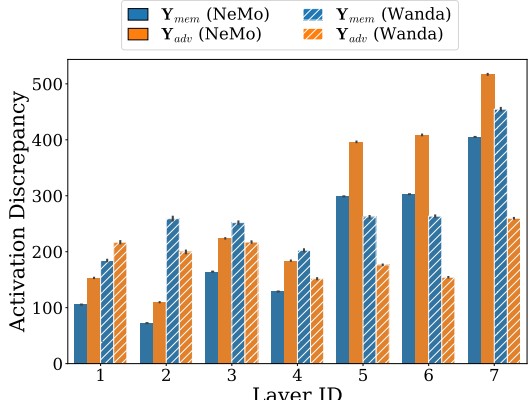

Figure 4: **Diverse activations refute locality.** Although adversarial embeddings trigger the same image, their activations exhibit high discrepancy.

We analyze activations from the layers where NeMo and Wanda operate, specifically, the value layers of cross-attention modules for NeMo and the second linear layers of the transformer blocks' two-layer feed-forward networks for Wanda. In total, we compute activations for seven cross-attention modules, indexed from 1 to 7, spanning the three Down blocks (indices 1 to 6, each block has two modules) and the Mid block of the DM's U-Net (Ronneberger et al., 2015), following the setup of NeMo.

Surprisingly, as shown in Fig. 4, the discrepancy among activations for adversarial embeddings in $\mathbf{Y}_{adv}$ is comparable to that for randomly selected memorized prompts in $\mathbf{Y}_{mem}$. This finding indicates that different adversarial embeddings, even when generating the same image, cause distinct activation pattern, contradicting the expectation that a common output implies similar activations. While Wanda shows slightly reduced discrepancy for $\mathbf{Y}_{adv}$, the variability remains substantial, suggesting each adversarial embedding invokes a unique activation pattern. Extending this analysis to all U-Net layers (see Fig. 12 in Appx. J.3) yields similar results, providing further evidence against the locality assumption in memorization.

### 4.3 IMAGES ARE NOT MEMORIZED IN A SUBSET OF WEIGHTS

While high discrepancy scores suggest that different subsets of weights contribute to data replication, we further assess the consistency of pruning-based mitigation methods across adversarial embeddings for the same image. Since NeMo and Wanda select weights for pruning based on activation patterns, we expect the identified sets of weights to vary with different adversarial triggers. To quantify this, we define a *weight agreement* metric as the intersection over union of weights identified for pruning by NeMo or Wanda between two adversarial embeddings, averaged across all pairs. Higher values indicate greater overlap of the identified weights. The metric's formal definition is given in Eq. (7) (see Appx. J.2). Weight agreement is set to 1, representing a perfect overlap, if no weights are selected for either embedding in a given layer.

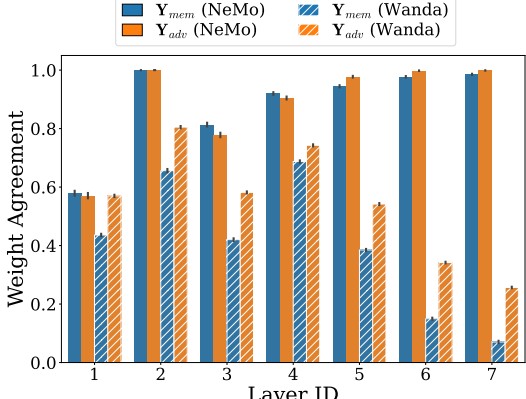

Figure 5: **Locality fails in the model's weights.** Large activation discrepancy (Fig. 4) results in low weight agreement, further undermining the idea that weights responsible for replicating a memorized image can be pinpointed and pruned.

As shown in Fig. 5, the overlap in identified weights for adversarial prompts in $\mathbf{Y}_{adv}$ is limited: Wanda 's agreement remains below 0.6 for most layers, while NeMo exceeds 0.8 except in the first layer, where the overlap drops to about 0.6, similar to results for the set of diverse memorized prompts $\mathbf{Y}_{mem}$. Notably, this high agreement in deeper layers is mainly due to NeMo attributing

most memorization-related weights to the first layer, as further visualized in Fig. 13 (left) in Appx. J.2. Despite appearing more stable, iterative pruning experiments (see Appx. H.6) reveal that NeMo still identifies different weights for different adversarial embeddings after previous weights are pruned.

Crucially, the weight agreement among adversarial embeddings in $Y_{adv}$ is comparable to that among distinct memorized prompts in $Y_{mem}$, reinforcing the finding that both pruning-based approaches struggle to consistently localize the weights responsible for memorization. This further challenges the locality assumption and underscores the limitations of such memorization mitigation strategies.

### 4.4 ABANDONING THE LOCALITY ASSUMPTION YIELDS ROBUST MITIGATION METHOD

The consistent evidence against memorization locality from the previous sections suggests that memorization removal in DMs cannot be achieved by targeting just a subset of weights. Instead, our findings point to the need for mitigation strategies that operate at the level of the entire model. To verify this, we evaluate a simple but powerful method: adversarial fine-tuning, where all model parameters are adjusted to remove memorized content.

**Approach.** We employ Dori (Section 3.1) to generate multiple adversarial embeddings for each memorized image and fine-tune the DM in an adversarial manner, inspired by adversarial training (Szegedy et al., 2014; Goodfellow et al., 2014; Madry et al., 2018). During training, adversarial embeddings should produce images that are semantically close to the memorized samples, but not exact replicas. To facilitate this, we first craft a set of *surrogate images* $\widetilde{x}$, obtained by prompting the DM with memorized prompts under a pruning-based mitigation. These surrogate images resemble the memorized samples in content and style, yet differ in their details, ensuring they are no replications (see Appx. I.2 for visual examples). In addition, at each fine-tuning step, we iteratively collect a set of adversarial embeddings $y_{adv}$ that trigger replication of the memorized images. We then fine-tune the DM on a mix of these surrogate images and adversarial embeddings with a loss defined as:

$$\mathcal{L}_{Adv}(\widetilde{x}_0, \epsilon, y_{adv}, t, \theta) = \| \epsilon - \epsilon_\theta \left( \widetilde{x}_t, t, y_{adv} \right) \|_2^2. \tag{2}$$

The loss encourages the model to avoid replicating the memorized training image triggered by the adversarial embedding. Instead, it guides generation toward the surrogate images, which preserve the semantic content without being exact copies. Using a diverse set of surrogate images ensures that no *new* memorized image is inadvertently introduced into the DM. In addition, we use the standard diffusion loss $\mathcal{L}_{DM}$ defined in Eq. (3), to train on non-memorized image-captions pairs from the LAION (Schuhmann et al., 2022) dataset, to preserve the model's general utility. The final optimization loss is $\mathcal{L} = \mathcal{L}_{DM} + \mathcal{L}_{Adv}$. An algorithmic description of our adversarial fine-tuning method, as well as the full setup, is provided in Appx. I.

**Results.** We find that our adversarial fine-tuning procedure quickly removes memorized content. Table 3 (bottom row) presents the evaluation results after fine-tuning for five epochs. The results show that adversarial embeddings can no longer trigger data replication (except for a single, highly duplicated training sample), demonstrating a permanent mitigation relative to pruning-based methods. At the same time,

Table 3: **Dori 🐟 against fine-tuning removal.**

| Method | ↓ SSCD$_{\text{Orig}}$ | ↓ MR |
|---|---|---|
| ESD + 🐟 | $0.90 \pm 0.04$ | 0.98 |
| Concept Ablation + 🐟 | $0.91 \pm 0.04$ | 0.97 |
| SISS + 🐟 | $0.60 \pm 0.22$ | 0.39 |
| **Our Mitigation + 🐟** | $\mathbf{0.36 \pm 0.14}$ | **0.02** |

the model's utility is preserved: the FID score improves from 14.44 to 13.61 after fine-tuning, suggesting no harm to the image quality, as extended results in Tab. 5 (see Appx. G) show. We provide comprehensive analyses of the parameters and performance of the method in Appx. I. These results indicate that even a single fine-tuning epoch substantially reduces memorization and can prevent data replication in most cases. We also experimented with fine-tuning the DM with LoRA adapters (Hu et al., 2021), but found it unsuccessful in mitigating memorization, further underscoring that effective memorization mitigation requires global model adjustments.

**Baseline.** We compare our method with the state-of-the-art fine-tuning approach for data point unlearning in DMs, namely SISS (Alberti et al.), which aims to remove a specific (not necessarily memorized) image from the pre-trained DM. We remove a single image from the U-Net by performing 35 update steps. For each image, we start from the original DM, following the default setting for SISS. We describe SISS and its setup in Appx. E.2. While our results in Tab. 3 show that it successfully drops SSCD$_{\text{Orig}}$ below the memorization threshold of 0.7, we note that SISS fails to remove *39%* of

memorized samples, as indicated by the memorization rate (MR). This highlights that the method is still limited for the reliable mitigation of memorized images when faced with adversarial embeddings.

**Results for Stable Diffusion v2.0.** In Sec. 3.4 we show that also for a more advanced model, *i.e.,* Stable Diffusion v2.0 the pruning efforts fail to remove memorized images. We test if our mitigation also works for this model. To this end, we apply it to the model using the same hyperparameters as for SD-v1.4, *i.e.,* 5 epochs of fine-tuning with Dori embeddings obtained

Table 4: **Our mitigation successfully removes memorized images from Stable Diffusion v2.0.**

| Setting | ↓ SSCD$_{\text{Orig}}$ | ↓MR | ↓FID |
|---|---|---|---|
| NeMo + | $0.87 \pm 0.02$ | 1.00 | **15.35** |
| Wanda + | $0.70 \pm 0.04$ | 0.53 | 16.79 |
| **Our Mitigation +** | $\mathbf{0.14 \pm 0.06}$ | **0.06** | 16.01 |

for the models' memorized images. Then, we test if the mitigation is successful using our Dori. In Tab. 4, we show that our fine-tuning strategy provides a substantially more robust mitigation, reflected in a significantly lower memorization rate (MR), while preserving overall image quality (as measured by FID). We provide extended results in Appx. H.8.

**Applying Concept Unlearning.** Finally, for completeness, we also evaluate ESD (Gandikota et al., 2023) and Concept Ablation (Kumari et al., 2023), state-of-the-art concept unlearning methods for DMs. As discussed in Sec. 2, concept unlearning successfully targets the removal of broad, high-level concepts from generative models, such as style or entire object categories, but it is not designed to remove specific, *individual memorized data points*, which is the goal of memorization mitigation. Nevertheless, as the closest available baselines from concept unlearning, we include these approaches in our evaluation. As shown in Tab. 3 (top two rows), and Tab. 5 (see Appx. G.2), both methods fail to reliably prevent the replication of memorized data when confronted with adversarial trigger embeddings. We hypothesize this is because these methods rely on a single prompt, or its augmented versions, as a replication trigger, which is insufficient for thorough memorization mitigation, as we already demonstrated in Section 4.1. We extend the discussion and experimental setup in Appx. G.

Overall, our results highlight that existing concept and data unlearning methods are ill-suited for mitigating memorization. Although concept unlearning methods can suppress broad categories, they fail to reliably eliminate memorized content. Similarly, the current state-of-the-art data unlearning method (SISS), which claims to be effective at removing memorized images, also remains vulnerable to adversarial embeddings. Moreover, while existing mitigation methods are effective at preventing data replication when triggered from the prompt space, fully removing memorization requires novel approaches. Such methods must abandon the locality assumption in input and weight space, and operate *globally*, for instance, through adversarial fine-tuning with many triggers, as in our approach.

# 5 DISCUSSION AND CONCLUSIONS

Our findings reveal that memorization in text-to-image DMs is not inherently a local phenomenon. While pruning-based methods such as NeMo and Wanda can suppress the generation of memorized training images when prompted with the original captions, they do not remove the image memorization from the model. In particular, we show that the memorized images can still be reliably regenerated using diverse adversarial embeddings. While our results strongly indicate the limitations of targeting only local weights or activation patterns, we refrain from claiming that any form of locality can be definitively ruled out. Rather, our work highlights the significant challenge in reliably identifying which specific weights drive memorization, especially since current methods based on activation patterns fail in the presence of adversarial triggers that induce divergent activations. We show that effective memorization removal requires global model interventions, as realized by our adversarial fine-tuning approach, which robustly eliminates memorized images while preserving the model's generative performance. Overall, our insights underscore the need for novel global, model-wide memorization mitigation strategies to support the responsible deployment of generative models.

## REPRODUCIBILITY STATEMENT

For reproducibility, we describe all settings, models, datasets, and hyperparameters used during our experiments. Our Appendix further provides detailed descriptions for all steps of our method. All datasets, models, and methods used in this paper are publicly available. Additionally, we submitted our source code as supplementary material and will make the code publicly available upon acceptance.

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

## A  LIMITATIONS

Our analysis of the locality of memorization within model parameters focuses on a selected subset of layers. While it is possible that signs of locality may also be present in other components—such as self-attention mechanisms or convolutional layers—we chose to concentrate on the layers where existing mitigation methods are typically applied and where initial success in suppressing replication has been observed. This targeted approach allows us to provide concrete and meaningful insights into the locality hypothesis. Notably, to our knowledge, no current methods explicitly aim to identify memorization-related weights outside the studied layers. Furthermore, supporting evidence from the NeMo paper (Appendix C.9) indicates that pruning convolutional layers does not effectively reduce memorization, suggesting that our chosen focus captures the most relevant regions for intervention.

We focus our research only on one model, Stable Diffusion v1.4, since only for this DM a set of memorized images is currently known. We acknowledge that such a narrow scope limits the generalizability of our results. However, we would like to point out that other works successfully advanced the understanding of memorization by analyzing Stable Diffusion v1.4.

We designed Dori *solely* as an assessment tool for analyzing the phenomenon of memorization in text-to-image DMs, and *not* as a real-world attack against DMs. We acknowledge that, in some scenarios, Dori *could* be misused to elicit replication of suspected memorized image by a malicious third-party. Yet, such scenarios are highly unlikely, since the access assumptions for Dori, *i.e.,* the full access to the model's weights and code, are very strong, and hard to meet in practice.

Additionally, we recognize that our adversarial fine-tuning method for removing memorized content involves a higher computational cost compared to pruning-based approaches. This is due to the need for generating adversarial inputs, creating surrogate samples, and extending the fine-tuning dataset with non-memorized data to preserve utility. We see this as a valuable trade-off, as our method offers the first reliable and permanent mitigation that truly removes memorized images from the model. Nonetheless, we believe there is substantial potential for future work to build on our findings and develop more efficient mitigation strategies that retain our method's effectiveness while reducing computational overhead.

## B  HARD- AND SOFTWARE DETAILS

We conducted all experiments on NVIDIA DGX systems running NVIDIA DGX Server Version 5.2.0 and Ubuntu 20.04.6 LTS. The machines are equipped with 2 TB of RAM and feature NVIDIA A100-SXM4-40GB GPUs. The respective CPUs are AMD EPYC 7742 64-core. Our experiments utilized CUDA 12.2, Python 3.10.13, and PyTorch 2.2.0 with Torchvision 0.17.0 Paszke et al. (2019). Notably, all experiments are conducted on single GPUs.

All models used in our experiments are publicly available on Hugging Face. We accessed them using the Hugging Face diffusers library (version 0.27.1).

To facilitate reproducibility, we provide a Dockerfile along with our code. Additionally, all hyperparameters required to reproduce the results presented in this paper are included.

## C  MODEL AND DATASET DETAILS

Our experiments primarily use Stable Diffusion v1-4 (Rombach et al., 2022), which is publicly available at `https://huggingface.co/CompVis/stable-diffusion-v1-4`. Comprehensive information about the data, training parameters, limitations, and environmental impact can be found at that URL. The model is released under the CreativeML OpenRAIL M license.

The memorized prompts analyzed in our study originate from the LAION2B-en (Schuhmann et al., 2022) dataset, which was used to train the DM. We use a set of memorized prompts provided by Wen et al. (2024)[1], who identified them using the extraction tool developed by Webster (2023). The LAION dataset is licensed under the Creative Commons CC-BY 4.0. As the images in the dataset may be subject to copyright, we do not include them in our codebase; instead, we provide URLs that allow

---

[1]Available at `https://github.com/YuxinWenRick/diffusion_memorization`.

users to retrieve the images directly from their original sources. For performing our fine-tuning-based mitigation method, we furthermore downloaded 100k images from the LAION aesthetics dataset, a subset of LAION5B.

## D    DECLARATION ON LARGE LANGUAGE MODELS' (LLMS') USAGE

LLMs were only used to assist in writing (grammar check, phrasing), and to implement boilerplate, repetitive code for data processing and visualizations. No novelty of our work came from an LLM assistant. All new ideas and methods described in the paper are developed by the authors without the assistance of any AI system.

# E    EXTENDED BACKGROUND

## E.1    TEXT-TO-IMAGE DIFFUSION MODELS

We present the technical details of DM training: During training, a time step $t \sim \mathcal{U}(1, T)$ and a noise vector $\epsilon \sim \mathcal{N}(\mathbf{0}, \boldsymbol{I})$ are randomly sampled to create a noisy image $\boldsymbol{x}_t = \sqrt{\bar{\alpha}_t}\boldsymbol{x}_0 + \sqrt{1 - \bar{\alpha}_t}\epsilon$ based on the training image $\boldsymbol{x}_0$. The amount of noise added is controlled by a noise scheduler $\bar{\alpha}_t$, for which there are multiple choices (Song & Ermon, 2020; Nichol & Dhariwal, 2021; Kingma et al., 2021; Karras et al., 2022). The training objective for the noise predictor $\epsilon_{\boldsymbol{\theta}}$ is then to predict the noise $\epsilon$ that has been added:

$$\mathcal{L}_{DM}(\boldsymbol{x}_0, \epsilon, \boldsymbol{y}, t, \boldsymbol{\theta}) = \|\epsilon - \epsilon_{\boldsymbol{\theta}}(\boldsymbol{x}_t, t, \boldsymbol{y})\|_2^2. \tag{3}$$

Training and generating samples with DMs can be computationally expensive. The latent DM framework (Rombach et al., 2022) reduces this burden by operating in a lower-dimensional latent space instead of the pixel space. This latent space is learned by a separately trained variational autoencoder (Kingma et al., 2013; Van Den Oord et al., 2017) that encodes images into compact representations and decodes generated latents back to the image space.

## E.2    SUBTRACTED IMPORTANCE SAMPLED SCORES (SISS)

Alberti et al. propose a novel *data unlearning* method, which aims to remove arbitrary images from the DM, while retaining overall generative capabilities. They perform a full fine-tuning on the U-Net, with minimization objective $\mathcal{L}_{s,\lambda}(\theta)$, defined as:

$$\mathbb{E}_{p_X(x)}\mathbb{E}_{p_A(a)}\mathbb{E}_{q_\lambda(m_t|x,a)}\left[\frac{n}{n-k}\frac{q(m_t|x)}{(1-\lambda)q(m_t|x) + \lambda q(m_t|a)}\left\|\frac{m_t - \gamma_t x}{\sigma_t} - \epsilon_\theta(m_t, t)\right\|_2^2\right.$$

$$\left. - (1+s)\frac{k}{n-k}\frac{q(m_t|a)}{(1-\lambda)q(m_t|x) + \lambda q(m_t|a)}\left\|\frac{m_t - \gamma_t a}{\sigma_t} - \epsilon_\theta(m_t, t)\right\|_2^2\right] \tag{4}$$

Here $X$ denotes the subset of the DM's training data of size $n$, $A \subset X$ a set of images to unlearn of size $k$, $q_\lambda(m_t|x,a) := (1-\lambda)q(m_t|x) + \lambda q(m_t|a)$ is a mixture distribution—a weighted average of data densities $q(m_t|x)$ and $q(m_t|a)$ parameterized by $\lambda \in [0; 1]$, and $\gamma_t$ and $\sigma_t$ are the DDPM (Ho et al., 2020) forward process parameters at timestep $t$. $\mathcal{L}_{s,\lambda}(\theta)$ provides a middle ground between naive deletion, *i.e.,* fine-tuning only on $X \setminus A$, and NegGrad (Golatkar et al., 2020), which performs gradient ascent on $A$, but is known for its instability. Parameter $\lambda$, with the default value of 0.5 in their work regulates how much SISS resembles the former and the latter removal methods. To increase the strength of the method, a *superfactor* hyperparameter $s > 0$ is introduced.

For SISS to work we need an access to the subset of the training data $X$, and images to remove $A$.

**Applying SISS to Memorization Mitigation** is straightforward. We use the default setting specified in the paper for Stable Diffusion v-1.4, *i.e.,* we apply SISS on a single memorized image at a time. Authors provide sets $A$ and $X$ for a small subset of 33 VM images they attempt to remove from the DM in their work. We extend them to the remaining VM samples for a fair comparison with other methods. Specifically, following their approach we add random tokens to the memorized prompt and generate 128 images from the original DM, where $A$ would consist of replicas of the memorized image, while $X$ would contain the remaining (non-memorized) images. We note that the method employed by the authors (originally proposed by Wen et al. (2024)) is unreliable in creating prompts that can generate both novel and memorized images, and for 9 of the memorized samples we fail to craft such (partially memorized) prompts even after varying the number of added tokens. Effectively, 9 out of 112 VM images remain in the model, since we are not able to provide the SISS method with the necessary $X$ and $A$ sets. Notably, the $X$ set is akin to set of *surrogate samples* used in our mitigation method, which we obtain reliably with a weak, pruning-based removal method (NeMo).

Once we have access to $X$ and $A$, we run SISS with the default hyperparameters for 35 update steps, with learning rate of $10^{-5}$, batch size of 1 and gradient accumulation of 16. For each image we start from the original U-Net, following the setup in the original work. We note that such a setup has a limited applicability in the real-world use-cases, as some DMs (specifically: SD-v1.4) may memorize more than one image.

# F ADDITIONAL DETAILS AND EXPERIMENTS ON ADVERSARIAL EMBEDDING OPTIMIZATION

In the following, we elaborate on the design of the adversarial optimization Eq. (2) used to obtain $y_{adv}$ to trigger generation of $x_{mem}$. First, we provide the algorithm in Alg. 1. Then we showcase that naive unconstrained optimization would yield False Positives ($y_{adv}$ that are capable of forcing the DM to generate *arbitrary* images), see Appx. F.1. Motivated by this finding, we experiment with the varying strength of the optimization, and arrive at the final constraint of 50 optimization steps, which allows us to successfully craft $y_{adv}$ if the optimization target (image) is memorized, and fail to provide $y_{adv}$ for all other (non-memorized) targets. We evaluate constraining schemes that work in the embedding space in Appx. F.3, and show that they are unsuccessful at preventing False Positives.

---

**Algorithm 1 Finding Dori 🐠 with Adversarial Embeddings**

---

**Input:**

      DM $\epsilon_\theta$                                         ▷ optionally after pruning-based mitigation applied

      Memorized training image $x_{\mathrm{mem}}$

      Memorized training prompt $p_{\mathrm{mem}}$

      Number of optimization steps $N$

      Learning rate $\eta$

**Output:**

      Adversarial embedding $y_{\mathrm{adv}}$

      $y_{\mathrm{adv}}^{(0)} \leftarrow \mathrm{encode\_text}(p_{\mathrm{mem}})$                    ▷ alternatively, initialize $y_{\mathrm{adv}}^{(0)} \sim \mathcal{N}(\mathbf{0}, \mathbf{I})$

      **for** $i \in \{1, \ldots, N\}$ **do**

          $\epsilon \sim \mathcal{N}(\mathbf{0}, \mathbf{I})$

          $t \sim \mathrm{Uniform}(\{1, \ldots, T\})$           ▷ sample discrete timestep from noise schedule

          $\tilde{x}_t \leftarrow \mathrm{add\_noise}(x_{\mathrm{mem}}, \epsilon, t)$      ▷ adding noise using the training noise scheduler

          $\hat{\epsilon} \leftarrow \epsilon_\theta \left( \tilde{x}_t, t, y_{\mathrm{adv}}^{(i-1)} \right)$

          $y_{\mathrm{adv}}^{(i)} \leftarrow y_{\mathrm{adv}}^{(i-1)} - \eta \cdot \nabla_{y_{\mathrm{adv}}^{(i-1)}} \|\epsilon - \hat{\epsilon}\|_2^2$     ▷ update adv. embedding with gradient descent

      **end for**

      **return** $y_{\mathrm{adv}}^{(N)}$

---

## F.1 CAN WE MAKE A DM TO OUTPUT ANY IMAGE WITH ADVERSARIAL EMBEDDINGS?

To assess whether Dori's ability to replicate memorized images is truly due to memorization, we also test whether adversarial text embeddings can be used to generate an arbitrary (non-memorized) image, as described in Sec. 3.1. Intuitively, we expect that we can force an 800M parameter model to produce a specific output vector (latent representation of an image) of size 16,384, given we perform an unconstrained gradient-based optimization of an input vector (text embedding) of size 59,136. In effect, if the model can be forced to produce arbitrary, non-memorized images using these embeddings, it would suggest that Dori is not exploiting memorization, but rather steering the model toward designated outputs—regardless of whether the content was memorized.

To generate non-memorized images with Dori, we sample 100 images from the COCO2014 training set and run the optimization for 1000 steps for each sampled image. Using the resulting adversarial embeddings, we generate 5 images per embedding with SD-v1.4 and compute the SSCD scores between the generated and original images. The SSCD scores for all examples exceed the memorization threshold of 0.7, and qualitatively, Fig. 6 shows that the images are replicated almost perfectly.

While this initially might seem as if Dori is not only replicating memorized samples, we demonstrate in Appx. F.2 that there is, in fact, a difference between triggering generation of memorized versus non-memorized content.

Original    Generated from $\mathbf{y}_{adv}$    Generated from $\mathbf{y}_{adv}$    Generated from $\mathbf{y}_{adv}$    Generated from $\mathbf{y}_{adv}$    Generated from $\mathbf{y}_{adv}$

Figure 6: **Arbitrary image replication.** We find that when pushed to the extreme, Dori search yields generations (columns from two to six from the left) of non-memorized data (first from the left).

## F.2 COMPARING BEHAVIORAL DIFFERENCES BETWEEN SETS

Our findings from Appx. F.1 undermine our adversarial-based memorization identification. In effect, it may seem that our results regarding NeMo and Wanda (Sec. 3.3) locality in the embedding space (Sec. 4.1). and locality in the model's activations (Sec. 4.2) and weights (Sec. 4.3) become invalid. Indeed, if we are able to trigger generation of *any* image, then we should not claim that NeMo and Wanda only conceal the memorization instead of fully removing it, and the findings regarding localization would be false, as the obtained adversarial embeddings $y_{adv}$ yield little information about how the model (and the embedding space) behaves when faced with memorized data.

To ensure correctness of our methodology, and—in effect—the findings, we investigate if there is any difference between the optimization process for memorized and other (non-memorized) images. We compare how the L2 norm of text embeddings progress during optimization, as well as how early we cross the 0.7 SSCD threshold. We analyze two sets of memorized images (100 VM samples and 100 TM samples), and a set of 100 images from COCO2014 train. Moreover, we analyze two sets of generated images from SD-v1.4: generated using 100 captions from COCO2014 train, and using 100 prompts of images that have been a subject of template memorization. The latter generated set addresses limitations of our detection metric—SSCD$_{Orig}$—which relies on all semantic and compositional parts of two compared images to match closely to cross the memorization threshold of 0.7. In case of template memorization, the model replicates only a part of a training image, *e.g.,* the background, specific objects, or replicates the semantic contents of the image, while varying features of low importance, like textures. We note that generated images and memorized template images will differ when it comes to the low importance features, effectively lowering SSCD score, however, the semantic composition of the generated images will match the one of memorized. We add generated images from 100 non-training prompts (COCO2014) to test the worst-case False Positive scenario of our method. If the model is *already able to* generate an image from some input, the optimization should converge the fastest for these images, even though they are neither part of the training data, nor memorized.

In Fig. 7 (right) we show how the SSCD score progresses with the optimization. We note that for verbatim memorization (and generated template memorization) we need only a handful of optimization steps to obtain $y_{adv}$ that reliably triggers generation of the images. For non-memorized data we reach SSCD above 0.7 after approximately 500 steps. Notably, we require as much as 200 steps to craft $y_{adv}$ that reliably produces generated (non-memorized) images.

These results show that the optimization process is indeed different for memorized images than for other images. Building on these findings we allow the optimization procedure to only perform 50 update steps—a value that guarantees generation of memorized images (if present in the model), while preventing False Positives, *i.e.,* generation of non-memorized images. This constraint ensures methodological correctness of our adversarial-based approach, and proves our results in Secs. 3.3 and 4 are valid. We also provide a qualitative comparison of generations with adversarial embeddings for memorized samples (after pruning) and non-memorized content in Fig. 8.

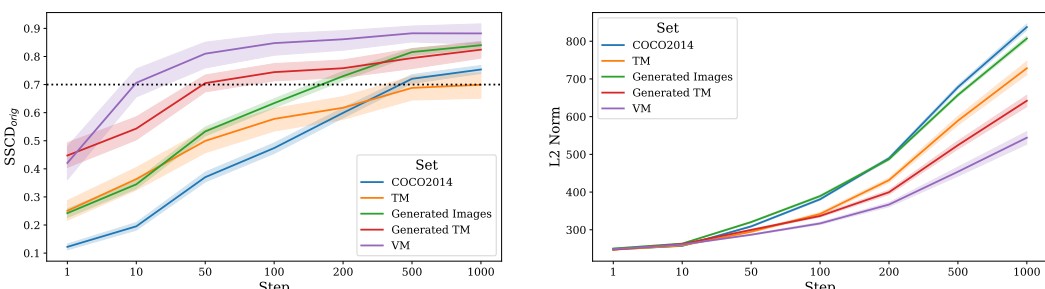

Figure 7: **Finding Dori with more optimization steps.** We note that our method starts producing False Positives, *i.e.,* replicating non-memorized data, only after 500 optimization steps (left). Notably, to achieve non-training data replication, the norm of the optimized embedding raises drastically (right).

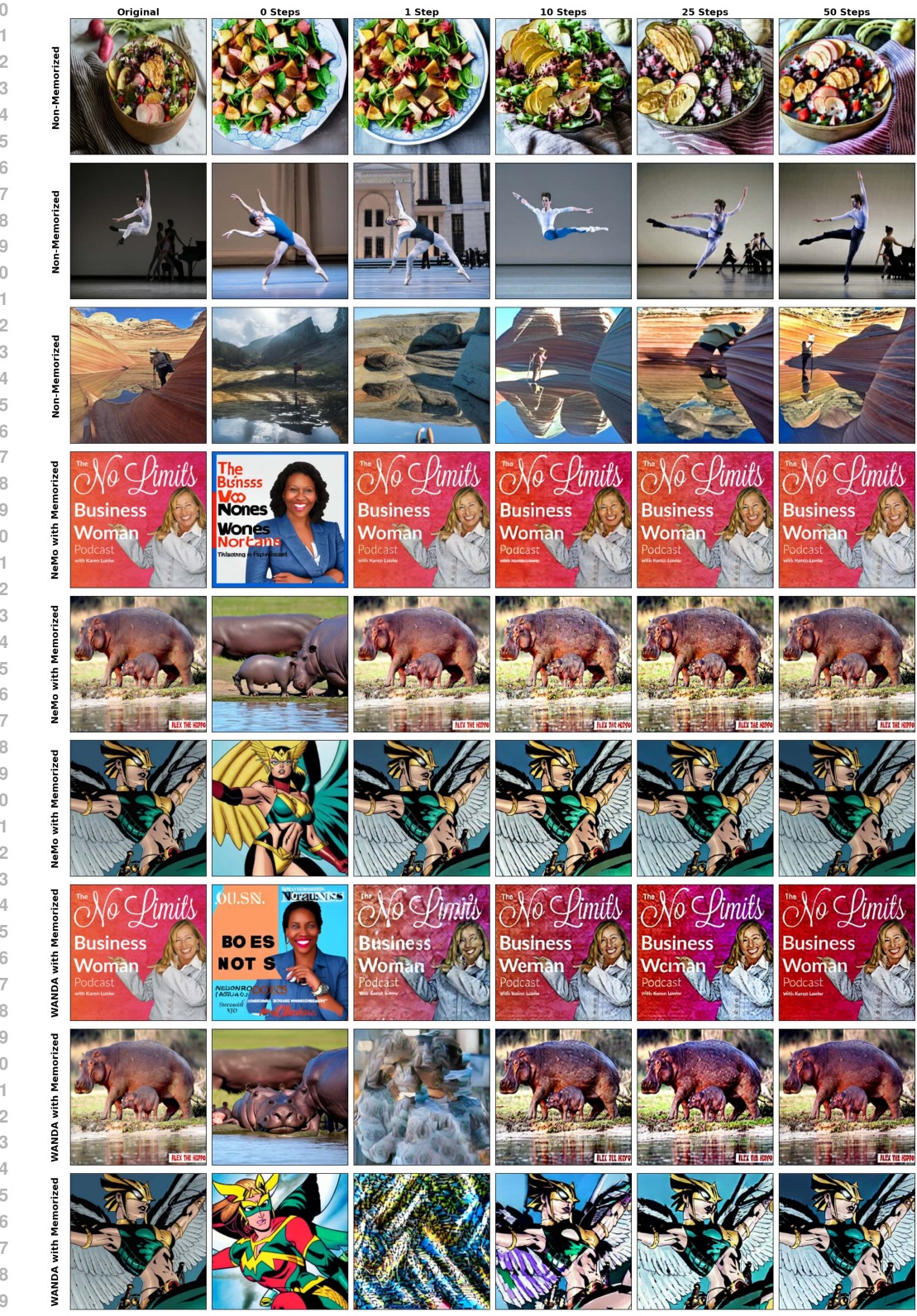

Figure 8: **Comparison of applying Dori to memorized and non-memorized content.** We qualitatively compare Dori with up to 50 optimization steps when applied to non-memorized samples (rows one to three) to our experimental setting, where we applied it to memorized content after using pruning with Nemo (rows four to six) and Wanda (rows seven to nine). Whereas Dori clearly triggers close replication of memorized content, it is not capable of replicating the non-memorized content after 50 optimization steps.

### F.3 CAN THE EMBEDDINGS THEMSELVES BE CONSTRAINED?

The findings in Appx. F.2 show that unconstrained optimization can lead to "replication" of an arbitrary image. In our work we default to restricting the number of optimization steps to prevent that false replication. We validate the soundness of that limit empirically, showing in Tab. 1 (second row) that we alleviate the problem. An alternative approach to prevent triggering arbitrary data would be to investigate the embedding space itself, and based on its characteristics, define meaningful constraints on the optimization.

We focus on L2 norms of the embeddings, as in Fig. 7 (right) we observe that the norms stay low for memorized content, while to replicate non-memorized content, the embeddings have to have their norm significantly increased. To get a glimpse at the embedding space, we embed 400k prompts from COCO 2014 train dataset, and note that the distribution of the norms is centered around 250, with standard deviation of around 2. Additionally, we perform gradient optimization of tokens to obtain minimal and maximal possible L2 norm of a text embedding. We find discrete inputs that lower L2 norm down to 220, and inputs that are able to increase the norm to 280.

With the limits of the embedding space established, we constrain our optimization to craft adversarial embeddings with L2 norm below the maximum possible value: 280. To this end, at each optimization step $i$, if $||\boldsymbol{y}_{adv}^{(i)}||_2^2 > 280$, we project it back to norm ball of 280 by $\boldsymbol{y}_{adv}^{(i)} \leftarrow \boldsymbol{y}_{adv}^{(i)} \cdot \frac{280}{||\boldsymbol{y}_{adv}^{(i)}||_2^2}$, an approach inspired by Projected Gradient Descent (Madry et al., 2018). Interestingly, even with such constraint, we are able obtain adversarial embeddings that trigger generation of non-memorized content, however, it requires more optimization steps, 2000 instead of 500. Next, we constrain the optimization even further, and expect embeddings to have norms smaller than 220—the lower boundary. Notably, memorized data is still replicated after merely 50 optimization steps *even after pruning*, while to replicate non-memorized data we need 10,000 steps.

We conclude that constraining adversarial embeddings might be a futile strategy to prevent non-memorized data replication, as even under heavy constraints, we are able to find embeddings that trigger generation of arbitrary images. Thus, we suggest limiting the number of update steps, and initializing optimization at a fixed point in text embedding space, to alleviate the issue with False Positives.

### F.4 HOW DO ADVERSARIAL EMBEDDINGS RELATE TO NATURAL PROMPTS?

In our experiments (Section 4.1 and Appx. J.1), we initialize the Dori optimization from either randomly sampled embeddings (Fig. 3), or text embeddings obtained from real, non-memorized text prompts (Fig. 10). In both cases, we observe that a memorized image can be successfully triggered from any place in the embedding space. Importantly, almost every adversarial embedding stays close to its initialization, be it a random vector or a real text embedding. This finding suggests that adversarial embeddings that cause replication of memorized images *do not* relate to any semantically meaningful text embeddings, and are instead uniformly scattered in the text embedding space.

# G CONCEPT UNLEARNING IS NOT SUITABLE FOR MEMORIZATION MITIGATION RESEARCH

Concept unlearning techniques aim to permanently suppress the generation of high-level concepts, such as nudity. Although related to the problem of undesired data memorization and replication, we show in this section that existing concept unlearning methods are failing to detect and prevent data replication.

## G.1 UNLEARNDIFFATK IS NOT CAPABLE OF FINDING DATA REPLICATION TRIGGERS

In Section 3, we introduce our method for searching adversarial embeddings that trigger data replication, even after pruning-based mitigation strategies are applied. While similar strategies have been proposed to critically evaluate concept unlearning techniques, we show that UnlearnDiffAtk (Zhang et al., 2024c), the state-of-the-art adversarial evaluation method for concept unlearning, is not suitable for identifying data replication triggers.

A naive approach to breaking memorization mitigation is to use UnlearnDiffAtk to find a prompt $p'$ that induces the generation of the memorized image $x$. We employ UnlearnDiffAtk in combination with GCG (Zou et al., 2023), an adversarial optimization strategy specifically designed for discrete prompt optimization. In particular, we apply GCG to solve $\min_{p'} \mathbb{E}t, \epsilon, ||\epsilon - \epsilon\theta(x_t, t, p')||_2^2$ after NeMo or Wanda pruning, aiming to identify a prompt $p'$ that triggers replication of $x$. This optimization objective corresponds to the one used by UnlearnDiffAtk.

UnlearnDiffAtk is unsuccessful in breaking pruning-based mitigation methods, *i.e.,* no memorized images are generated from prompt $p'$ after a mitigation is applied. Its failure highlights a core problem with concept removal methods: they focus on discrete text space, instead of exploring the full risk surface, like continuous text embeddings. While for content moderation, such a scope is sufficient, memorization is a privacy risk, which should be evaluated under the worst-case, unconstrained scenario.

## G.2 CONCEPT REMOVAL METHODS FAIL TO ROBUSTLY MITIGATE MEMORIZATION AGAINST DORI

Beyond using concept unlearning techniques to detect memorization in DMs, which we find to be ineffective in practice, a natural question is whether methods from concept unlearning can be leveraged to address data replication. However, we show that even state-of-the-art approaches are unsuitable for mitigating memorization, underscoring the need for novel methods specifically designed for preventing data replication.

We compare the mitigation capabilities of ESD (Gandikota et al., 2023) and Concept Ablation (Kumari et al., 2023) against our adversarial fine-tuning–based mitigation strategy, with results reported in Tab. 5. Both concept unlearning methods reduce the extent of data replication, as reflected by the SSCD scores. However, compared to our approach, a larger portion of memorization remains. Moreover, Concept Ablation degrades model utility, as evidenced by the higher FID score.

While these two methods seem to fail to fully remove memorization, they are also quite computational expensive. While ESD could be applied directly, guiding the model away from memorized samples, Concept Ablation requires paraphrasing memorized prompts via ChatGPT to generate a set of anchor prompts. For 7 highly memorized prompts, no valid paraphrase could be generated that did not trigger the memorized image, so these memorized images remained in the model—highlighting a limitation for using Concept Ablation to remove verbatim memorization compared to our method, which directly targets such cases. Since we did not observe strong success in removing memorization when running the methods for their original runtimes, we ran both of them significantly longer until we observed a plateau in their success: ESD was fine-tuned for 10k steps (10 times longer than in their paper) and Concept Ablation for 12k steps (15 times longer than in their paper). ESD completed in 1 hour and 7 minutes while Concept Ablation required 17 hours 45 minutes due to generating 1000 paraphrased images per prompt (roughly 8 minutes per prompt).

The most striking difference emerges when evaluating the methods with adversarial embeddings, denoted by the 🌐 symbol. While our mitigation method remains robust, both ESD and Concept

Table 5: **Full adversarial fine-tuning is a robust memorization mitigation technique.** Contrary to naive application of methods derived from concept removal area, we show that a tailored approach can successfully address memorization in a DM. By resigning from a faulty locality idea, and incorporating diverse set of triggers alongside full weight update we show that robust mitigation is possible.

| Setting | Memorization Type | $\downarrow \text{SSCD}_{\text{Orig}}$ | $\downarrow \text{SSCD}_{\text{Gen}}$ | $\uparrow A_{\text{CLIP}}$ | $\downarrow$MR | $\downarrow$FID |
|---|---|---|---|---|---|---|
| ESD (Gandikota et al., 2023) | Verbatim | $0.24 \pm 0.09$ | $0.27 \pm 0.11$ | $0.34 \pm 0.02$ | 0.11 | 13.45 |
| **ESD +** 🐟 | Verbatim | $\mathbf{0.90} \pm 0.04$ | $0.97 \pm 0.02$ | $0.33 \pm 0.01$ | **0.98** | |
| Concept Ablation (Kumari et al., 2023) | Verbatim | $0.39 \pm 0.18$ | $0.49 \pm 0.25$ | $0.34 \pm 0.02$ | 0.29 | 16.03 |
| **Concept Ablation +** 🐟 | Verbatim | $\mathbf{0.91} \pm 0.04$ | $0.96 \pm 0.02$ | $0.32 \pm 0.02$ | **0.97** | |
| Our Mitigation | Verbatim | $0.15 \pm 0.07$ | $0.15 \pm 0.07$ | $0.33 \pm 0.01$ | 0.00 | 13.61 |
| **Our Mitigation +** 🐟 | Verbatim | $\mathbf{0.36} \pm 0.14$ | $0.54 \pm 0.10$ | $0.30 \pm 0.02$ | **0.02** | |

Ablation expose memorized content under adversarial embeddings, suggesting that these methods conceal rather than truly mitigate memorization.

## H  ADDITIONAL EXPERIMENTS ON PRUNING-BASED MITIGATION

We find that close data replication is primarily triggered by VM prompts, while TM prompts lead to lower apparent replication. However, because TM prompts tend to produce partial replications that differ in non-semantic aspects of image composition, like the pattern on a phone case, SSCD-based metrics are less informative in this case than for VM prompts.

We extend our evaluation by **diversity**, since typically, memorized images are consistently replicated, regardless of the choice of the initial noise $\epsilon$. Conversely, for any other input, images generated by a DM are diverse under varying initial noise. We quantify diversity as the average pairwise cosine similarity $D_{\text{SSCD}}$ between SSCD embeddings of images generated from the same input but different initial noise. Images generated after mitigation should exhibit greater diversity, indicated by lower $D_{\text{SSCD}}$ values.

### H.1  HYPERPARAMETERS

We followed the default hyperparameters for NeMo and Wanda reported in the respective publications.

**NeMo:** We set the memorization score threshold to $\tau_{\text{mem}} = 0.428$, corresponding to the mean SSIM score plus one standard deviation, as measured on a holdout set of 50,000 LAION prompts. For the stronger variant of NeMo, reported in Table 13, we lowered the threshold to $\tau_{\text{mem}} = 0.288$, which corresponds to the mean SSIM score minus one standard deviation. While we follow the original evaluation procedure by individually identifying and disabling neurons for each memorized prompt, we compute the FID and KID metrics by simultaneously deactivating all neurons identified for VM and TM prompts, respectively. This approach provides a more consistent estimate of the pruning's overall impact on model utility.

**Wanda:** For Wanda, we follow the experimental setup of Chavhan et al. (2024). Specifically, we use all 500 memorized prompts to identify weights in the second fully connected layer of the cross-attention mechanism. As in Chavhan et al. (2024), we select the top $1\%$ of weights with the highest Wanda scores. These weights are then aggregated across the first 10 time steps and pruned to mitigate memorization. Additional results for identifying weights using Wanda per memorized prompt, for 10 and for all 500 memorized prompts, can be seen in Table 8. Results for different number of time steps and different values of sparsity can be found in Table 9 and Table 10, respectively.

### H.2  SENSITIVITY ANALYSIS OF ADVERSARIAL EMBEDDING OPTIMIZATION

In Table 13, we compare the results of NeMo and Wanda with and without adversarial embedding optimization. Application of adversarial embeddings is denoted by 🐟. Additionally, we repeat the experiments using different numbers of adversarial optimization steps, denoted by *Adv. Steps* in the table. All optimizations are initialized from the memorized training embedding. Notably, a single

optimization step is already sufficient to circumvent the mitigation introduced by NeMo. In the case of Wanda, approximately 25 optimization steps are required before clear replication is triggered.

In addition to the main paper, we also report results for TM prompts. While the SSCD scores are substantially lower than those for VM prompts, we note that replication of memorized content is still possible. However, the SSCD score fails to adequately capture TM memorization due to the semantic variations in the generated images.

At the bottom of the table, we also report results for adversarial embedding optimization applied to non-memorized training images, to evaluate whether replication can be triggered for non-memorized content. However, even after 150 optimization steps, SSCD scores remain below the memorization threshold of 0.7.

Table 6: Comparison of different numbers of adversarial embedding optimization steps. Embeddings are initialized with their corresponding *training prompt embeddings*. 🐟 denotes the application of adversarial embeddings.

| Setting | Adv. Steps | Memorization Type | ↓ $SSCD_{Orig}$ | ↓ $SSCD_{Gen}$ | ↓ $D_{SSCD}$ | ↑ $A_{CLIP}$ | ↓MR | ↓FID | ↓KID |
|---|---|---|---|---|---|---|---|---|---|
| No Mitigation | – | Verbatim | 0.90 ± 0.04 | N/A | 1.00 ± 0.00 | 0.33 ± 0.01 | 1.00 | 14.44 | 0.0061 |
| | – | Template | 0.17 ± 0.09 | N/A | 0.90 ± 0.08 | 0.33 ± 0.02 | 1.00 | | |
| NeMo (Hintersdorf et al., 2024) | – | Verbatim | 0.33 ± 0.18 | 0.40 ± 0.21 | 0.46 ± 0.13 | 0.34 ± 0.02 | 0.20 | 15.16 | 0.0061 |
| | – | Template | 0.23 ± 0.08 | 0.54 ± 0.28 | 0.55 ± 0.10 | 0.34 ± 0.03 | 0.15 | 18.97 | 0.0048 |
| Wanda (Chavhan et al., 2024) | – | Verbatim | 0.20 ± 0.08 | 0.21 ± 0.09 | 0.37 ± 0.07 | 0.34 ± 0.02 | 0.00 | 16.86 | 0.0065 |
| | – | Template | 0.17 ± 0.05 | 0.18 ± 0.08 | 0.38 ± 0.09 | 0.34 ± 0.03 | 0.00 | 17.51 | 0.0070 |
| **NeMo + 🐟** | 1 | Verbatim | 0.86 ± 0.07 | 0.94 ± 0.04 | 1.00 ± 0.00 | 0.32 ± 0.01 | 0.73 | 15.16 | 0.0061 |
| | | Template | 0.28 ± 0.11 | 0.51 ± 0.28 | 0.62 ± 0.21 | 0.33 ± 0.02 | 0.21 | 18.97 | 0.0048 |
| **NeMo + 🐟** | 10 | Verbatim | 0.81 ± 0.06 | 0.88 ± 0.05 | 0.99 ± 0.01 | 0.32 ± 0.01 | 0.79 | 15.16 | 0.0061 |
| | | Template | 0.42 ± 0.13 | 0.21 ± 0.15 | 0.72 ± 0.13 | 0.32 ± 0.02 | 0.14 | 18.97 | 0.0048 |
| **NeMo + 🐟** | 25 | Verbatim | 0.88 ± 0.04 | 0.95 ± 0.03 | 1.00 ± 0.00 | 0.32 ± 0.01 | 0.94 | 15.16 | 0.0061 |
| | | Template | 0.50 ± 0.12 | 0.20 ± 0.15 | 0.75 ± 0.10 | 0.32 ± 0.02 | 0.13 | 18.97 | 0.0048 |
| **NeMo + 🐟** | 50 | Verbatim | 0.91 ± 0.03 | 0.97 ± 0.02 | 1.00 ± 0.00 | 0.33 ± 0.02 | 0.99 | 15.16 | 0.0061 |
| | | Template | 0.55 ± 0.12 | 0.17 ± 0.12 | 0.79 ± 0.11 | 0.32 ± 0.02 | 0.12 | 18.97 | 0.0048 |
| **NeMo + 🐟** | 100 | Verbatim | 0.93 ± 0.02 | 0.96 ± 0.02 | 1.00 ± 0.00 | 0.32 ± 0.02 | 1.00 | 15.16 | 0.0061 |
| | | Template | 0.60 ± 0.14 | 0.17 ± 0.12 | 0.86 ± 0.09 | 0.32 ± 0.02 | 0.10 | 18.97 | 0.0048 |
| **NeMo + 🐟** | 150 | Verbatim | 0.92 ± 0.02 | 0.96 ± 0.02 | 1.00 ± 0.00 | 0.32 ± 0.02 | 0.99 | 15.16 | 0.0061 |
| | | Template | 0.65 ± 0.16 | 0.17 ± 0.12 | 0.93 ± 0.06 | 0.32 ± 0.02 | 0.08 | 18.97 | 0.0048 |
| **NeMo (strong, $\tau_{mem} = 0.288$) + 🐟** | 50 | Verbatim | 0.91 ± 0.03 | 0.96 ± 0.02 | 1.00 ± 0.00 | 0.33 ± 0.02 | 0.88 | 14.92 | 0.0064 |
| | | Template | 0.55 ± 0.12 | 0.19 ± 0.12 | 0.79 ± 0.10 | 0.32 ± 0.02 | 0.15 | 18.85 | 0.0042 |
| **Wanda + 🐟** | 1 | Verbatim | 0.11 ± 0.05 | 0.11 ± 0.06 | 0.58 ± 0.08 | 0.24 ± 0.04 | 0.02 | 16.86 | 0.0065 |
| | | Template | 0.19 ± 0.07 | 0.16 ± 0.07 | 0.51 ± 0.11 | 0.32 ± 0.03 | 0.00 | 17.51 | 0.0070 |
| **Wanda + 🐟** | 10 | Verbatim | 0.58 ± 0.11 | 0.64 ± 0.11 | 0.76 ± 0.14 | 0.31 ± 0.02 | 0.18 | 16.86 | 0.0065 |
| | | Template | 0.37 ± 0.10 | 0.17 ± 0.09 | 0.61 ± 0.12 | 0.32 ± 0.03 | 0.02 | 17.51 | 0.0070 |
| **Wanda + 🐟** | 25 | Verbatim | 0.69 ± 0.07 | 0.77 ± 0.05 | 0.90 ± 0.07 | 0.32 ± 0.02 | 0.49 | 16.86 | 0.0065 |
| | | Template | 0.45 ± 0.11 | 0.17 ± 0.10 | 0.70 ± 0.09 | 0.32 ± 0.03 | 0.07 | 17.51 | 0.0070 |
| **Wanda + 🐟** | 50 | Verbatim | 0.76 ± 0.05 | 0.82 ± 0.05 | 0.96 ± 0.02 | 0.32 ± 0.01 | 0.73 | 16.86 | 0.0065 |
| | | Template | 0.51 ± 0.11 | 0.16 ± 0.09 | 0.75 ± 0.08 | 0.32 ± 0.02 | 0.16 | 17.51 | 0.0070 |
| **Wanda + 🐟** | 100 | Verbatim | 0.80 ± 0.05 | 0.85 ± 0.04 | 0.98 ± 0.01 | 0.32 ± 0.02 | 0.86 | 16.86 | 0.0065 |
| | | Template | 0.53 ± 0.12 | 0.15 ± 0.08 | 0.78 ± 0.08 | 0.32 ± 0.02 | 0.26 | 17.51 | 0.0070 |
| **Wanda + 🐟** | 150 | Verbatim | 0.81 ± 0.04 | 0.85 ± 0.04 | 0.99 ± 0.01 | 0.32 ± 0.02 | 0.91 | 16.86 | 0.0065 |
| | | Template | 0.54 ± 0.14 | 0.15 ± 0.09 | 0.82 ± 0.08 | 0.32 ± 0.02 | 0.32 | 17.51 | 0.0070 |
| **Non-Memorized Images** | – | None | 0.17 ± 0.05 | N/A | 0.35 ± 0.06 | 0.35 ± 0.02 | 0.00 | 14.44 | 0.0061 |
| **Non-Memorized Images + 🐟** | 1 | None | 0.17 ± 0.04 | N/A | 0.34 ± 0.06 | 0.34 ± 0.02 | 0.00 | 14.44 | 0.0061 |
| **Non-Memorized Images + 🐟** | 10 | None | 0.28 ± 0.05 | N/A | 0.48 ± 0.06 | 0.32 ± 0.02 | 0.00 | 14.44 | 0.0061 |
| **Non-Memorized Images + 🐟** | 25 | None | 0.39 ± 0.06 | N/A | 0.58 ± 0.06 | 0.32 ± 0.02 | 0.00 | 14.44 | 0.0061 |
| **Non-Memorized Images + 🐟** | 50 | None | 0.48 ± 0.06 | N/A | 0.67 ± 0.07 | 0.32 ± 0.02 | 0.02 | 14.44 | 0.0061 |
| **Non-Memorized Images + 🐟** | 100 | None | 0.58 ± 0.06 | N/A | 0.79 ± 0.07 | 0.32 ± 0.02 | 0.08 | 14.44 | 0.0061 |
| **Non-Memorized Images + 🐟** | 150 | None | 0.65 ± 0.06 | N/A | 0.88 ± 0.06 | 0.32 ± 0.02 | 0.25 | 14.44 | 0.0061 |

## H.3 STARTING EMBEDDING OPTIMIZATION FROM RANDOM EMBEDDINGS

We repeat the experiments on adversarial embedding optimization, but instead of initializing from the memorized training prompt embedding, we start each optimization from random Gaussian noise. Remarkably, the results closely match those obtained when initializing from the memorized prompt, indicating that data replication can be triggered from various positions in the embedding space.

Table 7: Comparison of different numbers of adversarial embedding optimization steps. Embeddings are *initialized randomly*. 🐟 denotes the application of adversarial embeddings.

| Setting | Adv. Steps | Memorization Type | ↓ $SSCD_{Orig}$ | ↓ $SSCD_{Gen}$ | ↓ $D_{SSCD}$ | ↑ $A_{CLIP}$ | ↓MR |
|---|---|---|---|---|---|---|---|
| **NeMo +** 🐟 | 1 | Verbatim | $0.07 \pm 0.02$ | $0.06 \pm 0.02$ | $0.31 \pm 0.06$ | $0.21 \pm 0.02$ | 0.00 |
| | | Template | $0.10 \pm 0.03$ | $0.10 \pm 0.03$ | $0.26 \pm 0.05$ | $0.28 \pm 0.02$ | 0.00 |
| **NeMo +** 🐟 | 10 | Verbatim | $0.81 \pm 0.06$ | $0.89 \pm 0.05$ | $0.99 \pm 0.01$ | $0.32 \pm 0.01$ | 0.79 |
| | | Template | $0.42 \pm 0.13$ | $0.21 \pm 0.15$ | $0.72 \pm 0.14$ | $0.32 \pm 0.02$ | 0.14 |
| **NeMo +** 🐟 | 25 | Verbatim | $0.88 \pm 0.04$ | $0.95 \pm 0.03$ | $1.00 \pm 0.00$ | $0.32 \pm 0.01$ | 0.94 |
| | | Template | $0.50 \pm 0.13$ | $0.20 \pm 0.15$ | $0.75 \pm 0.11$ | $0.32 \pm 0.02$ | 0.13 |
| **NeMo +** 🐟 | 50 | Verbatim | $0.91 \pm 0.03$ | $0.97 \pm 0.02$ | $1.00 \pm 0.00$ | $0.33 \pm 0.02$ | 0.99 |
| | | Template | $0.55 \pm 0.12$ | $0.18 \pm 0.12$ | $0.79 \pm 0.10$ | $0.32 \pm 0.02$ | 0.12 |
| **NeMo +** 🐟 | 100 | Verbatim | $0.93 \pm 0.02$ | $0.96 \pm 0.02$ | $1.00 \pm 0.00$ | $0.33 \pm 0.02$ | 1.00 |
| | | Template | $0.60 \pm 0.14$ | $0.17 \pm 0.12$ | $0.87 \pm 0.10$ | $0.32 \pm 0.02$ | 0.10 |
| **NeMo +** 🐟 | 150 | Verbatim | $0.92 \pm 0.02$ | $0.96 \pm 0.02$ | $1.00 \pm 0.00$ | $0.32 \pm 0.02$ | 1.00 |
| | | Template | $0.64 \pm 0.15$ | $0.16 \pm 0.12$ | $0.93 \pm 0.06$ | $0.32 \pm 0.02$ | 0.07 |
| **Wanda +** 🐟 | 1 | Verbatim | $0.07 \pm 0.02$ | $0.07 \pm 0.02$ | $0.38 \pm 0.10$ | $0.21 \pm 0.02$ | 0.00 |
| | | Template | $0.07 \pm 0.02$ | $0.07 \pm 0.02$ | $0.43 \pm 0.12$ | $0.22 \pm 0.02$ | 0.00 |
| **Wanda +** 🐟 | 10 | Verbatim | $0.28 \pm 0.10$ | $0.28 \pm 0.13$ | $0.42 \pm 0.07$ | $0.29 \pm 0.03$ | 0.02 |
| | | Template | $0.27 \pm 0.09$ | $0.12 \pm 0.05$ | $0.47 \pm 0.16$ | $0.29 \pm 0.02$ | 0.00 |
| **Wanda +** 🐟 | 25 | Verbatim | $0.68 \pm 0.10$ | $0.70 \pm 0.10$ | $0.84 \pm 0.13$ | $0.31 \pm 0.02$ | 0.37 |
| | | Template | $0.46 \pm 0.11$ | $0.15 \pm 0.08$ | $0.68 \pm 0.09$ | $0.31 \pm 0.02$ | 0.10 |
| **Wanda +** 🐟 | 50 | Verbatim | $0.80 \pm 0.06$ | $0.84 \pm 0.06$ | $0.97 \pm 0.02$ | $0.32 \pm 0.02$ | 0.78 |
| | | Template | $0.54 \pm 0.12$ | $0.14 \pm 0.08$ | $0.76 \pm 0.10$ | $0.31 \pm 0.02$ | 0.26 |
| **Wanda +** 🐟 | 100 | Verbatim | $0.85 \pm 0.05$ | $0.87 \pm 0.05$ | $0.99 \pm 0.01$ | $0.32 \pm 0.02$ | 0.89 |
| | | Template | $0.60 \pm 0.15$ | $0.15 \pm 0.10$ | $0.84 \pm 0.10$ | $0.32 \pm 0.02$ | 0.37 |
| **Wanda +** 🐟 | 150 | Verbatim | $0.86 \pm 0.04$ | $0.87 \pm 0.04$ | $0.99 \pm 0.00$ | $0.32 \pm 0.01$ | 0.92 |
| | | Template | $0.64 \pm 0.16$ | $0.16 \pm 0.10$ | $0.90 \pm 0.08$ | $0.31 \pm 0.03$ | 0.44 |

## H.4 Additional Hyperparameter Evaluation for Wanda

Table 8: As shown, applying Wanda across all prompts is less effective at mitigating memorization compared to applying it individually per prompt. However, as discussed in Appx. H.7, applying Wanda per prompt and aggregating the found neurons over all 500 prompts comes at the high cost of reduced overall performance because of so many weights being pruned. In the setting with 10 prompts, we randomly sample 10 prompts across 5 different seeds and report the average results. This setup proves less effective at mitigating memorization than using the full set of 500 prompts.

| Setting | Memorization Type | ↓ $\text{SSCD}_{\text{Orig}}$ | ↓ $\text{SSCD}_{\text{Gen}}$ | ↓ $D_{\text{SSCD}}$ | ↑ $A_{\text{CLIP}}$ | ↓MR |
|---|---|---|---|---|---|---|
| Wanda (Chavhan et al., 2024) per Prompt | Verbatim | $0.11 \pm 0.03$ | $0.12 \pm 0.03$ | $0.27 \pm 0.06$ | $0.32 \pm 0.02$ | 0.00 |
| | Template | $0.14 \pm 0.04$ | $0.13 \pm 0.04$ | $0.35 \pm 0.10$ | $0.32 \pm 0.03$ | 0.00 |
| Wanda (Chavhan et al., 2024) 10 Prompts | Verbatim | $0.22 \pm 0.10$ | $0.24 \pm 0.11$ | $0.41 \pm 0.09$ | $0.34 \pm 0.02$ | 0.01 |
| | Template | $0.19 \pm 0.06$ | $0.24 \pm 0.13$ | $0.42 \pm 0.10$ | $0.34 \pm 0.03$ | 0.01 |
| Wanda (Chavhan et al., 2024) all Prompts | Verbatim | $0.20 \pm 0.08$ | $0.21 \pm 0.09$ | $0.37 \pm 0.07$ | $0.34 \pm 0.02$ | 0.00 |
| | Template | $0.17 \pm 0.05$ | $0.18 \pm 0.08$ | $0.38 \pm 0.09$ | $0.34 \pm 0.03$ | 0.00 |
| **Wanda per Prompt +** 🐠 | Verbatim | $0.69 \pm 0.07$ | $0.76 \pm 0.06$ | $0.91 \pm 0.05$ | $0.32 \pm 0.02$ | 0.46 |
| | Template | $0.52 \pm 0.10$ | $0.17 \pm 0.10$ | $0.75 \pm 0.07$ | $0.32 \pm 0.02$ | 0.15 |
| **Wanda 10 Prompts +** 🐠 | Verbatim | $0.75 \pm 0.06$ | $0.81 \pm 0.05$ | $0.97 \pm 0.02$ | $0.32 \pm 0.01$ | 0.71 |
| | Template | $0.50 \pm 0.11$ | $0.16 \pm 0.09$ | $0.74 \pm 0.09$ | $0.32 \pm 0.02$ | 0.15 |
| **Wanda all Prompts +** 🐠 | Verbatim | $0.76 \pm 0.05$ | $0.82 \pm 0.05$ | $0.96 \pm 0.02$ | $0.32 \pm 0.01$ | 0.73 |
| | Template | $0.51 \pm 0.11$ | $0.16 \pm 0.09$ | $0.75 \pm 0.08$ | $0.32 \pm 0.02$ | 0.16 |

Table 9: Even when applying Wanda Chavhan et al. (2024) with a higher number of time steps it is still possible to break it using Dori.

| Setting | Number of Timesteps | Memorization Type | ↓ $\text{SSCD}_{\text{Orig}}$ | ↓ $\text{SSCD}_{\text{Gen}}$ | ↓ $D_{\text{SSCD}}$ | ↑ $A_{\text{CLIP}}$ | ↓MR |
|---|---|---|---|---|---|---|---|
| **Wanda** | 1 | Verbatim | $0.22 \pm 0.08$ | $0.24 \pm 0.09$ | $0.38 \pm 0.08$ | $0.34 \pm 0.02$ | 0.01 |
| | | Template | $0.18 \pm 0.05$ | $0.20 \pm 0.08$ | $0.39 \pm 0.09$ | $0.33 \pm 0.02$ | 0.00 |
| | 10 | Verbatim | $0.20 \pm 0.08$ | $0.21 \pm 0.09$ | $0.37 \pm 0.07$ | $0.34 \pm 0.02$ | 0.00 |
| | | Template | $0.17 \pm 0.05$ | $0.18 \pm 0.08$ | $0.38 \pm 0.09$ | $0.34 \pm 0.03$ | 0.00 |
| | 20 | Verbatim | $0.20 \pm 0.08$ | $0.21 \pm 0.07$ | $0.39 \pm 0.08$ | $0.34 \pm 0.03$ | 0.00 |
| | | Template | $0.17 \pm 0.04$ | $0.17 \pm 0.06$ | $0.39 \pm 0.09$ | $0.33 \pm 0.03$ | 0.00 |
| | 30 | Verbatim | $0.20 \pm 0.08$ | $0.20 \pm 0.07$ | $0.37 \pm 0.07$ | $0.34 \pm 0.02$ | 0.00 |
| | | Template | $0.18 \pm 0.05$ | $0.17 \pm 0.06$ | $0.39 \pm 0.10$ | $0.33 \pm 0.03$ | 0.00 |
| | 40 | Verbatim | $0.20 \pm 0.08$ | $0.21 \pm 0.07$ | $0.38 \pm 0.07$ | $0.34 \pm 0.02$ | 0.00 |
| | | Template | $0.18 \pm 0.05$ | $0.17 \pm 0.07$ | $0.39 \pm 0.10$ | $0.33 \pm 0.03$ | 0.00 |
| | 50 | Verbatim | $0.20 \pm 0.07$ | $0.20 \pm 0.07$ | $0.38 \pm 0.07$ | $0.34 \pm 0.02$ | 0.00 |
| | | Template | $0.17 \pm 0.04$ | $0.17 \pm 0.06$ | $0.41 \pm 0.11$ | $0.33 \pm 0.03$ | 0.00 |
| **Wanda +** 🐠 | 1 | Verbatim | $0.77 \pm 0.05$ | $0.82 \pm 0.05$ | $0.97 \pm 0.01$ | $0.32 \pm 0.01$ | 0.79 |
| | | Template | $0.51 \pm 0.11$ | $0.16 \pm 0.09$ | $0.75 \pm 0.09$ | $0.32 \pm 0.02$ | 0.18 |
| | 10 | Verbatim | $0.76 \pm 0.05$ | $0.82 \pm 0.05$ | $0.96 \pm 0.02$ | $0.32 \pm 0.01$ | 0.73 |
| | | Template | $0.52 \pm 0.11$ | $0.16 \pm 0.09$ | $0.75 \pm 0.09$ | $0.32 \pm 0.02$ | 0.15 |
| | 20 | Verbatim | $0.76 \pm 0.05$ | $0.82 \pm 0.05$ | $0.96 \pm 0.02$ | $0.32 \pm 0.01$ | 0.72 |
| | | Template | $0.52 \pm 0.11$ | $0.16 \pm 0.09$ | $0.75 \pm 0.09$ | $0.32 \pm 0.02$ | 0.17 |
| | 30 | Verbatim | $0.74 \pm 0.05$ | $0.80 \pm 0.05$ | $0.96 \pm 0.02$ | $0.32 \pm 0.02$ | 0.68 |
| | | Template | $0.51 \pm 0.11$ | $0.16 \pm 0.09$ | $0.75 \pm 0.09$ | $0.32 \pm 0.02$ | 0.14 |
| | 40 | Verbatim | $0.74 \pm 0.05$ | $0.81 \pm 0.05$ | $0.96 \pm 0.02$ | $0.32 \pm 0.01$ | 0.69 |
| | | Template | $0.50 \pm 0.12$ | $0.16 \pm 0.09$ | $0.74 \pm 0.09$ | $0.32 \pm 0.02$ | 0.18 |
| | 50 | Verbatim | $0.73 \pm 0.06$ | $0.79 \pm 0.05$ | $0.96 \pm 0.02$ | $0.32 \pm 0.02$ | 0.59 |
| | | Template | $0.49 \pm 0.12$ | $0.16 \pm 0.09$ | $0.74 \pm 0.09$ | $0.32 \pm 0.02$ | 0.14 |

## H.5 INCREASING SPARSITY FOR WANDA

Table 10: Applying Wanda Chavhan et al. (2024) with higher sparsity does not change the fact that the method seems to only conceal memorization instead of completely removing it from the model. Increasing the sparsity also comes at the cost of reduced image quality, as the FID and the KID values suggest.

| Setting | Sparsity | Memorization Type | ↓ $\text{SSCD}_{\text{Orig}}$ | ↓ $\text{SSCD}_{\text{Gen}}$ | ↓ $D_{\text{SSCD}}$ | ↑ $A_{\text{CLIP}}$ | ↓MR | FID | KID |
|---|---|---|---|---|---|---|---|---|---|
| Wanda | 1% | Verbatim | $0.20 \pm 0.08$ | $0.21 \pm 0.09$ | $0.37 \pm 0.07$ | $0.34 \pm 0.02$ | 0.00 | 16.86 | 0.0067 |
| | | Template | $0.17 \pm 0.05$ | $0.18 \pm 0.08$ | $0.38 \pm 0.09$ | $0.34 \pm 0.03$ | 0.00 | 17.51 | 0.0068 |
| | 2% | Verbatim | $0.19 \pm 0.07$ | $0.20 \pm 0.07$ | $0.36 \pm 0.07$ | $0.33 \pm 0.02$ | 0.00 | 18.17 | 0.0066 |
| | | Template | $0.17 \pm 0.05$ | $0.16 \pm 0.07$ | $0.38 \pm 0.08$ | $0.33 \pm 0.03$ | 0.00 | 19.55 | 0.0073 |
| | 3% | Verbatim | $0.17 \pm 0.07$ | $0.17 \pm 0.06$ | $0.34 \pm 0.06$ | $0.33 \pm 0.02$ | 0.00 | 20.37 | 0.0075 |
| | | Template | $0.16 \pm 0.05$ | $0.15 \pm 0.06$ | $0.38 \pm 0.09$ | $0.32 \pm 0.03$ | 0.00 | 22.40 | 0.0086 |
| | 4% | Verbatim | $0.17 \pm 0.06$ | $0.17 \pm 0.05$ | $0.34 \pm 0.06$ | $0.33 \pm 0.02$ | 0.00 | 23.07 | 0.0088 |
| | | Template | $0.14 \pm 0.05$ | $0.14 \pm 0.06$ | $0.37 \pm 0.09$ | $0.32 \pm 0.03$ | 0.00 | 24.69 | 0.0097 |
| | 5% | Verbatim | $0.15 \pm 0.05$ | $0.16 \pm 0.05$ | $0.32 \pm 0.05$ | $0.32 \pm 0.02$ | 0.00 | 25.53 | 0.0102 |
| | | Template | $0.13 \pm 0.05$ | $0.14 \pm 0.05$ | $0.39 \pm 0.10$ | $0.32 \pm 0.03$ | 0.00 | 26.61 | 0.0106 |
| | 10% | Verbatim | $0.12 \pm 0.03$ | $0.13 \pm 0.04$ | $0.33 \pm 0.06$ | $0.31 \pm 0.02$ | 0.00 | 37.34 | 0.0168 |
| | | Template | $0.11 \pm 0.04$ | $0.13 \pm 0.04$ | $0.39 \pm 0.10$ | $0.30 \pm 0.03$ | 0.00 | 36.69 | 0.0166 |
| **Wanda +** 🐟 | 1% | Verbatim | $0.76 \pm 0.06$ | $0.82 \pm 0.05$ | $0.96 \pm 0.02$ | $0.32 \pm 0.01$ | 0.73 | 16.86 | 0.0067 |
| | | Template | $0.51 \pm 0.12$ | $0.16 \pm 0.09$ | $0.75 \pm 0.09$ | $0.32 \pm 0.02$ | 0.16 | 17.51 | 0.0068 |
| | 2% | Verbatim | $0.71 \pm 0.07$ | $0.76 \pm 0.06$ | $0.90 \pm 0.05$ | $0.32 \pm 0.02$ | 0.54 | 18.17 | 0.0066 |
| | | Template | $0.45 \pm 0.11$ | $0.15 \pm 0.08$ | $0.71 \pm 0.08$ | $0.31 \pm 0.03$ | 0.09 | 19.55 | 0.0073 |
| | 3% | Verbatim | $0.65 \pm 0.08$ | $0.73 \pm 0.06$ | $0.87 \pm 0.06$ | $0.31 \pm 0.02$ | 0.32 | 20.37 | 0.0075 |
| | | Template | $0.39 \pm 0.10$ | $0.13 \pm 0.07$ | $0.70 \pm 0.08$ | $0.31 \pm 0.03$ | 0.04 | 22.40 | 0.0086 |
| | 4% | Verbatim | $0.62 \pm 0.08$ | $0.66 \pm 0.08$ | $0.81 \pm 0.08$ | $0.31 \pm 0.02$ | 0.17 | 23.07 | 0.0088 |
| | | Template | $0.35 \pm 0.12$ | $0.14 \pm 0.07$ | $0.68 \pm 0.08$ | $0.31 \pm 0.03$ | 0.03 | 24.69 | 0.0097 |
| | 5% | Verbatim | $0.56 \pm 0.10$ | $0.62 \pm 0.09$ | $0.77 \pm 0.10$ | $0.31 \pm 0.02$ | 0.08 | 25.53 | 0.0102 |
| | | Template | $0.29 \pm 0.10$ | $0.13 \pm 0.07$ | $0.68 \pm 0.09$ | $0.30 \pm 0.04$ | 0.02 | 26.61 | 0.0106 |
| | 10% | Verbatim | $0.40 \pm 0.13$ | $0.45 \pm 0.14$ | $0.67 \pm 0.10$ | $0.30 \pm 0.02$ | 0.01 | 37.34 | 0.0168 |
| | | Template | $0.22 \pm 0.08$ | $0.12 \pm 0.06$ | $0.63 \pm 0.10$ | $0.28 \pm 0.04$ | 0.01 | 36.69 | 0.0166 |

## H.6 ITERATIVE APPLICATION OF NEMO

Table 11: We apply NeMo Hintersdorf et al. (2024) iteratively such that after each round of pruning, we search for new adversarial embeddings that can still trigger memorization, and then apply NeMo again to prune the newly identified weights. Despite multiple iterations, this process does not completely eliminate memorization, as adversarial embeddings can still uncover residual memorized content. Due to the high computational cost of repeated NeMo applications and searching for adversarial embeddings, we focus our analysis on prompts known to be verbatim memorized. In some cases, after several iterations, NeMo no longer detects any memorization. When this happens, we analyze the outputs generated from the adversarial embeddings that NeMo failed to flag.

| Method | Iterations | ↓ $\text{SSCD}_{\text{Orig}}$ | ↓ $\text{SSCD}_{\text{Gen}}$ | ↓ $D_{\text{SSCD}}$ | ↑ $A_{\text{CLIP}}$ | ↓MR |
|---|---|---|---|---|---|---|
| | 1 | $0.92 \pm 0.03$ | $0.94 \pm 0.03$ | $1.00 \pm 0.00$ | $0.33 \pm 0.02$ | 0.99 |
| | 2 | $0.92 \pm 0.03$ | $0.94 \pm 0.03$ | $1.00 \pm 0.00$ | $0.32 \pm 0.02$ | 1.00 |
| NeMo Hintersdorf et al. (2024) Adv. Images | 3 | $0.92 \pm 0.03$ | $0.95 \pm 0.03$ | $1.00 \pm 0.00$ | $0.33 \pm 0.01$ | 0.99 |
| | 4 | $0.92 \pm 0.03$ | $0.95 \pm 0.03$ | $1.00 \pm 0.00$ | $0.33 \pm 0.02$ | 0.99 |
| | 5 | $0.92 \pm 0.03$ | $0.95 \pm 0.03$ | $1.00 \pm 0.00$ | $0.32 \pm 0.02$ | 0.99 |
| | 1 | $0.35 \pm 0.19$ | $0.34 \pm 0.22$ | $0.48 \pm 0.14$ | $0.34 \pm 0.02$ | 0.18 |
| | 2 | $0.23 \pm 0.09$ | $0.23 \pm 0.11$ | $0.41 \pm 0.09$ | $0.35 \pm 0.02$ | 0.04 |
| NeMo Hintersdorf et al. (2024) Mitigated Images | 3 | $0.21 \pm 0.09$ | $0.20 \pm 0.09$ | $0.39 \pm 0.08$ | $0.35 \pm 0.02$ | 0.01 |
| | 4 | $0.20 \pm 0.07$ | $0.19 \pm 0.08$ | $0.39 \pm 0.08$ | $0.34 \pm 0.02$ | 0.00 |
| | 5 | $0.20 \pm 0.07$ | $0.19 \pm 0.08$ | $0.37 \pm 0.08$ | $0.34 \pm 0.02$ | 0.00 |

## H.7   COST OF SUCCESSFUL MEMORIZATION REMOVAL WITH WANDA

The results in Table 10 indicate that Wanda might be successful in removing memorized content from the model (low $SSCD_{Orig}$ at sparsity 10%) with limited harm to the alignment between the prompt and the generated images for benign input (high $A_{CLIP}$). However, FID scores appear to increase significantly with pruning (increase from 16.68 to 37.34 for VM samples). We investigate the harm that Wanda with 10% weights pruned causes to the model. The weights pruned by Wanda not only correspond to the memorized image, but also partially encode concepts present in the memorized content. For example, the memorized image in Fig. 1 (❶) consists also of a concept of woman. We show that in order to fully remove the image from the model, weights responsible for benign concepts, present in memorized data, are negatively affected.

We verify that idea by generating 100 images from a set of 10 prompts, which are paraphrases of the memorized prompts. Then, we compute CLIP similarity between the paraphrases and generated images, $A_{Concepts}$), to capture how the alignment changes with high pruning. The paraphrases are obtained by prompting LLama-3.1-8B-Instruct (Grattafiori et al., 2024), with the system prompt `"You are a paraphrasing engine. Preserve every tag and keyword."`, and the user prompt `"Write 10 alternative phrasings of:"'CAPTION"'`. Return only the paraphrasings, no other text. The format should be ['prompt1', 'prompt2', ...]". For example, for the prompt `"Living in the Light with Ann Graham Lotz"` we obtain `"Embracing Life in the Radiance of God with Ann Graham Lotz"`, `"A Life of Radiant Faith with Ann Graham Lotz"`, `"Living Life in the Illumination of God with Ann Graham Lotz"`, `"In the Presence of God's Radiant Light with Ann Graham Lotz"`, `"Faith in the Light of God with Ann Graham Lotz"`, `"Radiant Living with Ann Graham Lotz"`, `"In God's Illuminating Light with Ann Graham Lotz"`, `"Ann Graham Lotz on Living in God's Radiant Presence"`, `"Radiant Faith Living with Ann Graham Lotz"`, `"Living Life in God's Illuminating Light with Ann Graham Lotz"`. Additionally, we quantify image quality of the concepts by computing FID score (denoted $FID_{Concept}$) between 10,000 images generated from the prompts before and after pruning for VM and TM samples.

The results in Table 12 show that the concepts associated with the memorized images suffer after mitigation with Wanda. We observe a significant drop from $A_{CLIP}$ of around 0.37 to 0.33 for VM, which suggests that the generated images no longer follow the prompt. Moreover, the quality of the generated images degrades. $FID_{Concepts}$ above 80 for VM and above 90 for TM samples indicates significant harm to the model, corroborated by the last row of Table 10. Additionally, we provide qualitative results of damage to concepts in Appx. K.4.

Table 12: **Successful memorization removal with Wanda requires significant damage to the model.** While 10% sparsity ratio for Wanda mitigates memorization even under Dori, we observe a sharp drop in the generation quality ($FID_{Concept}$) and alignment between the prompt and generated images ($A_{Concept}$) for paraphrases of prompts used to remove memorization.

| Setting | Memorization | $SSCD_{Orig}$ | $A_{Concepts}$ | $FID_{Concepts}$ | ↓MR |
|---|---|---|---|---|---|
| No Mitigation | Verbatim | $0.90 \pm 0.04$ | $0.37 \pm 0.02$ | N/A | 1.00 |
| | Template | $0.17 \pm 0.09$ | $0.36 \pm 0.02$ | N/A | 1.00 |
| Wanda + 10% pruned + 🐟 | Verbatim | $0.40 \pm 0.13$ | $0.33 \pm 0.02$ | 80.70 | 0.01 |
| | Template | $0.22 \pm 0.08$ | $0.33 \pm 0.02$ | 92.46 | 0.01 |

## H.8 ADDITIONAL EXPERIMENTS ON STABLE DIFFUSION 2.0

We performed additional experiments on Stable Diffusion v2.0 using a subset of the prompts of Webster (2023), consistent with the setup in Ren et al. (2024). To stabilize the evaluation, we first collected all original images and filtered the generated samples using the SSCD score (threshold = 0.7), yielding 32 clearly memorized image–text pairs. We then conducted the following experiments and analyses: We both applied NeMo and Wanda to mitigate memorization via pruning. Since we didn't see memorization to be mitigated with the default parameters for NeMo, we even increased the pruning strength by setting $\theta_{min} = 0.25$ and the reduction of $\theta$ for each iteration in the initial neuron selection to 0.2. As a result more initial neurons are found which leads to more neurons being pruned using NeMo. We then evaluated the robustness of these pruned models against adversarial embeddings generated with Dori, using the same parameters as in the main paper. We then fine-tuned the Stable Diffusion 2.0 model for 5 epochs to apply our fine-tuning–based mitigation. Afterwards, we evaluated its effectiveness against both memorized text prompts and adversarial embeddings, again, using the same parameters as in the main paper.

Our results are consistent with the findings reported in the main paper: while both NeMo and Wanda initially appear effective at mitigating memorization, Dori remains capable of inducing data replication. In contrast, our fine-tuning strategy provides a substantially more robust mitigation, reflected in a significantly lower memorization rate (MR), while preserving overall image quality (as measured by FID/KID). As for the other experiments in the main paper we blocked the top 500 most frequently found neurons for NeMo for generating the COCO images for the FID/KID calculation.

Table 13: Additional experiments on Stable Diffusion 2.0. Embeddings are initialized with their corresponding *training prompt embeddings*. 🐟 denotes the application of adversarial embeddings.

| Setting | Adv. Steps | ↓ SSCD$_{Orig}$ | ↓ SSCD$_{Gen}$ | ↓ $D_{SSCD}$ | ↑ $A_{CLIP}$ | ↓MR | ↓FID | ↓KID |
|---|---|---|---|---|---|---|---|---|
| No Mitigation | - | $0.77 \pm 0.04$ | - | $0.98 \pm 0.00$ | $0.31 \pm 0.01$ | 1.0 | 15.21 | 0.0076 |
| NeMo | - | $0.23 \pm 0.07$ | $0.20 \pm 0.04$ | $0.71 \pm 0.07$ | $0.32 \pm 0.01$ | 0.06 | 15.35 | 0.0072 |
| Wanda | - | $0.09 \pm 0.03$ | $0.10 \pm 0.03$ | $0.45 \pm 0.10$ | $0.29 \pm 0.01$ | 0.00 | 16.79 | 0.0075 |
| Our Mitigation | - | $0.06 \pm 0.03$ | $0.02 \pm 0.03$ | $0.67 \pm 0.05$ | $0.30 \pm 0.01$ | 0.00 | 16.01 | 0.0071 |
| NeMo + 🐟 | 50 | $0.87 \pm 0.02$ | $0.74 \pm 0.12$ | $0.99 \pm 0.00$ | $0.31 \pm 0.01$ | 1.00 | 15.35 | 0.0072 |
| Wanda + 🐟 | 50 | $0.70 \pm 0.04$ | $0.19 \pm 0.05$ | $0.84 \pm 0.08$ | $0.31 \pm 0.01$ | 0.53 | 16.79 | 0.0075 |
| Our Mitigation + 🐟 | 50 | $0.14 \pm 0.06$ | $0.09 \pm 0.04$ | $0.70 \pm 0.04$ | $0.31 \pm 0.01$ | 0.06 | 16.01 | 0.0071 |

# I ADDITIONAL DETAILS AND EXPERIMENTS ON ADVERSARIAL FINE-TUNING

## I.1 ALGORITHMIC DESCRIPTION

Alg. 2 provides an algorithmic overview of our adversarial fine-tuning mitigation method. As described in the main paper, we begin by generating images from memorized prompts using a mitigation technique that preserves alignment with the prompt while avoiding replication of training data. Alternatively, these images can be generated using a separate DM that has not been trained on the corresponding samples. To preserve the model's general utility, a second set of non-memorized samples is incorporated during fine-tuning.

In the algorithmic description, surrogate and non-memorized samples are processed in separate batches to clearly illustrate the fine-tuning steps. However, in practice, embeddings and samples from both batches are concatenated to avoid redundant forward passes and to accelerate optimization. For each surrogate sample, a fixed adversarial embedding is used throughout an epoch. These embeddings are either initialized from the memorized prompt embedding or from random Gaussian noise. The adversarial fine-tuning loss, $\mathcal{L}_{adv}$, is computed exclusively on surrogate samples and their corresponding adversarial embeddings to reduce memorization.

In parallel, model utility is preserved through a utility loss, $\mathcal{L}_{non\text{-}mem}$, which is computed solely on the non-memorized samples. In our experiments, surrogate and non-memorized batches are of equal size by default; however, increasing the size of the non-memorized batch places greater emphasis on utility preservation.

---

**Algorithm 2 Fine-Tuning DM to Mitigate Memorization**

---

**Input:**

    DM $\epsilon_{\boldsymbol{\theta}}$

    Non-memorized images and corresponding prompts $\mathcal{D}_{\text{non-mem}}$

    Memorized images and corresponding prompts $\mathcal{D}_{\text{mem}}$

    Surrogate images $\mathcal{D}_{\text{surrogate}}$             ▷ Images generated with active mitigation

    Learning rate $\eta$, epochs $E$, steps per image $S$

**Output:**

    Fine-tuned model $\epsilon_{\boldsymbol{\theta}}$

  **for** epoch $\in \{1, \ldots, E\}$ **do**

    **for** $(\boldsymbol{x}_{\text{mem}}, \boldsymbol{p}_{\text{mem}}) \in \mathcal{D}_{\text{mem}}$ **do**

      **if** epoch mod 2 == 1 **then**

        $\boldsymbol{y}_{\text{adv}} \leftarrow \text{find\_adv\_embedding}(\boldsymbol{x}_{\text{mem}}, \boldsymbol{p}_{\text{mem}})$      ▷ Start from text embedding

      **else**

        $\boldsymbol{y}_{\text{adv}} \leftarrow \text{find\_adv\_embedding}(\boldsymbol{x}_{\text{mem}}, \text{random})$      ▷ Start from random embedding

      **end if**

      **for** $s \in \{1, \ldots, S\}$ **do**

        Sample surrogate image $\boldsymbol{x}_{\text{surr}} \in \mathcal{D}_{\text{surrogate}}$

        Sample non-memorized $(\boldsymbol{x}_{\text{non-mem}}, \boldsymbol{p}_{\text{non-mem}}) \in \mathcal{D}_{\text{non-mem}}$

        $\boldsymbol{\epsilon}_{\text{surr}} \sim \mathcal{N}(0, \boldsymbol{I})$             ▷ Update with surrogate/adversarial sample

        $t \sim \text{Uniform}(1, T)$

        $\tilde{\boldsymbol{x}}_{\text{surr}}^{(t)} \leftarrow \text{add\_noise}(\boldsymbol{x}_{\text{surr}}, \boldsymbol{\epsilon}_{\text{surr}}, t)$

        $\hat{\boldsymbol{\epsilon}}_{\text{surr}} \leftarrow \epsilon_{\boldsymbol{\theta}}(\tilde{\boldsymbol{x}}_{\text{surr}}^{(t)}, t, \boldsymbol{y}_{\text{adv}})$

        $\mathcal{L}_{\text{adv}} \leftarrow \|\boldsymbol{\epsilon}_{\text{surr}} - \hat{\boldsymbol{\epsilon}}_{\text{surr}}\|_2^2$

        $\boldsymbol{y}_{\text{non-mem}} \leftarrow \text{encode\_text}(\boldsymbol{p}_{\text{non-mem}})$

        $\boldsymbol{\epsilon}_{\text{non-mem}} \sim \mathcal{N}(0, \boldsymbol{I})$          ▷ Update with non-memorized sample

        $t \sim \text{Uniform}(1, T)$

        $\tilde{\boldsymbol{x}}_{\text{non-mem}}^{(t)} \leftarrow \text{add\_noise}(\boldsymbol{x}_{\text{non-mem}}, \boldsymbol{\epsilon}_{\text{non-mem}}, t)$

        $\hat{\boldsymbol{\epsilon}}_{\text{non-mem}} \leftarrow \epsilon_{\boldsymbol{\theta}}(\tilde{\boldsymbol{x}}_{\text{non-mem}}^{(t)}, t, \boldsymbol{y}_{\text{non-mem}})$

        $\mathcal{L}_{\text{non-mem}} \leftarrow \|\boldsymbol{\epsilon}_{\text{non-mem}} - \hat{\boldsymbol{\epsilon}}_{\text{non-mem}}\|_2^2$

        $\mathcal{L}_{\text{total}} \leftarrow \mathcal{L}_{\text{adv}} + \mathcal{L}_{\text{non-mem}}$          ▷ Aggregate both losses

        $\boldsymbol{\theta} \leftarrow \boldsymbol{\theta} - \eta \cdot \nabla_{\boldsymbol{\theta}} \mathcal{L}_{\text{total}}$

      **end for**

    **end for**

  **end for**

  **return** $\epsilon_{\boldsymbol{\theta}}$

---

## I.2 QUALITATIVE EXAMPLES OF SURROGATE IMAGES

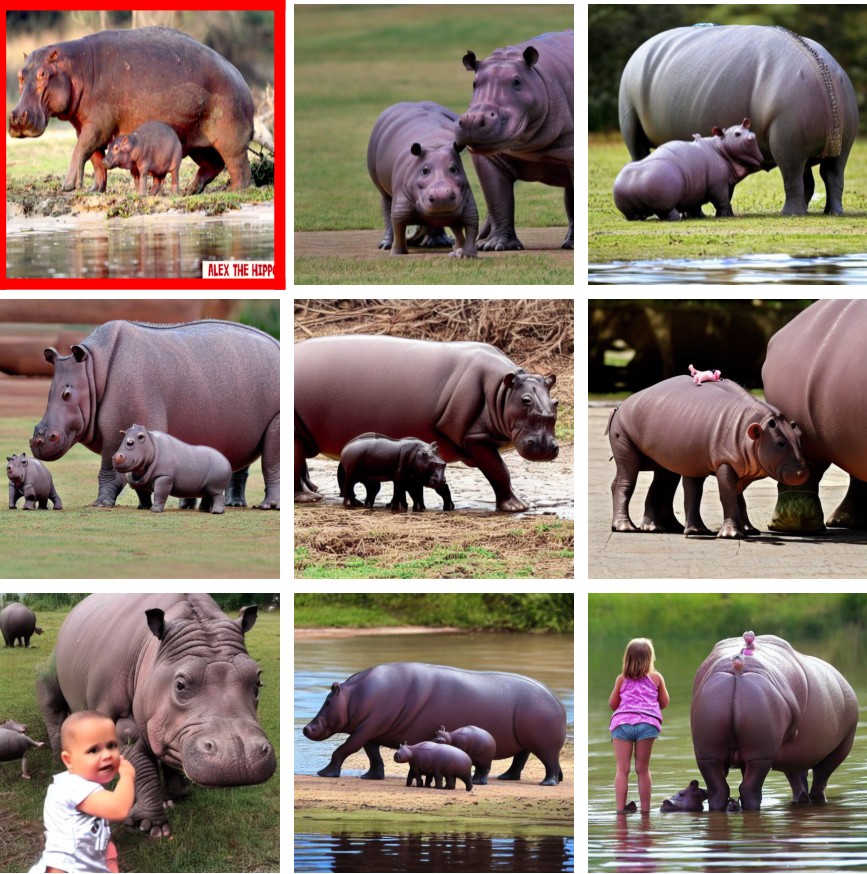

Figure 9: **Examples of surrogate images used for adversarial fine-tuning.** Multiple surrogate samples were generated for the memorized image (top left), preserving its semantic content without replicating the original. The images were created using NeMo to prevent exact replication.

## I.3 SENSITIVITY ANALYSIS

We extensively analyze the different components and hyperparameters used in our adversarial fine-tuning procedure. In all settings, we report intermediate training results after 1 to 50 training epochs.

Table 14 presents results for using 25, 50, and 100 adversarial embedding optimization steps during training to craft the adversarial embeddings used in the mitigation loss $\mathcal{L}_{Adv}$. The number of steps used during training is indicated in the *Setting* column. We further evaluate the mitigation effect using the unchanged embeddings of the memorized training prompts (denoted by 0 in the *Adv. Steps* column), as well as the same embeddings optimized with 50 steps. Results are reported only for VM prompts, which the model has been fine-tuned on.

Table 15 repeats the previous analysis, but explores the impact of starting the adversarial optimization from random embeddings instead of training prompt embeddings. We additionally compare the mitigation effect of adversarial embeddings crafted with 50 and 100 optimization steps, respectively.

Table 14: Comparison of performing adversarial fine-tuning with different numbers of adversarial embedding optimization steps during training. Embeddings are initialized with their corresponding *training prompt embeddings*. Results are reported for VM prompts only.

| Setting | Adv. Steps | Epochs | $\downarrow$ SSCD$_{\text{Orig}}$ | $\downarrow$ SSCD$_{\text{Gen}}$ | $\downarrow$ $D_{\text{SSCD}}$ | $\uparrow$ $A_{\text{CLIP}}$ | $\downarrow$MR | $\downarrow$ FID | $\downarrow$ KID |
|---|---|---|---|---|---|---|---|---|---|
| **No Mitigation** | 0 | – | $0.88 \pm 0.06$ | $1.0 \pm 0.0$ | $0.99 \pm 0.01$ | $0.32 \pm 0.02$ | 0.98 | 14.44 | 0.0060 |
| **Training with 25 adv. steps** | 0 | 1 | $0.16 \pm 0.05$ | $0.17 \pm 0.06$ | $0.35 \pm 0.09$ | $0.33 \pm 0.02$ | 0.00 | 15.26 | 0.0058 |
| | | 5 | $0.15 \pm 0.06$ | $0.16 \pm 0.07$ | $0.40 \pm 0.09$ | $0.33 \pm 0.02$ | 0.00 | 14.33 | 0.0052 |
| | | 10 | $0.14 \pm 0.07$ | $0.15 \pm 0.07$ | $0.43 \pm 0.09$ | $0.33 \pm 0.02$ | 0.00 | 14.86 | 0.0052 |
| | | 20 | $0.14 \pm 0.06$ | $0.16 \pm 0.06$ | $0.49 \pm 0.10$ | $0.32 \pm 0.02$ | 0.00 | 15.03 | 0.0047 |
| | | 30 | $0.12 \pm 0.06$ | $0.14 \pm 0.05$ | $0.52 \pm 0.10$ | $0.33 \pm 0.01$ | 0.00 | 15.46 | 0.0049 |
| | | 40 | $0.13 \pm 0.05$ | $0.14 \pm 0.05$ | $0.54 \pm 0.10$ | $0.32 \pm 0.02$ | 0.00 | 15.56 | 0.0047 |
| | | 50 | $0.12 \pm 0.07$ | $0.14 \pm 0.07$ | $0.54 \pm 0.11$ | $0.32 \pm 0.02$ | 0.00 | 15.52 | 0.0047 |
| **Training with 25 adv. steps** | 50 | 1 | $0.57 \pm 0.24$ | $0.59 \pm 0.23$ | $0.64 \pm 0.26$ | $0.32 \pm 0.01$ | 0.37 | 15.26 | 0.0058 |
| | | 5 | $0.38 \pm 0.18$ | $0.39 \pm 0.18$ | $0.51 \pm 0.12$ | $0.32 \pm 0.01$ | 0.12 | 14.33 | 0.0052 |
| | | 10 | $0.38 \pm 0.16$ | $0.39 \pm 0.19$ | $0.56 \pm 0.15$ | $0.32 \pm 0.02$ | 0.11 | 14.86 | 0.0052 |
| | | 20 | $0.39 \pm 0.18$ | $0.40 \pm 0.18$ | $0.56 \pm 0.13$ | $0.32 \pm 0.02$ | 0.24 | 15.03 | 0.0047 |
| | | 30 | $0.32 \pm 0.13$ | $0.36 \pm 0.16$ | $0.51 \pm 0.12$ | $0.32 \pm 0.02$ | 0.15 | 15.46 | 0.0049 |
| | | 40 | $0.27 \pm 0.11$ | $0.29 \pm 0.13$ | $0.53 \pm 0.10$ | $0.32 \pm 0.02$ | 0.07 | 15.56 | 0.0047 |
| | | 50 | $0.32 \pm 0.15$ | $0.34 \pm 0.14$ | $0.52 \pm 0.09$ | $0.33 \pm 0.02$ | 0.12 | 15.52 | 0.0047 |
| **Training with 50 adv. steps** | 0 | 1 | $0.14 \pm 0.05$ | $0.15 \pm 0.05$ | $0.33 \pm 0.08$ | $0.33 \pm 0.02$ | 0.00 | 15.66 | 0.0062 |
| | | 5 | $0.15 \pm 0.07$ | $0.15 \pm 0.07$ | $0.35 \pm 0.08$ | $0.33 \pm 0.01$ | 0.00 | 13.61 | 0.0047 |
| | | 10 | $0.13 \pm 0.05$ | $0.14 \pm 0.06$ | $0.37 \pm 0.09$ | $0.33 \pm 0.02$ | 0.00 | 15.16 | 0.0049 |
| | | 20 | $0.12 \pm 0.05$ | $0.13 \pm 0.05$ | $0.44 \pm 0.08$ | $0.32 \pm 0.02$ | 0.00 | 15.56 | 0.0051 |
| | | 30 | $0.14 \pm 0.05$ | $0.14 \pm 0.05$ | $0.45 \pm 0.09$ | $0.32 \pm 0.02$ | 0.00 | 15.47 | 0.0053 |
| | | 40 | $0.12 \pm 0.05$ | $0.14 \pm 0.06$ | $0.50 \pm 0.08$ | $0.32 \pm 0.01$ | 0.00 | 16.65 | 0.0055 |
| | | 50 | $0.12 \pm 0.05$ | $0.14 \pm 0.06$ | $0.52 \pm 0.09$ | $0.32 \pm 0.01$ | 0.00 | 16.02 | 0.0055 |
| **Training with 50 adv. steps** | 50 | 1 | $0.64 \pm 0.16$ | $0.69 \pm 0.16$ | $0.75 \pm 0.20$ | $0.32 \pm 0.01$ | 0.17 | 15.66 | 0.0062 |
| | | 5 | $0.36 \pm 0.14$ | $0.38 \pm 0.14$ | $0.54 \pm 0.10$ | $0.30 \pm 0.02$ | 0.02 | 13.61 | 0.0047 |
| | | 10 | $0.26 \pm 0.15$ | $0.27 \pm 0.16$ | $0.46 \pm 0.10$ | $0.30 \pm 0.02$ | 0.01 | 15.16 | 0.0049 |
| | | 20 | $0.29 \pm 0.13$ | $0.30 \pm 0.13$ | $0.46 \pm 0.07$ | $0.30 \pm 0.02$ | 0.00 | 15.56 | 0.0051 |
| | | 30 | $0.23 \pm 0.11$ | $0.28 \pm 0.12$ | $0.46 \pm 0.10$ | $0.30 \pm 0.02$ | 0.00 | 15.47 | 0.0053 |
| | | 40 | $0.22 \pm 0.12$ | $0.25 \pm 0.13$ | $0.48 \pm 0.09$ | $0.31 \pm 0.02$ | 0.00 | 16.65 | 0.0055 |
| | | 50 | $0.19 \pm 0.10$ | $0.21 \pm 0.11$ | $0.46 \pm 0.06$ | $0.31 \pm 0.02$ | 0.00 | 16.02 | 0.0055 |
| **Training with 100 adv. steps** | 0 | 1 | $0.13 \pm 0.04$ | $0.14 \pm 0.05$ | $0.29 \pm 0.07$ | $0.32 \pm 0.02$ | 0.00 | 15.32 | 0.0051 |
| | | 5 | $0.13 \pm 0.05$ | $0.14 \pm 0.05$ | $0.30 \pm 0.06$ | $0.32 \pm 0.02$ | 0.00 | 14.36 | 0.0049 |
| | | 10 | $0.13 \pm 0.05$ | $0.14 \pm 0.06$ | $0.34 \pm 0.09$ | $0.32 \pm 0.02$ | 0.00 | 15.56 | 0.0051 |
| | | 20 | $0.13 \pm 0.05$ | $0.13 \pm 0.05$ | $0.38 \pm 0.08$ | $0.32 \pm 0.02$ | 0.00 | 15.47 | 0.0053 |
| | | 30 | $0.12 \pm 0.05$ | $0.14 \pm 0.05$ | $0.42 \pm 0.07$ | $0.32 \pm 0.02$ | 0.00 | 15.34 | 0.0053 |
| | | 40 | $0.12 \pm 0.04$ | $0.13 \pm 0.04$ | $0.43 \pm 0.09$ | $0.32 \pm 0.02$ | 0.00 | 16.23 | 0.0052 |
| | | 50 | $0.12 \pm 0.04$ | $0.13 \pm 0.04$ | $0.46 \pm 0.09$ | $0.32 \pm 0.02$ | 0.00 | 17.52 | 0.0056 |
| **Training with 100 adv. steps** | 50 | 1 | $0.58 \pm 0.13$ | $0.61 \pm 0.11$ | $0.73 \pm 0.13$ | $0.31 \pm 0.01$ | 0.05 | 15.32 | 0.0051 |
| | | 5 | $0.31 \pm 0.10$ | $0.32 \pm 0.12$ | $0.44 \pm 0.07$ | $0.30 \pm 0.02$ | 0.01 | 14.36 | 0.0049 |
| | | 10 | $0.29 \pm 0.11$ | $0.30 \pm 0.12$ | $0.42 \pm 0.08$ | $0.29 \pm 0.02$ | 0.00 | 15.56 | 0.0051 |
| | | 20 | $0.25 \pm 0.11$ | $0.27 \pm 0.11$ | $0.40 \pm 0.07$ | $0.29 \pm 0.02$ | 0.00 | 15.47 | 0.0053 |
| | | 30 | $0.19 \pm 0.09$ | $0.20 \pm 0.11$ | $0.41 \pm 0.08$ | $0.30 \pm 0.02$ | 0.00 | 15.34 | 0.0053 |
| | | 40 | $0.22 \pm 0.10$ | $0.24 \pm 0.10$ | $0.41 \pm 0.06$ | $0.30 \pm 0.02$ | 0.00 | 16.23 | 0.0052 |
| | | 50 | $0.19 \pm 0.10$ | $0.20 \pm 0.09$ | $0.42 \pm 0.07$ | $0.30 \pm 0.02$ | 0.00 | 17.52 | 0.0056 |

Table 15: Comparison of performing adversarial fine-tuning with different numbers of adversarial embedding optimization steps during training. Embeddings are *initialized randomly*. Results are reported for VM prompts only.

| Setting | Adv. Steps | Epochs | ↓ SSCD$_{\text{Orig}}$ | ↓ SSCD$_{\text{Gen}}$ | ↓ $D_{\text{SSCD}}$ | ↑ $A_{\text{CLIP}}$ | ↓MR | ↓ FID | ↓ KID |
|---|---|---|---|---|---|---|---|---|---|
| **No Mitigation** | 0 | – | $0.88 \pm 0.06$ | $1.0 \pm 0.0$ | $0.99 \pm 0.01$ | $0.32 \pm 0.02$ | 1.00 | 14.44 | 0.0060 |
| **Training with 25 adv. steps** | 50 | 1 | $0.55 \pm 0.23$ | $0.58 \pm 0.22$ | $0.62 \pm 0.24$ | $0.32 \pm 0.02$ | 0.51 | 15.26 | 0.0058 |
| | | 5 | $0.49 \pm 0.20$ | $0.49 \pm 0.22$ | $0.53 \pm 0.16$ | $0.31 \pm 0.02$ | 0.27 | 14.33 | 0.0052 |
| | | 10 | $0.42 \pm 0.18$ | $0.45 \pm 0.19$ | $0.54 \pm 0.14$ | $0.31 \pm 0.02$ | 0.21 | 14.86 | 0.0052 |
| | | 20 | $0.45 \pm 0.18$ | $0.45 \pm 0.20$ | $0.57 \pm 0.16$ | $0.31 \pm 0.02$ | 0.24 | 15.03 | 0.0047 |
| | | 30 | $0.35 \pm 0.16$ | $0.39 \pm 0.17$ | $0.51 \pm 0.15$ | $0.31 \pm 0.02$ | 0.17 | 15.46 | 0.0049 |
| | | 40 | $0.28 \pm 0.14$ | $0.29 \pm 0.15$ | $0.52 \pm 0.12$ | $0.31 \pm 0.02$ | 0.10 | 15.56 | 0.0047 |
| | | 50 | $0.37 \pm 0.17$ | $0.39 \pm 0.17$ | $0.52 \pm 0.11$ | $0.32 \pm 0.02$ | 0.19 | 15.52 | 0.0047 |
| **Training with 25 adv. steps** | 100 | 1 | $0.89 \pm 0.05$ | $0.88 \pm 0.07$ | $0.99 \pm 0.01$ | $0.32 \pm 0.02$ | 0.88 | 15.26 | 0.0058 |
| | | 5 | $0.82 \pm 0.10$ | $0.81 \pm 0.12$ | $0.89 \pm 0.11$ | $0.32 \pm 0.02$ | 0.67 | 14.33 | 0.0052 |
| | | 10 | $0.74 \pm 0.15$ | $0.77 \pm 0.16$ | $0.85 \pm 0.14$ | $0.32 \pm 0.01$ | 0.55 | 14.86 | 0.0052 |
| | | 20 | $0.76 \pm 0.14$ | $0.79 \pm 0.14$ | $0.86 \pm 0.13$ | $0.32 \pm 0.02$ | 0.55 | 15.03 | 0.0047 |
| | | 30 | $0.76 \pm 0.12$ | $0.77 \pm 0.12$ | $0.92 \pm 0.08$ | $0.32 \pm 0.02$ | 0.60 | 15.46 | 0.0049 |
| | | 40 | $0.68 \pm 0.20$ | $0.70 \pm 0.23$ | $0.63 \pm 0.24$ | $0.32 \pm 0.02$ | 0.49 | 15.56 | 0.0047 |
| | | 50 | $0.76 \pm 0.12$ | $0.80 \pm 0.13$ | $0.83 \pm 0.16$ | $0.32 \pm 0.01$ | 0.60 | 15.52 | 0.0047 |
| **Training with 50 adv. steps** | 50 | 1 | $0.64 \pm 0.17$ | $0.68 \pm 0.16$ | $0.72 \pm 0.20$ | $0.32 \pm 0.01$ | 0.44 | 15.66 | 0.0062 |
| | | 5 | $0.37 \pm 0.12$ | $0.39 \pm 0.14$ | $0.54 \pm 0.11$ | $0.30 \pm 0.02$ | 0.04 | 13.61 | 0.0047 |
| | | 10 | $0.26 \pm 0.15$ | $0.27 \pm 0.16$ | $0.46 \pm 0.11$ | $0.30 \pm 0.02$ | 0.01 | 15.16 | 0.0049 |
| | | 20 | $0.29 \pm 0.12$ | $0.30 \pm 0.13$ | $0.46 \pm 0.07$ | $0.30 \pm 0.02$ | 0.02 | 15.56 | 0.0051 |
| | | 30 | $0.23 \pm 0.11$ | $0.28 \pm 0.13$ | $0.46 \pm 0.10$ | $0.30 \pm 0.02$ | 0.02 | 15.47 | 0.0053 |
| | | 40 | $0.22 \pm 0.12$ | $0.25 \pm 0.12$ | $0.48 \pm 0.09$ | $0.31 \pm 0.02$ | 0.02 | 16.65 | 0.0055 |
| | | 50 | $0.19 \pm 0.10$ | $0.21 \pm 0.11$ | $0.46 \pm 0.06$ | $0.31 \pm 0.02$ | 0.01 | 16.02 | 0.0055 |
| **Training with 50 adv. steps** | 100 | 1 | $0.83 \pm 0.08$ | $0.85 \pm 0.07$ | $0.94 \pm 0.06$ | $0.32 \pm 0.01$ | 0.73 | 15.66 | 0.0062 |
| | | 5 | $0.54 \pm 0.18$ | $0.56 \pm 0.17$ | $0.67 \pm 0.20$ | $0.31 \pm 0.01$ | 0.29 | 13.61 | 0.0047 |
| | | 10 | $0.47 \pm 0.20$ | $0.45 \pm 0.20$ | $0.52 \pm 0.17$ | $0.31 \pm 0.02$ | 0.21 | 15.16 | 0.0049 |
| | | 20 | $0.46 \pm 0.20$ | $0.47 \pm 0.20$ | $0.61 \pm 0.18$ | $0.31 \pm 0.02$ | 0.20 | 15.56 | 0.0051 |
| | | 30 | $0.37 \pm 0.19$ | $0.38 \pm 0.19$ | $0.53 \pm 0.15$ | $0.31 \pm 0.02$ | 0.16 | 15.47 | 0.0053 |
| | | 40 | $0.40 \pm 0.19$ | $0.40 \pm 0.21$ | $0.55 \pm 0.13$ | $0.32 \pm 0.02$ | 0.18 | 16.65 | 0.0055 |
| | | 50 | $0.34 \pm 0.17$ | $0.34 \pm 0.18$ | $0.52 \pm 0.10$ | $0.32 \pm 0.01$ | 0.13 | 16.02 | 0.0055 |
| **Training with 100 adv. steps** | 50 | 1 | $0.59 \pm 0.13$ | $0.61 \pm 0.10$ | $0.72 \pm 0.13$ | $0.32 \pm 0.01$ | 0.31 | 15.32 | 0.0051 |
| | | 5 | $0.32 \pm 0.11$ | $0.33 \pm 0.12$ | $0.44 \pm 0.07$ | $0.30 \pm 0.02$ | 0.02 | 14.36 | 0.0049 |
| | | 10 | $0.30 \pm 0.11$ | $0.30 \pm 0.12$ | $0.42 \pm 0.08$ | $0.29 \pm 0.02$ | 0.01 | 15.56 | 0.0051 |
| | | 20 | $0.25 \pm 0.11$ | $0.27 \pm 0.12$ | $0.40 \pm 0.07$ | $0.29 \pm 0.02$ | 0.00 | 15.47 | 0.0053 |
| | | 30 | $0.19 \pm 0.09$ | $0.20 \pm 0.11$ | $0.41 \pm 0.08$ | $0.30 \pm 0.02$ | 0.02 | 15.34 | 0.0053 |
| | | 40 | $0.22 \pm 0.10$ | $0.24 \pm 0.11$ | $0.41 \pm 0.06$ | $0.30 \pm 0.02$ | 0.00 | 16.23 | 0.0052 |
| | | 50 | $0.19 \pm 0.10$ | $0.20 \pm 0.09$ | $0.42 \pm 0.07$ | $0.30 \pm 0.02$ | 0.00 | 17.52 | 0.0056 |
| **Training with 100 adv. steps** | 100 | 1 | $0.77 \pm 0.11$ | $0.79 \pm 0.10$ | $0.86 \pm 0.13$ | $0.32 \pm 0.01$ | 0.64 | 15.32 | 0.0051 |
| | | 5 | $0.38 \pm 0.11$ | $0.37 \pm 0.13$ | $0.48 \pm 0.10$ | $0.30 \pm 0.02$ | 0.05 | 14.36 | 0.0049 |
| | | 10 | $0.34 \pm 0.13$ | $0.36 \pm 0.13$ | $0.48 \pm 0.07$ | $0.30 \pm 0.02$ | 0.04 | 15.56 | 0.0051 |
| | | 20 | $0.26 \pm 0.13$ | $0.29 \pm 0.14$ | $0.48 \pm 0.10$ | $0.29 \pm 0.02$ | 0.04 | 15.47 | 0.0053 |
| | | 30 | $0.22 \pm 0.11$ | $0.23 \pm 0.12$ | $0.43 \pm 0.08$ | $0.30 \pm 0.02$ | 0.04 | 15.34 | 0.0053 |
| | | 40 | $0.22 \pm 0.12$ | $0.24 \pm 0.13$ | $0.48 \pm 0.09$ | $0.30 \pm 0.02$ | 0.04 | 16.23 | 0.0052 |
| | | 50 | $0.19 \pm 0.10$ | $0.22 \pm 0.12$ | $0.48 \pm 0.09$ | $0.31 \pm 0.02$ | 0.04 | 17.52 | 0.0056 |

## I.4 ABLATION STUDY

We perform an ablation study to evaluate the impact of each individual component of the adversarial fine-tuning procedure. Specifically, we compare the default setting, where adversarial embeddings optimized for 50 steps are used over three consecutive fine-tuning steps, with alternative configurations. In one variant, only a single update is performed per embedding (*One Step*). In another, the model is fine-tuned exclusively on samples from the surrogate set (*No Utility Loss*). We also assess a setting where only non-memorized samples are used during fine-tuning (*No Mitigation Loss*). All configurations are evaluated across different training epochs, both when using the original training prompts without any adversarial optimization, and when using adversarial embeddings optimized for 50 steps.

The results in Table 16 show that fine-tuning the model for only a single step per adversarial embedding per epoch already achieves effective mitigation. This suggests that the fine-tuning process can be accelerated by reducing the number of updates per embedding. When the model is trained exclusively on surrogate samples—aiming to mitigate memorization without including any non-memorized samples to preserve utility—we observe strong mitigation, but at the cost of significantly degraded image quality, as reflected in the high FID and KID scores. Conversely, fine-tuning only on non-memorized samples while excluding surrogate samples helps maintain image quality but fails to provide sufficient mitigation against data replication.

Table 16: Comparison of adversarial fine-tuning performance with individual components ablated. Embeddings are initialized using training prompts. Results are reported for VM prompts only.

| Setting | Adv. Steps | Epochs | ↓ SSCD$_{\text{Orig}}$ | ↓ SSCD$_{\text{Gen}}$ | ↓ $D_{\text{SSCD}}$ | ↑ $A_{\text{CLIP}}$ | ↓MR | ↓ FID | ↓ KID |
|---|---|---|---|---|---|---|---|---|---|
| **No Mitigation** | 0 | – | 0.88 ± 0.06 | 1.0 ± 0.0 | 0.99 ± 0.01 | 0.32 ± 0.02 | 14.44 | 0.0060 | |
| **Default** | 0 | 1 | 0.14 ± 0.05 | 0.15 ± 0.05 | 0.33 ± 0.08 | 0.33 ± 0.02 | 0.00 | 15.66 | 0.0062 |
| | | 5 | 0.15 ± 0.07 | 0.15 ± 0.07 | 0.35 ± 0.08 | 0.33 ± 0.01 | 0.00 | 13.61 | 0.0047 |
| | | 10 | 0.13 ± 0.05 | 0.14 ± 0.06 | 0.37 ± 0.09 | 0.33 ± 0.02 | 0.00 | 15.16 | 0.0049 |
| | | 20 | 0.12 ± 0.05 | 0.13 ± 0.05 | 0.44 ± 0.08 | 0.32 ± 0.02 | 0.00 | 15.56 | 0.0051 |
| | | 30 | 0.14 ± 0.05 | 0.14 ± 0.05 | 0.45 ± 0.09 | 0.32 ± 0.02 | 0.00 | 15.47 | 0.0053 |
| | | 40 | 0.12 ± 0.05 | 0.14 ± 0.06 | 0.50 ± 0.08 | 0.32 ± 0.01 | 0.00 | 16.65 | 0.0055 |
| | | 50 | 0.12 ± 0.05 | 0.14 ± 0.06 | 0.52 ± 0.09 | 0.32 ± 0.01 | 0.00 | 16.02 | 0.0055 |
| **Default** | 50 | 1 | 0.64 ± 0.16 | 0.69 ± 0.16 | 0.75 ± 0.20 | 0.32 ± 0.01 | 0.17 | 15.66 | 0.0062 |
| | | 5 | 0.36 ± 0.14 | 0.38 ± 0.14 | 0.54 ± 0.10 | 0.30 ± 0.02 | 0.02 | 13.61 | 0.0047 |
| | | 10 | 0.26 ± 0.15 | 0.27 ± 0.16 | 0.46 ± 0.10 | 0.30 ± 0.02 | 0.01 | 15.16 | 0.0049 |
| | | 20 | 0.29 ± 0.13 | 0.30 ± 0.13 | 0.46 ± 0.07 | 0.30 ± 0.02 | 0.00 | 15.56 | 0.0051 |
| | | 30 | 0.23 ± 0.11 | 0.28 ± 0.12 | 0.46 ± 0.10 | 0.30 ± 0.02 | 0.00 | 15.47 | 0.0053 |
| | | 40 | 0.22 ± 0.12 | 0.25 ± 0.13 | 0.48 ± 0.09 | 0.31 ± 0.02 | 0.00 | 16.65 | 0.0055 |
| | | 50 | 0.19 ± 0.10 | 0.21 ± 0.11 | 0.46 ± 0.06 | 0.31 ± 0.02 | 0.00 | 16.02 | 0.0055 |
| **One Step** | 0 | 1 | 0.19 ± 0.05 | 0.18 ± 0.06 | 0.34 ± 0.08 | 0.33 ± 0.02 | 0.00 | 14.68 | 0.0056 |
| | | 5 | 0.13 ± 0.05 | 0.14 ± 0.05 | 0.33 ± 0.08 | 0.33 ± 0.02 | 0.00 | 15.07 | 0.0057 |
| | | 10 | 0.13 ± 0.05 | 0.14 ± 0.04 | 0.32 ± 0.07 | 0.33 ± 0.01 | 0.00 | 14.80 | 0.0056 |
| | | 20 | 0.14 ± 0.06 | 0.14 ± 0.05 | 0.37 ± 0.08 | 0.33 ± 0.02 | 0.00 | 14.92 | 0.0054 |
| | | 30 | 0.14 ± 0.05 | 0.15 ± 0.06 | 0.39 ± 0.08 | 0.33 ± 0.02 | 0.00 | 14.47 | 0.0045 |
| | | 40 | 0.12 ± 0.06 | 0.15 ± 0.06 | 0.44 ± 0.09 | 0.32 ± 0.02 | 0.00 | 15.75 | 0.0057 |
| | | 50 | 0.14 ± 0.06 | 0.14 ± 0.06 | 0.42 ± 0.07 | 0.33 ± 0.02 | 0.00 | 15.51 | 0.0051 |
| **One Step** | 50 | 1 | 0.69 ± 0.20 | 0.77 ± 0.16 | 0.73 ± 0.26 | 0.32 ± 0.02 | 0.49 | 14.68 | 0.0056 |
| | | 5 | 0.40 ± 0.14 | 0.43 ± 0.14 | 0.51 ± 0.13 | 0.32 ± 0.02 | 0.10 | 15.07 | 0.0057 |
| | | 10 | 0.31 ± 0.10 | 0.33 ± 0.11 | 0.47 ± 0.09 | 0.32 ± 0.02 | 0.03 | 14.80 | 0.0056 |
| | | 20 | 0.27 ± 0.10 | 0.29 ± 0.11 | 0.43 ± 0.08 | 0.32 ± 0.02 | 0.02 | 14.92 | 0.0054 |
| | | 30 | 0.26 ± 0.11 | 0.28 ± 0.12 | 0.47 ± 0.08 | 0.32 ± 0.02 | 0.01 | 14.47 | 0.0045 |
| | | 40 | 0.22 ± 0.09 | 0.23 ± 0.10 | 0.48 ± 0.09 | 0.32 ± 0.02 | 0.02 | 15.75 | 0.0057 |
| | | 50 | 0.22 ± 0.10 | 0.25 ± 0.12 | 0.49 ± 0.07 | 0.32 ± 0.02 | 0.00 | 15.51 | 0.0051 |
| **No Utility Loss** | 0 | 1 | 0.11 ± 0.04 | 0.13 ± 0.04 | 0.25 ± 0.08 | 0.31 ± 0.02 | 0.00 | 18.74 | 0.0057 |
| | | 5 | 0.12 ± 0.04 | 0.13 ± 0.04 | 0.28 ± 0.08 | 0.30 ± 0.01 | 0.00 | 30.96 | 0.0123 |
| | | 10 | 0.12 ± 0.04 | 0.13 ± 0.04 | 0.28 ± 0.06 | 0.30 ± 0.02 | 0.00 | 35.05 | 0.0156 |
| | | 20 | 0.11 ± 0.04 | 0.13 ± 0.04 | 0.38 ± 0.10 | 0.31 ± 0.02 | 0.00 | 63.03 | 0.0212 |
| | | 30 | 0.12 ± 0.05 | 0.13 ± 0.05 | 0.41 ± 0.11 | 0.30 ± 0.02 | 0.00 | 66.82 | 0.0237 |
| | | 40 | 0.10 ± 0.04 | 0.13 ± 0.04 | 0.48 ± 0.13 | 0.31 ± 0.01 | 0.00 | 82.38 | 0.0245 |
| | | 50 | 0.10 ± 0.05 | 0.12 ± 0.05 | 0.48 ± 0.10 | 0.31 ± 0.02 | 0.00 | 81.72 | 0.0262 |
| **No Utility Loss** | 50 | 1 | 0.42 ± 0.15 | 0.44 ± 0.16 | 0.57 ± 0.15 | 0.32 ± 0.02 | 0.09 | 18.74 | 0.0057 |
| | | 5 | 0.28 ± 0.11 | 0.30 ± 0.12 | 0.54 ± 0.08 | 0.31 ± 0.02 | 0.00 | 30.96 | 0.0123 |
| | | 10 | 0.20 ± 0.09 | 0.21 ± 0.10 | 0.52 ± 0.08 | 0.30 ± 0.02 | 0.00 | 35.05 | 0.0156 |
| | | 20 | 0.19 ± 0.08 | 0.23 ± 0.10 | 0.51 ± 0.08 | 0.30 ± 0.02 | 0.00 | 63.03 | 0.0212 |
| | | 30 | 0.20 ± 0.07 | 0.21 ± 0.08 | 0.48 ± 0.07 | 0.30 ± 0.02 | 0.00 | 66.82 | 0.0237 |
| | | 40 | 0.16 ± 0.05 | 0.19 ± 0.06 | 0.50 ± 0.08 | 0.20 ± 0.02 | 0.00 | 82.38 | 0.0245 |
| | | 50 | 0.16 ± 0.06 | 0.18 ± 0.06 | 0.47 ± 0.07 | 0.28 ± 0.03 | 0.00 | 81.72 | 0.0262 |
| **No Mitigation Loss** | 0 | 1 | 0.89 ± 0.06 | 0.98 ± 0.01 | 0.99 ± 0.01 | 0.33 ± 0.02 | 0.91 | 14.47 | 0.0056 |
| | | 5 | 0.86 ± 0.06 | 0.96 ± 0.01 | 0.99 ± 0.01 | 0.33 ± 0.02 | 0.86 | 14.45 | 0.0050 |
| | | 10 | 0.48 ± 0.10 | 0.57 ± 0.10 | 0.58 ± 0.13 | 0.33 ± 0.02 | 0.06 | 15.13 | 0.0052 |
| | | 20 | 0.62 ± 0.15 | 0.75 ± 0.13 | 0.64 ± 0.28 | 0.34 ± 0.02 | 0.30 | 14.40 | 0.0051 |
| | | 30 | 0.73 ± 0.11 | 0.87 ± 0.06 | 0.90 ± 0.10 | 0.33 ± 0.02 | 0.55 | 15.02 | 0.0046 |
| | | 40 | 0.63 ± 0.18 | 0.71 ± 0.19 | 0.71 ± 0.23 | 0.34 ± 0.02 | 0.34 | 16.04 | 0.0051 |
| | | 50 | 0.54 ± 0.17 | 0.65 ± 0.18 | 0.51 ± 0.11 | 0.33 ± 0.02 | 0.21 | 15.70 | 0.0049 |
| **No Mitigation Loss** | 50 | 1 | 0.91 ± 0.03 | 0.96 ± 0.02 | 1.00 ± 0.00 | 0.33 ± 0.02 | 1.00 | 14.47 | 0.0056 |
| | | 5 | 0.90 ± 0.03 | 0.96 ± 0.02 | 1.00 ± 0.00 | 0.33 ± 0.02 | 0.98 | 14.45 | 0.0050 |
| | | 10 | 0.88 ± 0.03 | 0.92 ± 0.04 | 1.00 ± 0.00 | 0.32 ± 0.02 | 0.96 | 15.13 | 0.0052 |
| | | 20 | 0.88 ± 0.04 | 0.94 ± 0.03 | 1.00 ± 0.00 | 0.32 ± 0.02 | 0.98 | 14.40 | 0.0051 |
| | | 30 | 0.88 ± 0.04 | 0.94 ± 0.03 | 1.00 ± 0.00 | 0.32 ± 0.01 | 0.97 | 15.02 | 0.0046 |
| | | 40 | 0.85 ± 0.04 | 0.92 ± 0.03 | 1.00 ± 0.00 | 0.32 ± 0.01 | 0.91 | 16.04 | 0.0051 |
| | | 50 | 0.86 ± 0.05 | 0.92 ± 0.04 | 1.00 ± 0.00 | 0.32 ± 0.02 | 0.91 | 15.70 | 0.0049 |

### I.5 COST DISCUSSION

Our mitigation method can seem costly at first, given it requires generating a set of surrogate images from the model, and performing a full fine-tuning of the whole DM. In total, the runtime of our method on a single A100 GPU can be approximated with the following equation

$$\text{Time} = (\text{N} \times 20) + (\text{E} \times \text{N} \times 10), \tag{5}$$

where Time is in seconds, N is the number of memorized images to remove, and E is the number of epochs we run our method. For N=112, E=5, we get about 2.5h of runtime, which we consider reasonable, given the effectiveness of our method.

An alternative approach would be to apply LoRA (Hu et al., 2021) adaptation on the DM, to lower the compute cost. We use LoRA with rank 128 on (1) all fully-connected layers, and (2) both fully-connected and convolutional layers. Unfortunately, in neither of those cases the memorization is fully mitigated, even after 50 epochs of our mitigation method. We are still able to craft adversarial embeddings to replicate memorized data. This result is consistent with our earlier findings: memorization is not confined to individual weights or layers, but is distributed across the entire model and requires updates to additional components.

### I.6 OUR MITIGATION IS ROBUST TO OTHER ADVERSARIAL OPTIMIZATION TECHNIQUES

While our adversarial fine-tuning is robust against Dori, we acknowledge that it might not be robust to other methods of obtaining adversarial embeddings. Since Dori is the first such method for triggering memorization, we apply UnlearnDiffAtk (Zhang et al., 2024c) against our fine-tuned model. See Appx. G.1 for details on the implementation. The prompt optimization algorithm is GCG (Zou et al., 2023), arguably the strongest adversarial optimization method available at the time of writing the paper. We find that our model is robust against UnlearnDiffAtk, *i.e.,* the adversarial prompts do not cause replication of memorized content.

### I.7 OUR MITIGATION DOES NOT CAUSE UNINTENDED MEMORIZATION OF OTHER SAMPLES

Model updates or interventions can, theoretically, always lead to unexpected side-effects. However, memorization in DMs mostly originates from training data duplications, and our fine-tuning method does not add any further duplications (if the dataset is sufficiently large).

To empirically support our claim that our adversarial fine-tuning does not add novel memorization, we generate 10 images for 1,000 LAION samples used for fine-tuning. We then compute the SSCD score between the generated images and the original images used for the fine-tuning. Since we do not find any image with $\text{SSCD}_{\text{Orig}} > 0.7$ we argue that none of the images used for fine-tuning is memorized. The result of our analysis is that the SSCD score between the generated images and the original images used for the fine-tuning is only $0.16 \pm 0.04$, validating that indeed the images used for fine-tuning are not memorized.

## J  ADDITIONAL DETAILS AND EXPERIMENTS ON LOCALITY

In this section, we summarize results from broader experiments regarding locality. In Appx. J.1 we provide t-SNE plots for adversarial embeddings optimized starting from text embeddings of non-memorized prompts, as well as pairwise L2 distances between embeddings, and in Appx. J.2 we define two scores used to assess locality in the model: activation discrepancy and weight agreement.

### J.1  LOCALITY IN THE TEXT EMBEDDING SPACE

We extend the analysis of adversarial embeddings in the text embedding space. Contrary to Section 4.1, we now initialize optimization from a randomly sampled non-memorized prompts—$\boldsymbol{y}_{nonmem}$—from LAION dataset, and perform 50 steps of optimization. In line with the previous results, all embeddings $\boldsymbol{y}_{adv}$ trigger successful replication of the memorized content, and are spread out in the text embedding space.

Additionally, we compute pairwise L2 distance in the embedding space for (1) random initialization ($\mathcal{N}(\boldsymbol{0}, \boldsymbol{I})$), (2) adversarial embeddings optimized from $\mathcal{N}(\boldsymbol{0}, \boldsymbol{I})$, (3) embeddings of non-memorized prompts ($\boldsymbol{y}_{nonmem}$), (4) adversarial embeddings optimized from $\boldsymbol{y}_{nonmem}$. To our surprise, the adversarial embeddings that trigger generation of *the same* memorized samples appear to be more spread out than randomly initialized embeddings, and are also more spread out than embeddings of non-memorized prompts, as it is visible in Fig. 11 (left).

Interestingly, initialization of optimization from $\mathcal{N}(\boldsymbol{0}, \boldsymbol{I})$ is more beneficial to finding Dori. We observe that the embeddings have to be changed less than when we initialize them from prompts, which is expressed by lower L2 distance between initializations and the final embeddings $\boldsymbol{y}_{adv}$ in Fig. 11 (right).

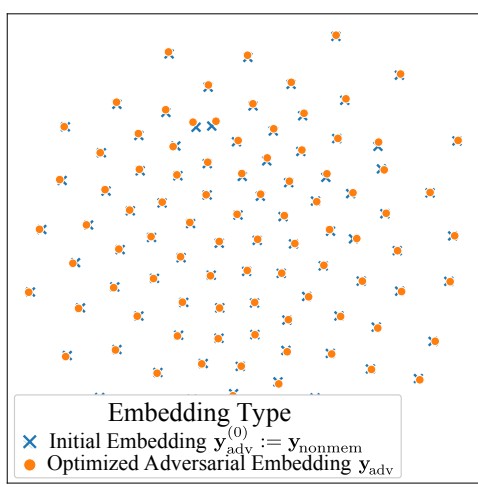

Embedding Type
× Initial Embedding $\mathbf{y}_{adv}^{(0)} := \mathbf{y}_{nonmem}$
● Optimized Adversarial Embedding $\mathbf{y}_{adv}$

Figure 10: **T-SNE visualization of** 100 **non-memorized text embeddings** $\boldsymbol{y}_{nonmem}$ **(blue) and adversarially crafted embeddings (orange)** $\boldsymbol{y}_{adv}$**, generated by perturbing the blue ones.** We observe identical behavior for non-random initialization as in Fig. 3 —adversarial embeddings are uniformly distributed in the text embedding space.

### J.2  LOCALITY IN THE MODEL'S WEIGHTS

Discrepancy between activations in a given layer is defined as

$$\text{Discrepancy}(\boldsymbol{y}_i, \boldsymbol{y}_j) = ||\text{Activations}(\boldsymbol{y}_i) - \text{Activations}(\boldsymbol{y}_j)||_2^2,$$

where Activations($\boldsymbol{y}$) outputs a vector of activations of a given layer further used to identify memorization weights by Wanda or NeMo. For NeMo, we obtain activations from passing the text embedding $\boldsymbol{y}$ through the value layer in cross-attention blocks, and Wanda utilizes activation of the feed-forward layer after the attention operator in cross-attention blocks. To assess mean pairwise discrepancy of in set $\boldsymbol{Y} = \{\boldsymbol{y}_i | i = 1, \dots, N\}$ of size $N$ we use

$$\text{MeanDiscrepancy}(\boldsymbol{Y}) = \frac{1}{(N-1)^2} \sum_{i=1}^{N} \sum_{j=1}^{N} \mathbb{1}(i \neq j) \text{Discrepancy}(\boldsymbol{y}_i, \boldsymbol{y}_j). \quad (6)$$

We define agreement between memorization weights identified in a single layer for two different input embeddings as

$$\text{Agreement}(\boldsymbol{y}_i, \boldsymbol{y}_j) = \frac{\# \left( \text{Weights}(\boldsymbol{y}_i) \cap \text{Weights}(\boldsymbol{y}_j) \right)}{\# \left( \text{Weights}(\boldsymbol{y}_i) \cup \text{Weights}(\boldsymbol{y}_j) \right)},$$

where $\boldsymbol{y}_i$ and $\boldsymbol{y}_j$ are two embeddings that trigger replication of some memorized image(s), and Weights($\boldsymbol{y}$) returns a set of weights identified by a pruning-based mitigation method, $\#\boldsymbol{Y}$ denotes

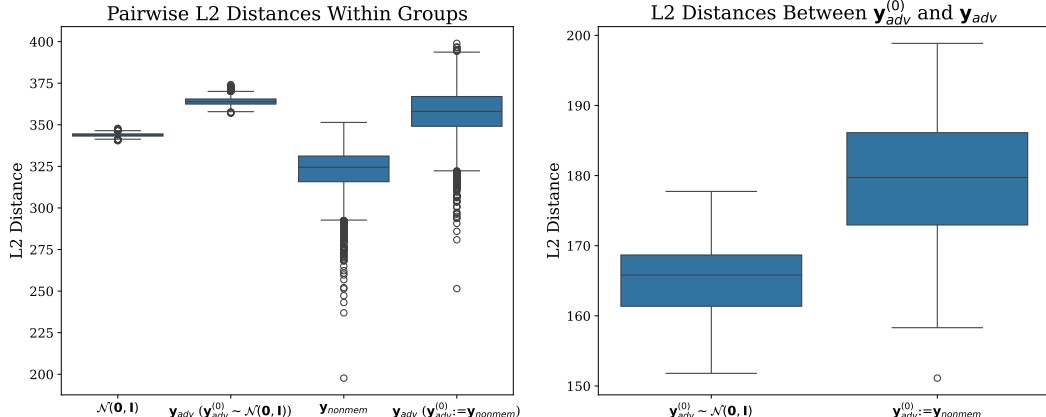

Figure 11: **L2 distances of input embeddings.** Left: we compute pairwise L2 distances in text embedding space within the set of 100 random embeddings ($\mathcal{N}(\mathbf{0}, \boldsymbol{I})$), set of adversarial embeddings optimized from random embeddings (second box from the left), 100 randomly selected non-memorized prompts ($\boldsymbol{y}_{nonmem}$) and adversarial embeddings optimized from non-memorized embeddings (fourth box from the left). We observe that after optimization, the adversarial embeddings are *more* spread out in the text embedding space than their initial points (be it $\mathcal{N}(\mathbf{0}, \boldsymbol{I})$ or randomly selected $\boldsymbol{y}_{nonmem}$). Right: We compute the L2 distance between the initialization and the final adversarial embeddings. We note that when initializing the optimization with $\boldsymbol{y}_{nonmem}$ we have to travel farther in the text embedding space to obtain an adversarial embedding $\boldsymbol{y}_{adv}$ that successfully triggers replication of the memorized image $\boldsymbol{x}_{mem}$.

the size of the set. Analogically to MeanDiscrepancy, we define the mean pairwise weight agreement for a set of embeddings $\boldsymbol{Y}$ as

$$\text{MeanAgreement}(\boldsymbol{Y}) = \frac{1}{(N-1)^2} \sum_{i=1}^{N} \sum_{j=1}^{N} \mathbb{1}(i \neq j) \text{Agreement}(\boldsymbol{y}_i, \boldsymbol{y}_j). \qquad (7)$$

### J.3 LOCALITY IS NOT PRESENT IN ANY LAYERS

Our findings from Section 4.3 suggest that locality of memorized data in the specific layers of the model is an illusion. In this section, we extend the analysis to other layers, specifically Q, K, V, and MLP layers in self- and cross-attention modules, as well as convolutional layers in ResNet modules of the U-Net. To evaluate locality, we default to activations, specifically the MeanDiscrepancy, defined in Appx. J.2, and omit weight agreement, since neither Wanda nor NeMo focuses their mitigation efforts on these additional layers.

The results in Fig. 12 further undermine the notion of locality. In all layer types, we observe high discrepancy for various adversarial embeddings $\boldsymbol{y}_{adv}$ associated with the same image. This shows that trying to localize neurons for pruning memorization mitigation based on activations can not work, as vastly different activation patterns lead to the same memorized image.

Interestingly, the discrepancy for $\boldsymbol{y}_{mem}$ associated with different images is significantly higher for layers that interact with the image features directly (like Q matrices in self-attention), and the effect strengthens deeper in the model. We argue that this is an expected behavior—outputs (and thus activations) of the model should diverge for different images.

### J.4 NEMO AND WANDA IDENTIFY MEMORIZATION WEIGHTS IN DIFFERENT LAYERS

We examine the behavior of pruning-based methods through the lens of their weights selection. To this end, we compute these weights for all VM samples, separately for each memorized image. In Fig. 13 we show that NeMo tend to identify memorization weights only in four out of seven layers. This result explains high weight agreement in layers two, six, and seven in Fig. 5, since when no weights are identified in a layer, we set agreement to 1. Results for Wanda contrast with the results

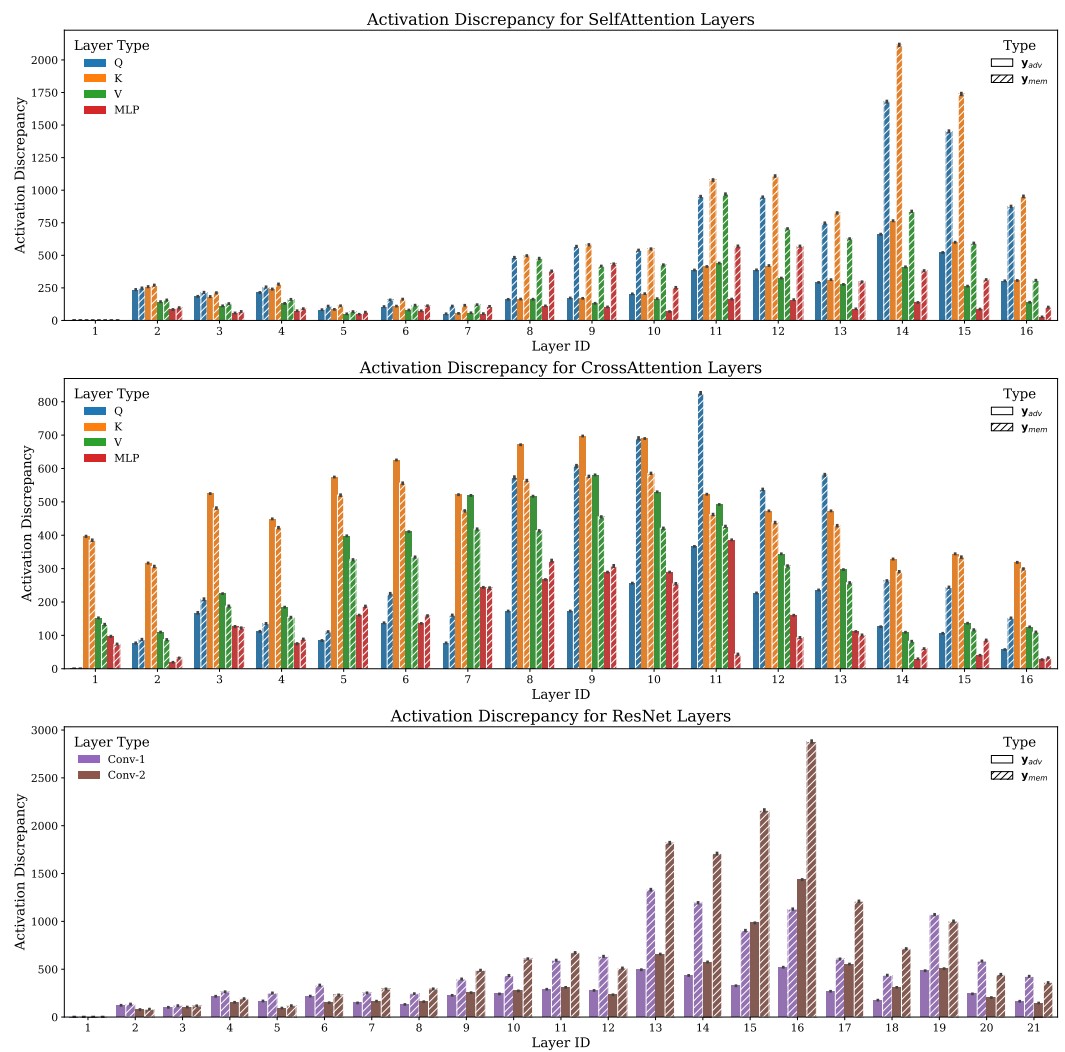

Figure 12: **Locality assumption fails in all layers of the U-Net.** We observe high activation discrepancy for adversarial embeddings triggering the same image, regardless of the layer type and depth.

for NeMo, as it finds more traces of memorized content in deeper layers of the model (five, six, and seven). Importantly, in these layers also the agreement drops significantly, as can be seen in Fig. 5.

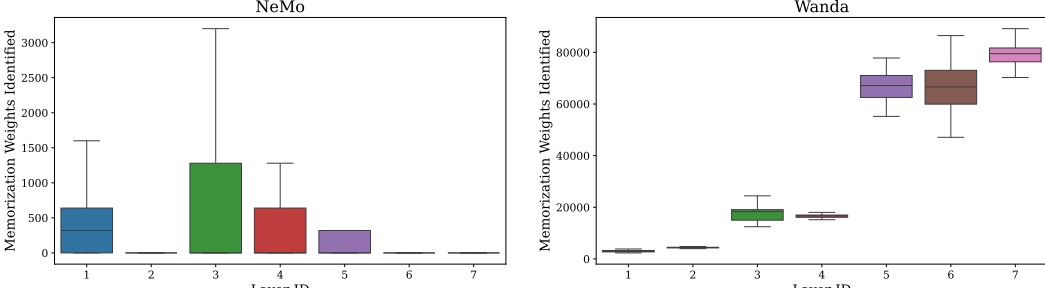

Figure 13: **Number of memorization weights per layer.** We observe that for NeMo, no weights are identified to prune in layers two, six, and seven (**left**). Conversely, Wanda identifies significantly more memorization weights in deeper layers. Interestingly, the drop in weight agreement for Wanda (Fig. 5) happens also in the deeper layers of the model.

## K QUALITATIVE RESULTS

### K.1 QUALITATIVE RESULTS FOR WANDA PRUNING

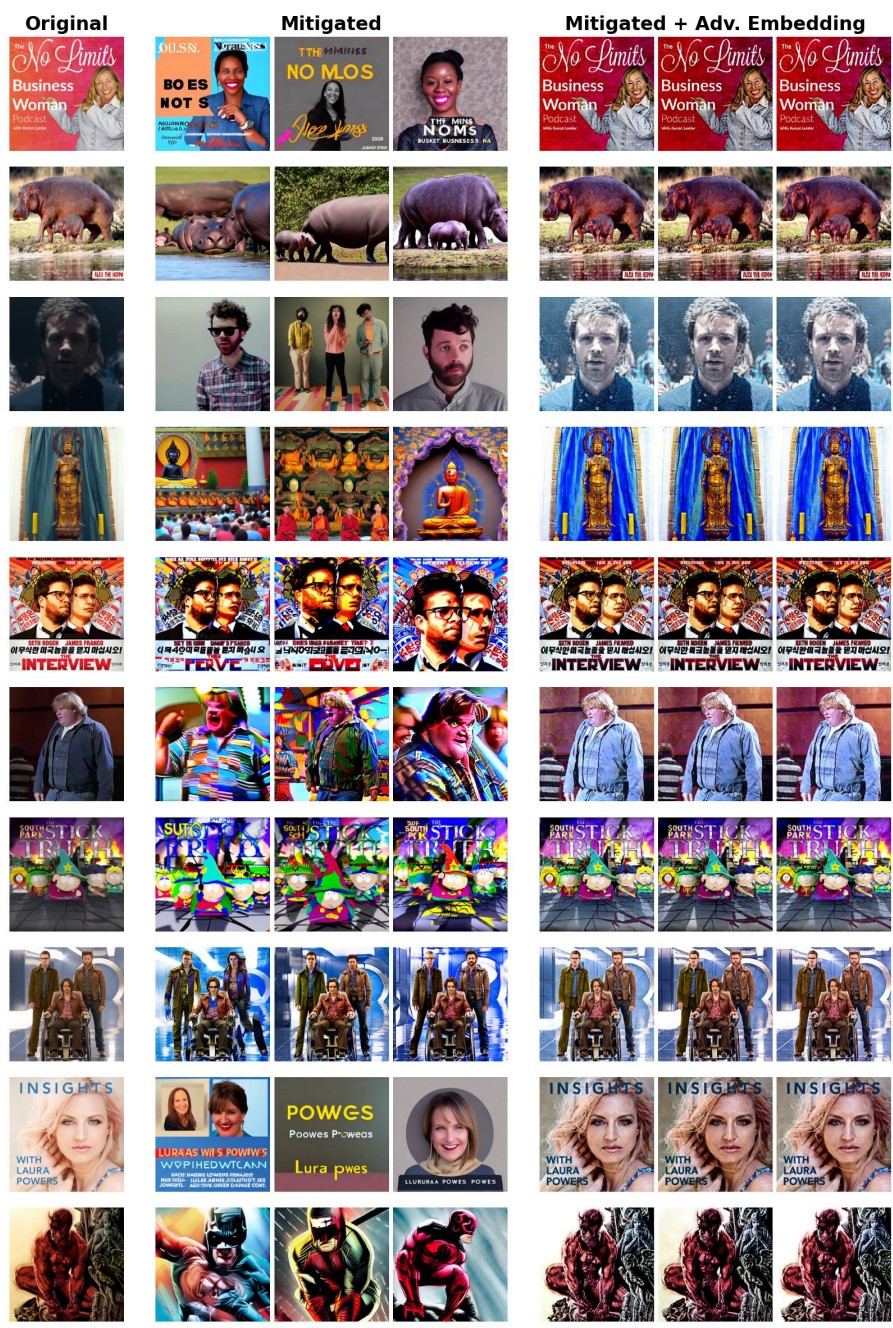

Figure 14: **Qualitative results after applying Wanda.** The first column shows the original training images. The next three columns show generations after applying the mitigation technique. The final three columns show generations from adversarial embeddings, also after applying the mitigation technique. The adversarial embeddings were initialized with memorized prompt embeddings and optimized for 50 steps.

## K.2    QUALITATIVE RESULTS FOR NEMO PRUNING

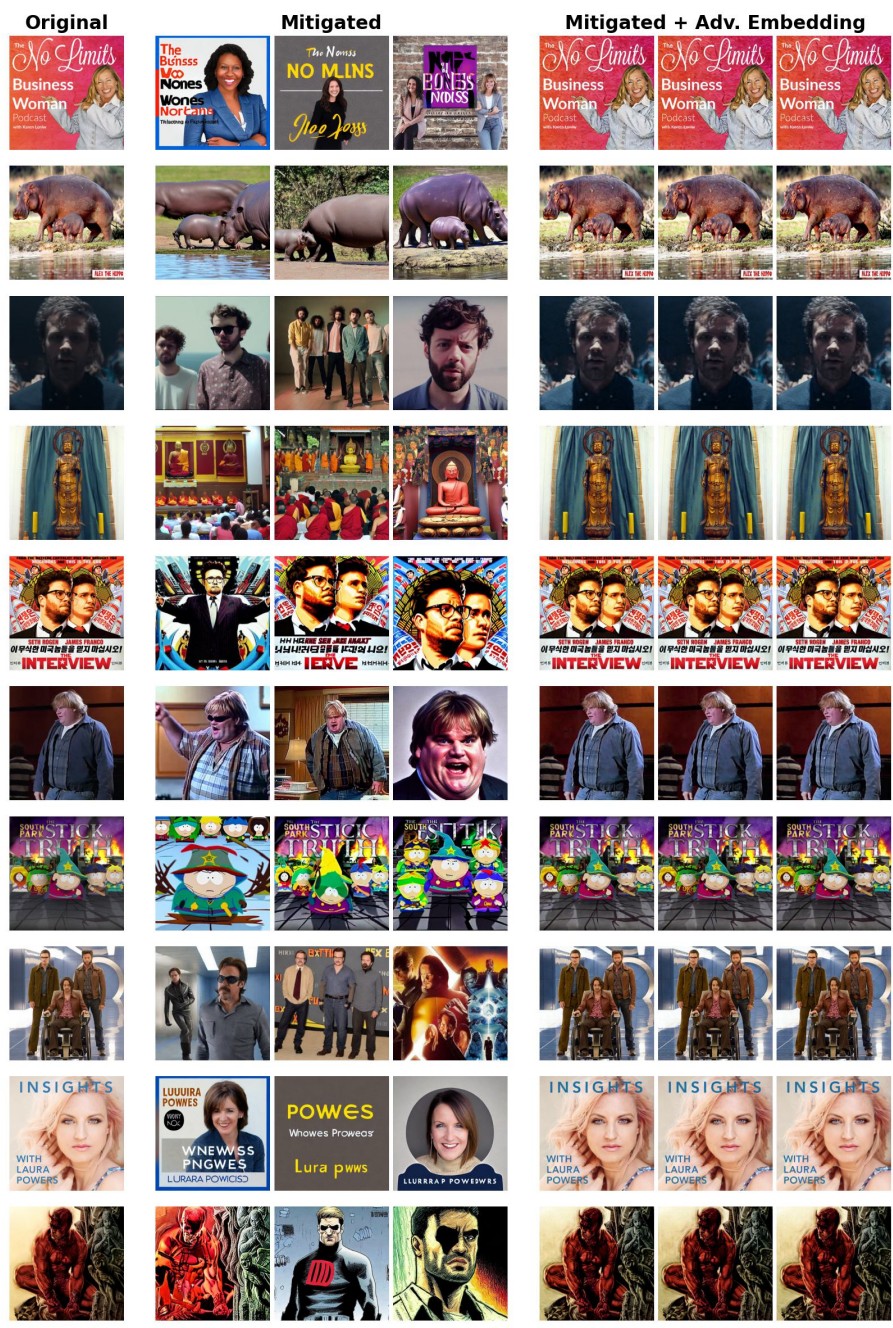

Figure 15: **Qualitative results after applying NeMo.** The first column shows the original training images. The next three columns show generations after applying the mitigation technique. The final three columns show generations from adversarial embeddings, also after applying the mitigation technique. The adversarial embeddings were initialized with memorized prompt embeddings and optimized for 50 steps.

## K.3 ADVERSARIAL FINE-TUNING

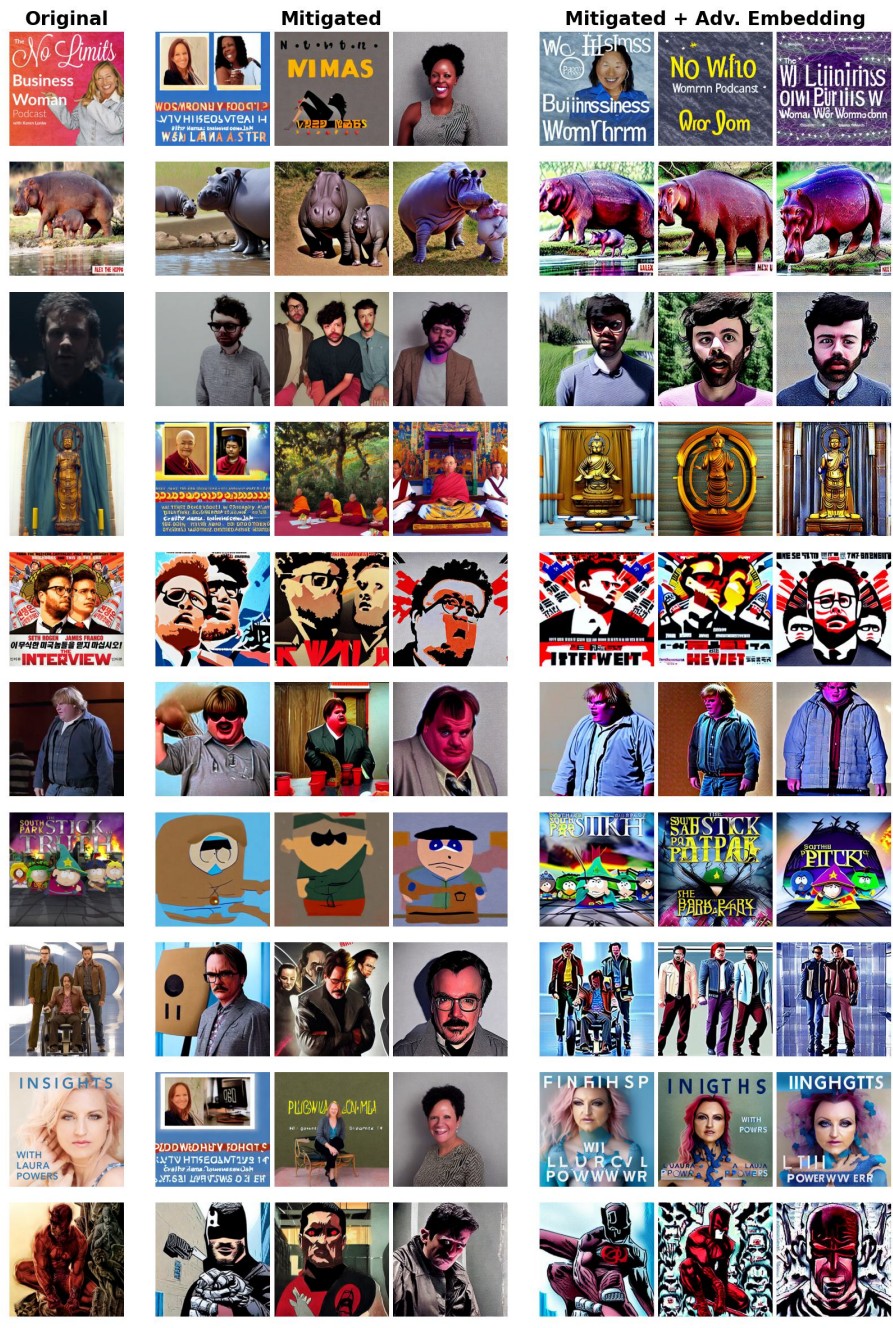

Figure 16: **Qualitative results for memorized content after applying our adversarial fine-tuning.** The first column shows the original training images. The next three columns show generations after fine-tuning the model for five epochs using the default parameters reported in the main paper. The final three columns show generations from adversarial embeddings, also after applying the mitigation technique. The adversarial embeddings were initialized with memorized prompt embeddings and optimized for 50 steps.

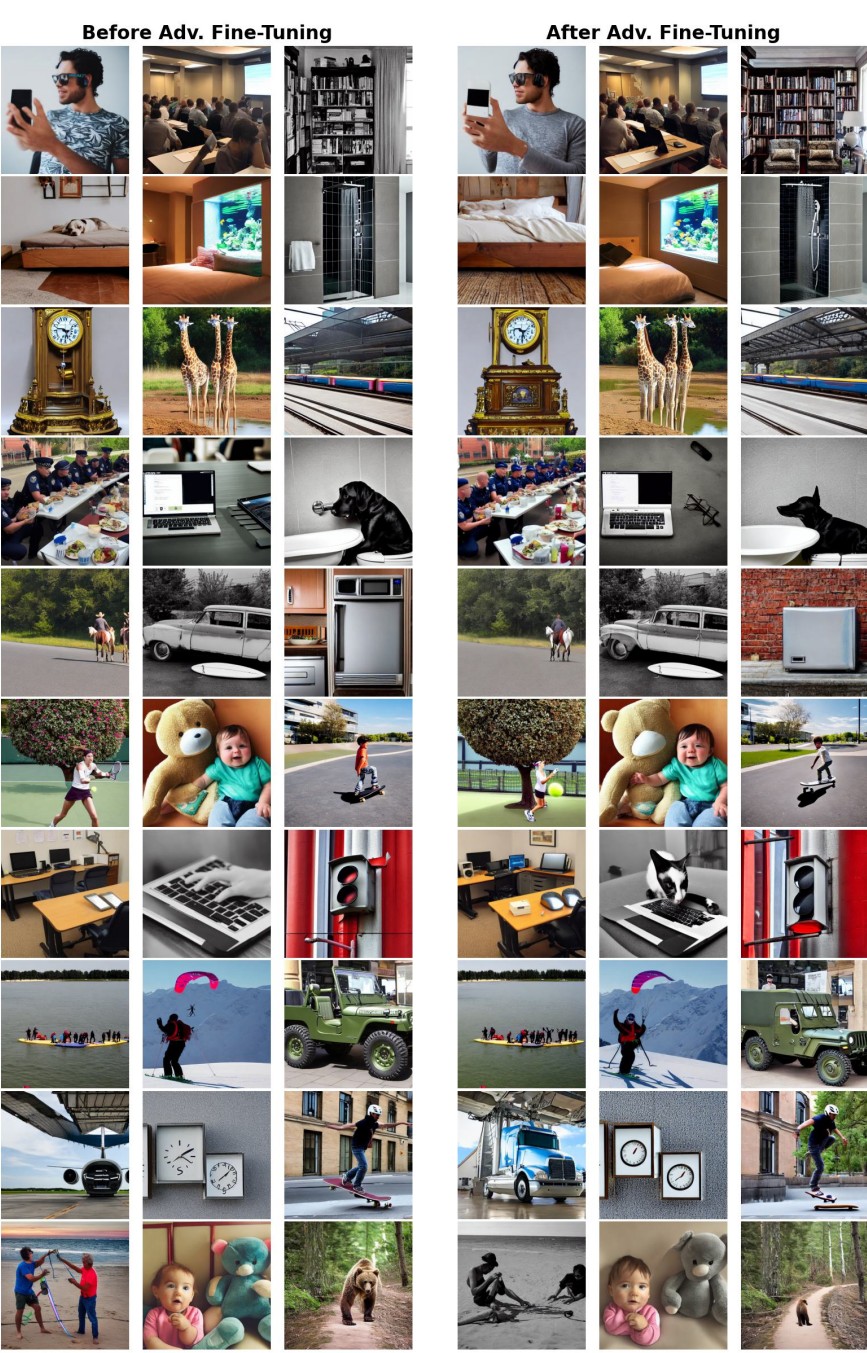

Figure 17: **Qualitative results on COCO after applying our adversarial fine-tuning.** The first three columns show images generated for 30 COCO prompts using the default Stable Diffusion v1.4 model. The last three columns show generations after fine-tuning the model for five epochs using our adversarial fine-tuning mitigation. The adversarial embeddings were initialized with memorized prompt embeddings and optimized for 50 steps.

### K.4 WANDA WITH 10% SPARSITY

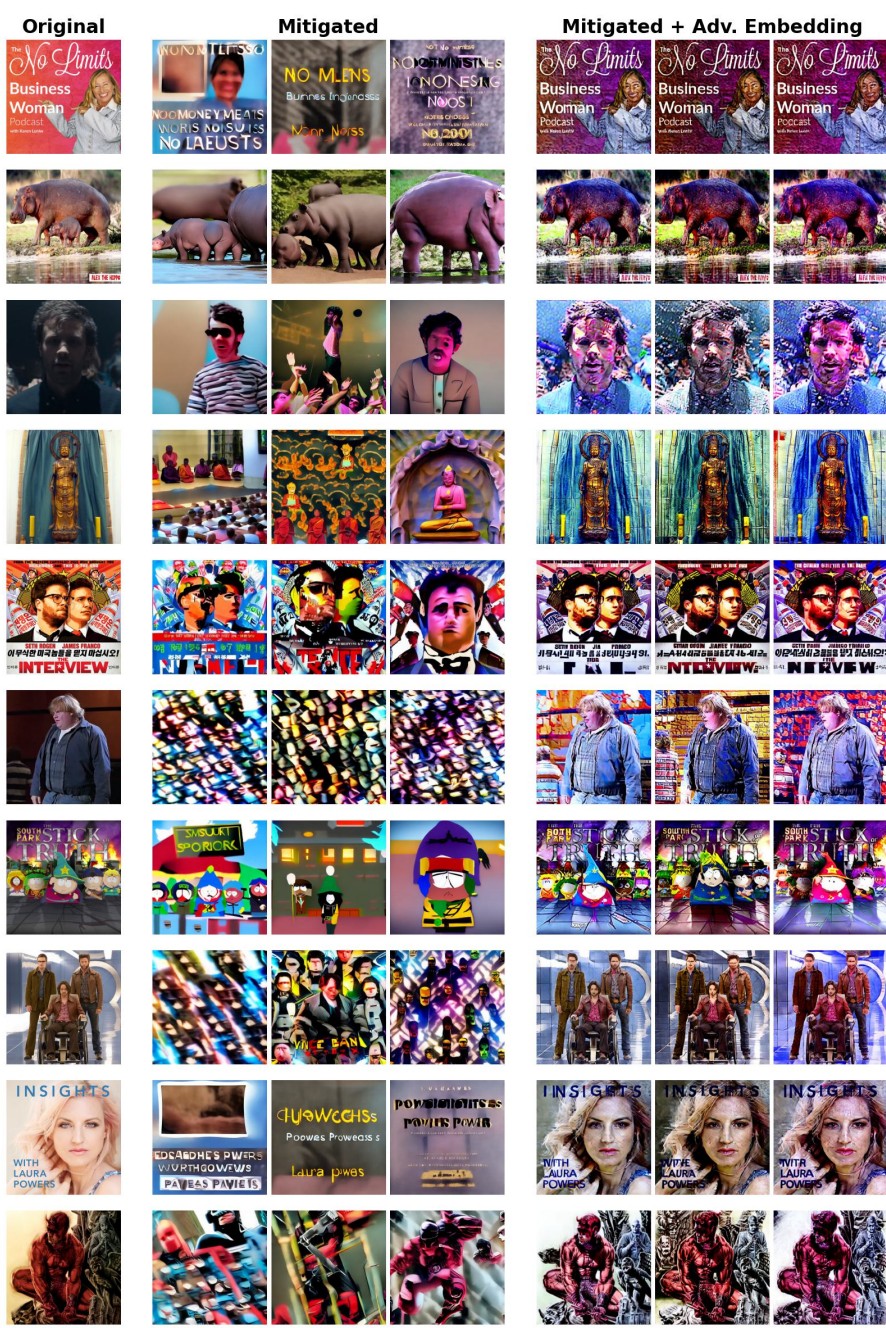

Figure 18: **Qualitative results after applying Wanda with a sparsity of 10%.** The first column shows the original training images. The next three columns show generations after applying the mitigation technique. The final three columns show generations from adversarial embeddings, also after applying the mitigation technique. The adversarial embeddings were initialized with memorized prompt embeddings and optimized for 50 steps.

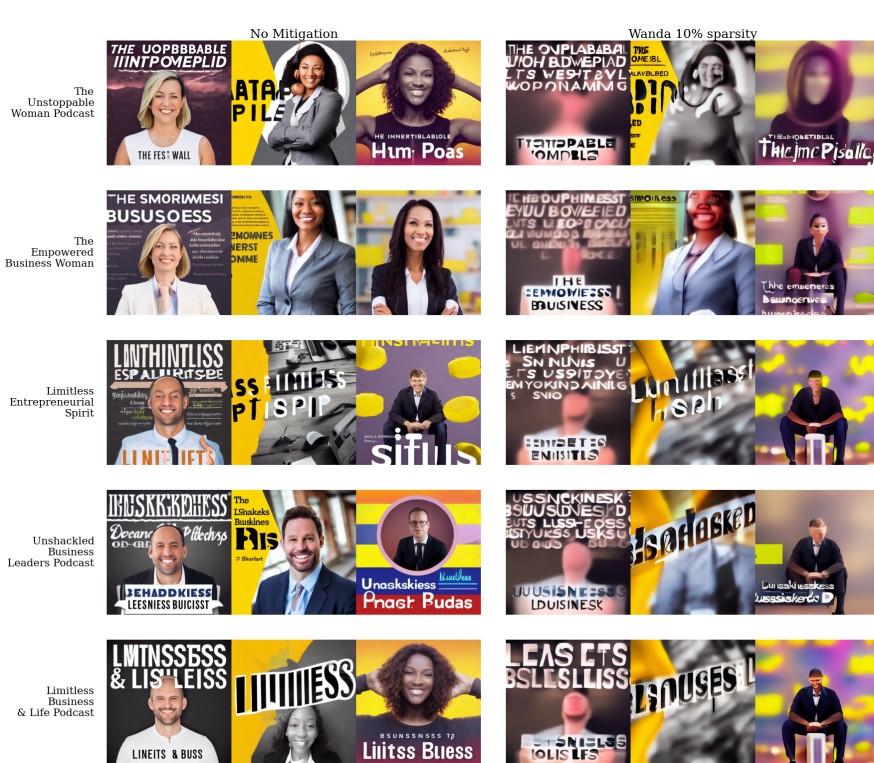

Figure 19: **Qualitative results for damage to concepts after applying Wanda with a sparsity of 10%.** On the left we show the paraphrased prompt for "The No Limits Business Woman Podcast" memorized prompt (VM). The first three images from the left depict generations from SD-v1.4 without mitigation, and the next three—images generated with Wanda after pruning 10% weights.

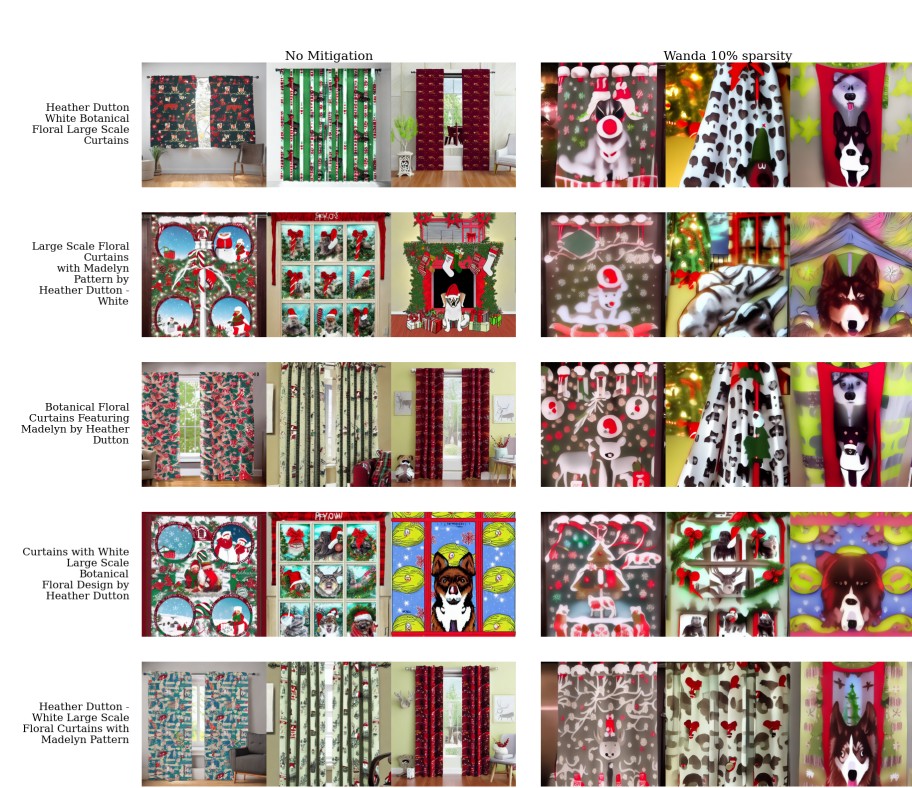

Figure 20: **Qualitative results for damage to concepts after applying Wanda with a sparsity of 10%.** On the left we show the paraphrased prompt for "Plymouth Curtain Panel featuring Madelyn - White Botanical Floral Large Scale by heatherdutton" memorized prompt (TM). The first three images from the left depict generations from SD-v1.4 without mitigation, and the next three—images generated with Wanda after pruning 10% weights.

## K.5 Qualitative Results for Iterative NeMo Pruning

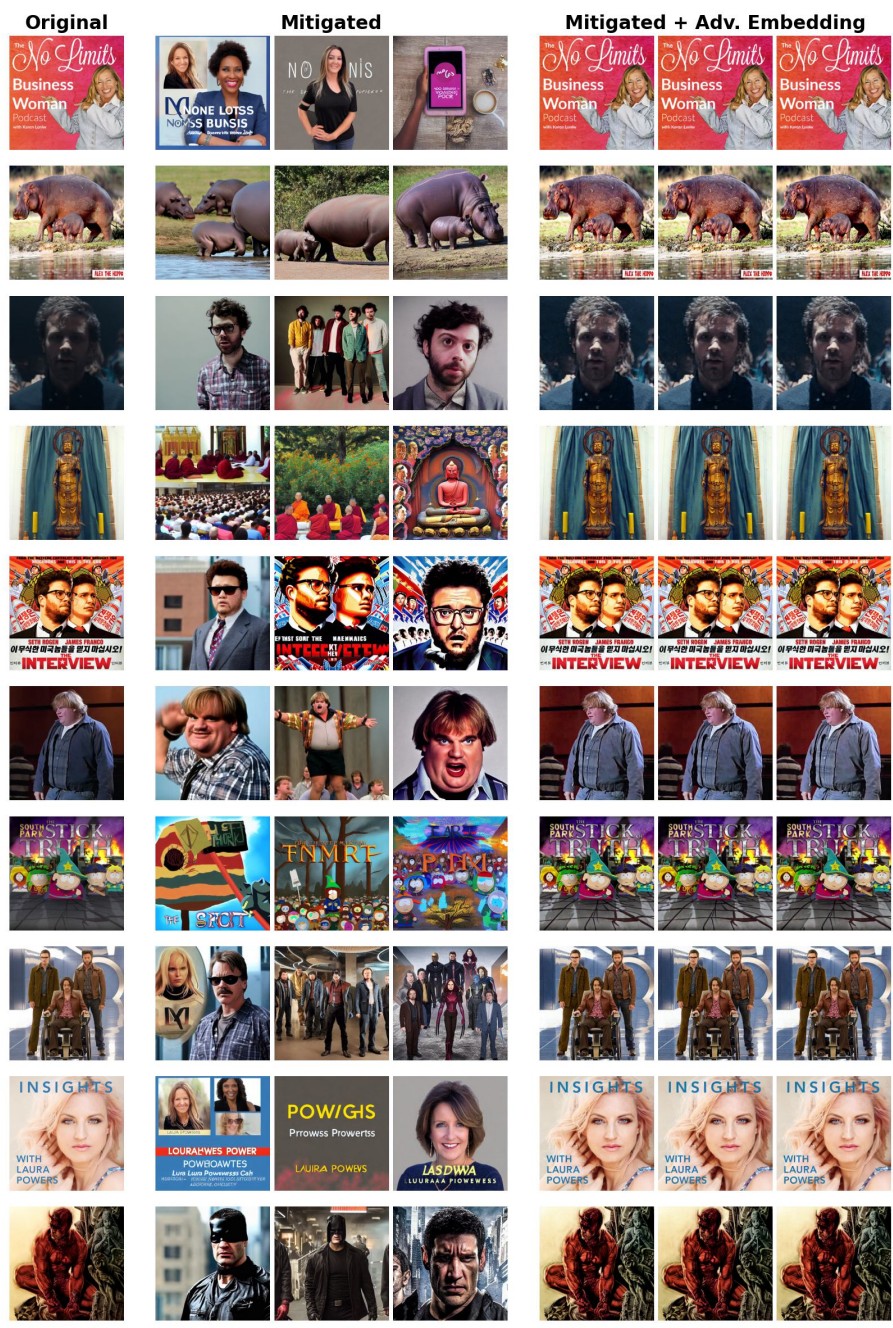

Figure 21: **Qualitative results after applying NeMo iteratively 5 times.** The first column shows the original training images. The next three columns show generations after applying the mitigation technique. The final three columns show generations from adversarial embeddings, also after applying the mitigation technique. The adversarial embeddings were initialized with memorized prompt embeddings and optimized for 50 steps.

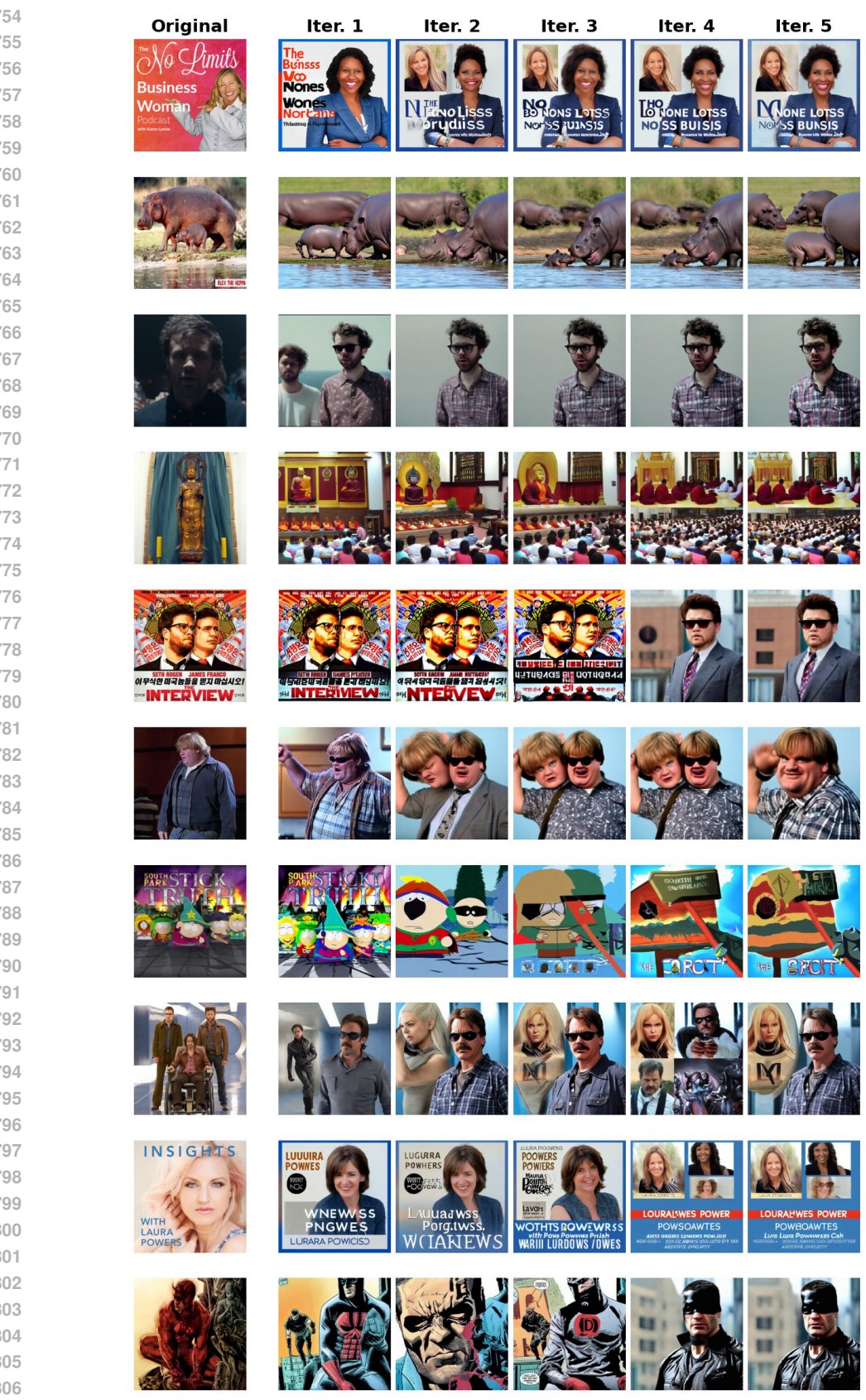

Figure 22: **Qualitative results after applying NeMo iteratively 5 times.** The first column shows the original training images. The next five columns show generations after applying the mitigation technique iteratively. It can be seen that after five iterations the quality seems to degrade a bit.

