# OpenReview forum: "Finding Dori: Memorization in Text-to-Image Diffusion Models Is Not Local"
_ICLR.cc/2026/Conference — Submitted to ICLR 2026_

### Official Review · Reviewer_kJxd · 2025-10-17

**Soundness:** 2
**Presentation:** 3
**Contribution:** 2
**Rating:** 0
**Confidence:** 5

**Summary:**

This paper shows that memorization in text-to-image diffusion models is not localized. It finds that pruning specific “memorization neurons” (as in NeMo or Wanda) only hides, rather than removes, memorized images—since adversarial text embeddings can still re-trigger those replications.

**Strengths:**

1. The paper presents compelling evidence that may overturn the previously assumed locality of memorization in diffusion models.

2. The writing is very clear.

**Weaknesses:**

I agree with some of the authors’ viewpoints — in particular, that memorization is indeed difficult to mitigate through pruning. The concept of locality may need to be considered in two parts: detection and mitigation. When memorization occurs, certain neural activations become significantly stronger, which can clearly be useful for detection. However, pruning these activations is not sufficient to mitigate memorization. We can observe strong compensatory effects — meaning that multiple neural structures (e.g., heads) can contribute to memorization. When one head or neuron is pruned, others can take over. Moreover, these neurons might even be dynamic rather than fixed.

That said, I still believe the authors’ method does not convincingly demonstrate that pruning fails to mitigate memorization. I do not think the second row of Table 1 adequately addresses this concern. The adversarial token embedding itself introduces a large amount of external information. It is possible that the image has already been forgotten, or that memorization has been mitigated, but the adversarial method injects external information, enabling the model to relearn the image during inference and thus generate it again. I recommend the author to refer this paper [1]. This gives a more clear explanation.

Table 4 in Appendix H.2 also supports this argument. For a non-memorized image, after 150 adversarial steps, the generated image’s SSCD score rises from 0.17 to 0.65 (± 0.06), only 0.05 below the threshold defined by the authors. This demonstrates that the proposed method fails to prevent memorization. And the adversarial token embedding privde large information for the model to learn during inference. Meanwhile, the template’s memorization SSCD reaches at most 0.65, which suggests that Nemo might indeed be helpful (although I also believe Nemo is not necessarily effective).

[1] Soft Token Attacks Cannot Reliably Audit Unlearning in Large Language Models.

**Questions:**

See weakness.

---

> ### Author Response · Authors · 2025-11-20
> **Dori does not cause replication of non-memorized images**
>
> Thank you for agreeing with us that the localization of memorization, as it has been done until now, is not adequate to mitigate memorization. We understand that the concern of the reviewer is that our adversarial embeddings might lead to providing additional information, with any given textual conditioning might result in a replicated image when using Dori.
>
> To visualize that this is actually not the case, we have plotted the memorization rate (MR) over a different number of Dori iterations. As can be seen, in the early optimization steps, the MR metric of the memorized (TM & VM) is rapidly increasing, while the MR of non-memorized images is only at 2% after 50 iterations of Dori. This means that when applying Dori for 50 iterations on non-memorized prompts, crafting adversarial embeddings has a false-positive rate of only 2%. For the memorized images, on the other hand, the MR is at 99% and 73% for NeMo and Wanda after 50 iterations of Dori, showcasing that Dori effectively triggers memorization without doing so for non-memorized prompts.
>
> Memorization rates on memorized samples after pruning-based mitigations (NeMo, Wanda) and on non-memorized samples are shown in the table below.
>
> | Optimization Steps| 0  | 1  | 10 | 25 | 50 |
> |---------|----|----|----|----|----|
> | NeMo    | 0.2 | 0.73 | 0.79 | 0.94 | 0.99 |
> | Wanda   | 0.00  | 0.02  | 0.18 | 0.49 | 0.73 |
> | Non-Memorized LAION Samples| 0.00  | 0.00  | 0.00  | 0.00  | 0.02  |
>
> We also provide qualitative examples of adversarial embeddings crafted from memorized prompts (after applying pruning-based mitigations) as well as from non-memorized prompts. The results shown in Fig. 2a and Fig. 2b (in the revised version of the paper) clearly illustrate that, for memorized content, adversarial embeddings still induce close replication of the target images. In contrast, for non-memorized content, there remains a clear visual distinction between the target images and the generated outputs.
>
> Because we explicitly assessed the risk of inadvertently overfitting the generator through adversarial embeddings (using 50 optimization steps, as described in the main paper) and found no such effect, both qualitatively and quantitatively, we are confident that our results robustly demonstrate the limitations of pruning-based mitigation methods. While Soft Token Attacks are similar in nature to our Dori, we want to highlight that we are talking about very different models here. The paper mentioned investigates LLMs and their ability to complete sequences. In contrast to that in our paper we are concerned with text-to-image models. As our analysis has shown, it seems that these text-to-image models seem to behave differently to changes in the text embeddings.
>
> We, therefore, kindly ask the reviewer to reassess their initial score.

---

> ### Public Comment · ~Shenghai_Yuan1 · 2025-11-28
> **Analysis of a Review Containing Factual Errors and Misinterpretations**
>
> This comment aims to clarify several factual inaccuracies and technical misunderstandings in the above review.
> It is not a rebuttal of opinions, but a correction of statements that contradict the paper text or the fundamentals of diffusion models.
> All citations refer to the official submission:
>
>
> ## 1. **Incorrect claim about “relearning” during inference**
>
> - The review states that adversarial embeddings “inject external information” and allow the model to “relearn” images.
> - This is **factually incorrect**: diffusion inference never updates model parameters.
>     - Only forward passes of \epsilon_\theta occur.
>     - No modifications to \theta can happen during inference.
> - Optimizing the input embedding y_{adv} cannot cause the model to relearn or reconstruct forgotten training data.
>
> ---
>
> ## 2. **Misinterpretation of non-memorized results (Table H.2)**
>
> - The review interprets SSCD ≈ 0.65 after 150 adversarial steps as evidence that mitigation fails.
> - However:
>     - The paper defines memorization as SSCD ≥ 0.7 (Sec. 3.2).
>     - Appendix F shows that non-memorized images typically require more than 500 steps to approximate.
> - Therefore:
>     - Memorized images re-trigger quickly.
>     - Non-memorized images do **not** re-trigger under the same compute budget.
> - The review’s interpretation contradicts the actual findings.
>
> ---
>
> ## 3. **Conflation of template vs. verbatim memorization**
>
> - The review uses template-level SSCD (~0.65) to argue that pruning “might still work.”
> - But Table 1 shows adversarial embeddings restore **verbatim** replication after pruning:
>     - NeMo: SSCDOrig ≈ 0.91
>     - Wanda: SSCDOrig ≈ 0.76
> - Template similarity cannot be used to refute verbatim-level evidence.
> - The paper explicitly distinguishes these two forms of memorization (Sec. 2.2).
>
> ---
>
> ## 4. **Inappropriate analogy to LLM soft-token attacks**
>
> - The referenced LLM work concerns:
>     - discrete token embeddings,
>     - audit reliability,
>     - architectures unrelated to diffusion models.
> - Diffusion models rely on continuous text embeddings y and score-based denoising.
> - The analogy is technically invalid.
>
> ---
>
> ## 5. **Logical inconsistency inside the review**
>
> - The review states:
>     - “multiple neural structures compensate; pruning is insufficient,”
>     - and also “pruning might still be effective.”
> - These two claims contradict each other.
> - The first statement actually supports the paper’s findings on distributed memorization.
>
> ---
>
> ## 6. **The “0: strong reject” rating is not supported by the review’s reasoning**
>
> - ICLR’s lowest rating is normally reserved for:
>     - fabricated or irreproducible data,
>     - unsound methodology,
>     - unethical experiments,
>     - no meaningful contribution.
> - Yet the review acknowledges:
>     - the paper is clearly written,
>     - the problem is important,
>     - pruning alone may be insufficient.
> - The assigned rating does not logically follow from the critique presented.
>
> ---
> In summary, the provided review does not accurately reflect the paper’s methods or results, and several key claims are factually incorrect. The concerns you raise are either based on technical misconceptions or directly contradicted by the evidence provided in the submission.

---

> > ### Author Response · Authors · 2025-12-01
> > **Rebuttal Summary**
> >
> > Since reviewers are not allowed to post any further comments, we would like to follow up with a summary of our rebuttal.
> >
> > In particular, we have:
> > - Clarified why adversarial embeddings do not introduce new information, not enabling the model to “relearn” forgotten content. We provided quantitative evidence showing that Dori triggers memorization only for truly memorized samples, while exhibiting an exceptionally low false-positive rate (2% after 50 steps) on non-memorized prompts.
> > - Added memorization-rate curves across optimization steps for NeMo, Wanda, and non-memorized samples, illustrating that pruning-based defenses collapse under adversarial embeddings only when the underlying content has not been removed. These results strengthen our conclusion that pruning does not erase memorization.
> > - Included additional qualitative comparisons (now in Fig. 2a and Fig. 2b) to visually demonstrate that adversarial embeddings reliably recover memorized images even after pruning, while leaving non-memorized images visually distinct.
> > - Explained why the concerns raised in [1] are not directly applicable to our setting: While LLMs use discrete embeddings, diffusion models rely on continuous text-embeddings. Beyond this conceptual mismatch, our empirical results further show that diffusion models respond differently to adversarial embeddings and do not exhibit the failure modes reported for LLMs.
> > - Addressed the reviewer’s question regarding pruning effectiveness by providing targeted analyses showing no overfitting or unintended “relearning” occurs during Dori optimization, further supporting our conclusion that pruning-based methods fail to remove memorized content.
> >
> > We hope this summary supports the assessment of our paper and provides a clear overview of how our rebuttal addressed the reviewers’ concerns. We sincerely thank everyone involved for their time and thoughtful consideration.

---

### Official Review · Reviewer_ocCi · 2025-10-31

**Soundness:** 2
**Presentation:** 3
**Contribution:** 3
**Rating:** 6
**Confidence:** 3

**Summary:**

In order to prevent diffusion models from generating replication of their training data and cause privacy issues, recent works have applied pruning based memory mitigation methods. This paper challenges the memorization locality assumption which is the fundamental of pruning based methods. They claimed that pruning methods do not fully erase the memory, and an adversarial text embedding may re-trigger the training data. They further show that memorization occurs at various places, and proposed a global fine-tuning method that overall solves the problem.

**Strengths:**

Strengths:
1. Meaningful observation. This paper provides an insightful questioning about the memorization locality assumption of pruning based mitigation methods, showing that the memory is not fully erased even after pruning. The observations related to memory have the potential to give awareness to later researchers studying memorization of diffusion models. This is the major reason I think it is worth accepting.
2. Clarity of the structure. The story line of the paper is clear, from the method inducing the observation, explaining the reason behind this observation, to the proposed new method based on this observation, which improves the reader’s experience.
3. Methodology. This paper presents an innovative method through adversarial embedding, adversarial embedding optimization and fine-tuning. The approach appears logical from my review, but there may be further work focusing on efficiency improvement.

**Weaknesses:**

Weaknesses:
1. Experiment fairness. In section 4.4 ‘our mitigation’ is optimized upon Dori’s adversarial embeddings, afterwards, it is compared to baselines again with Dori as in Table 2. I wonder if this experiment setting may cause an unfair situation to baselines. Since only ‘our mitigation’ is optimized on adversarial methods, the baselines have no chance to gain such advantage.
2. Supplementary experiments. if ‘our mitigation’ approach has another version without participation of adversarial embeddings, we may have a better comparison with the baseline similar to the format of Table 1, i.e. comparison with and without Dori, to present performance advantage comprehensively and fairly (which could also solve weakness 1).
3. Diagram issue. Figure 1, is the right part redundant with the left part (2) to (3). If so, we may merge the right part with the left and add the details between (2) and (3).

**Questions:**

Questions:
1. In 4.4 Approach, do we need to train a set of adversarial embeddings before each fine-tuning optimization? If so, would the time consumption of the training be a concern?
2. In 4.4 Approach, is the text embedding being optimized by both fine-tuning and Dori? Would that be a conflict or redundancy?

---

> ### Author Response · Authors · 2025-11-20
> **Evaluation fairness, details about adversarial embeddings**
>
> >**W1: ‘Our mitigation’ is optimized upon Dori’s adversarial embeddings, afterwards, it is compared to baselines again with Dori as in Table 2. [Would it] cause an unfair situation to baselines?**
>
> Our mitigation method fundamentally relies on adversarial training, i.e., training the model to be robust against adversarial examples. Because adversarial embeddings are an integral part of this procedure, there is no meaningful way to apply our method without them. To address concerns about fairness, we conducted **extensive sensitivity and ablation studies** in Appendix I.3 and I.4, including settings without the mitigation loss (i.e., without adversarial embeddings). In these cases, no mitigation effect is observed, which is expected because the fine-tuning effectively reduces to standard training without any mechanism to counteract memorization.
>
> More broadly, our goal is not to show that our method performs better simply because it uses adversarial embeddings; rather, our goal is to demonstrate that existing pruning-based methods do not yet account for the **fundamentally non-local nature** of memorization in diffusion models. Because these methods assume locality, they are inherently limited in their ability to erase memorized content.
> Our results suggest that explicitly incorporating mechanisms that address non-locality, such as adversarial training paired with full-finetuning of the model, can substantially improve mitigation effectiveness.
>
> >**W2: ‘Our mitigation’ [...] without adversarial embeddings [versus baselines].**
>
> As shown in Table 3, our method remains more effective than both ESD and Concept Ablation even when no adversarial embeddings (i.e., no Dori) are used during evaluation. In this setting, our mitigation reduces the memorization rate to 0.00, whereas ESD and Concept Ablation still exhibit **substantially higher memorization rates** (e.g., 0.11 MR for ESD and 0.29 for Concept Ablation). This demonstrates that our approach is not only robust to adversarial embeddings but also intrinsically more effective at mitigating memorization in standard (non-adversarial) scenarios.
> In addition, Appendix I.3 and I.4 provide detailed sensitivity analyses and ablation studies for our mitigation method. If there are additional comparisons or evaluation settings that the reviewer believes are missing, we would be happy to include them and provide the corresponding results.
>
> >**W3: Fig. 1 redundancy.**
>
> We would like to emphasize that the right part of Figure 1 is intended to highlight that there exist many adversarial embeddings that follow different trajectories, yet still converge to the memorized content. This visualization is central to our argument about the **non-locality of memorization**, which we provide in Section 4 of our paper. On the left side of Figure 1, we present localization-based mitigation strategies, which are **insufficient** to remove memorized images from the model, as we show in Section 3.
>
> >**Q1: In 4.4 Approach, do we need to train a set of adversarial embeddings before each fine-tuning optimization? would the time consumption of the training be a concern?**
>
> Yes, in each fine-tuning iteration we search for adversarial embeddings that trigger the memorized image. We measured the full runtime of our mitigation procedure, which is detailed in Appendix I.5. The results reported in the main paper were obtained after approximately 2.5 hours of training on a single A100 GPU, including the iterative search for adversarial embeddings. Compared to pruning-based methods, this runtime is reasonable, particularly since the model needs to be fine-tuned only once to effectively remove memorization.
>
>
> >**Q2: In 4.4 Approach, is the text embedding being optimized by both fine-tuning and Dori? Would that be a conflict or redundancy?**
>
> The text embeddings are optimized only using Dori, while the whole diffusion model is fine-tuned with the loss defined in Equation 2 to guide the generation from these embeddings towards surrogate (non-memorized) images. Our method steers the adversarial embeddings away from the memorized content to mitigate memorization, similarly to how adversarial training brings back adversarial examples back to the correct side of the decision boundary for image classifiers to increase robustness.
>
> Because of that, the fine-tuning is inherently at conflict with Dori, but this is desired, as it aims to prevent replication of memorized images even if a strong adversarial trigger generation, i.e., Dori, is applied.

---

> > ### Author Response · Authors · 2025-12-01
> > **Rebuttal Summary**
> >
> > Since reviewers are not allowed to post any further comments, we would like to follow up with a summary of our rebuttal.
> >
> > In particular, we have:
> > - Clarified why adversarial embeddings are an essential component of our mitigation method, and summarized the ablation and sensitivity studies (Appendix I.3–I.4) showing that fine-tuning without adversarial embeddings has no meaningful mitigation effect. These analyses address fairness concerns in comparison to the pruning-based baselines (W1).
> > - Highlighted our results comparing our method and prior baselines in settings without adversarial embeddings, demonstrating that our mitigation reduces memorization to 0.00 MR even in the standard evaluation. In contrast, ESD and Concept Ablation have higher memorization rates (W2).
> > - Explained the purpose of Figure 1 and why the right-hand visualization is necessary to illustrate the existence of many adversarial embeddings, following different trajectories and yet still converging to the same memorized output. This directly supports our argument about non-locality (W3).
> > - Addressed the questions about computational overhead by reporting the full measured runtime of our mitigation procedure. The fine-tuning takes approximately 2.5 hours on a single A100 GPU, including adversarial embedding search. Further details are available in Appendix I.5 (Q1).
> > - Clarified the interaction between fine-tuning and adversarial embedding optimization, explaining that Dori optimizes only the text embeddings while the fine-tuning updates the full model to push generations away from memorized images. This intentionally creates the desired conflict that prevents data replication even under adversarial triggers (Q2).
> >
> > We hope this summary supports the assessment of our paper and provides a clear overview of how our rebuttal addressed the reviewers’ concerns. We sincerely thank everyone involved for their time and thoughtful consideration.

---

### Official Review · Reviewer_oYWT · 2025-11-01

**Soundness:** 3
**Presentation:** 3
**Contribution:** 3
**Rating:** 8
**Confidence:** 4

**Summary:**

This paper addresses the memorization problem in diffusion models, which causes concerns about data privacy using these generative models. Specifically, this paper follows the line of work (NeMo and Wanda) that mitigates memorization by pruning the local weights responsible for triggering verbatim training data replication, and challenges their locality assumption. Using a series of adversarial experiments, the authors demonstrate that even after applying pruning-based defenses such as NeMo and Wanda, small perturbations in text embeddings can re-trigger verbatim generation of memorized training images. Based on this, the paper introduces Dori, an adversarial-embedding optimization procedure, to uncover these vulnerabilities and show that memorization is not localized in either text-embedding, activation, or weight space. Finally, it proposes adversarial fine-tuning, a global mitigation strategy that effectively removes memorized content without degrading model utility.

**Strengths:**

1. The paper is well-written and easy to follow.
2. The paper provides a novel and practical perspective on memorization in text-to-image diffusion models by explicitly challenging the locality assumption underlying prior pruning-based mitigation strategies.
3. The methodology is rigorous and comprehensive.
4. The experimental setup is sound, and the metrics used are clearly defined and justified. The authors thoroughly compare Dori against NeMo, Wanda, SISS, and concept-unlearning baselines (ESD, Concept Ablation). The results are also impressive.
5. The practical significance of this paper is high, as it contributes towards better privacy-preserving diffusion models and calls into question the reliability of existing “local pruning” defenses.

**Weaknesses:**

1. **Model Scope.** The analysis focuses exclusively on Stable Diffusion v1.4, the only model with known memorized prompts. While the authors justify this choice, it limits claims of generality. Extending the evaluation to even a partially curated SD v1.5 or fine-tuned variants would strengthen the argument. Could the authors comment on whether non-local memorization might emerge differently in larger or more recent models (e.g., SDXL or FLUX)?
2. **Computational Overhead.** The proposed adversarial fine-tuning requires multiple rounds of adversarial embedding optimization and full-parameter updates, which can be expensive. The paper acknowledges this but could provide clearer quantification of runtime and memory overhead relative to pruning methods.
3. **Limited Theoretical Analysis.** While the paper empirically demonstrates non-locality, it does not provide a deeper theoretical explanation for why memorization becomes distributed in diffusion models.

**Questions:**

Please see the above sections for details. In addition, do the authors expect similar non-locality in non-text-conditional (unconditional) diffusion models?

---

> ### Author Response · Authors · 2025-11-20
> **Model scope, runtime, theoretical explanation**
>
> >**W1: Model Scope.**
>
> Thank you for pointing that out. We did not include experiments on Stable Diffusion 2 as neither NeMo nor Wanda were tested on these models. While these prompts are classified as “memorized”, all these prompts are very similar and sometimes even have the same original image, making it rather unsuitable for evaluating memorization mitigation techniques. However, as requested by the reviewer, we performed additional experiments on Stable Diffusion v2.0 using the Webster [1] prompts, consistent with the setup in Ren et al [2]. To stabilize the evaluation, we first collected all original images and filtered the generated samples using the SSCD score (threshold = 0.7), yielding 32 clearly memorized image–text pairs. We then conducted the following experiments and analyses:
> - Applied NeMo and Wanda to mitigate memorization via pruning.
> - Evaluated the robustness of these pruned models against adversarial embeddings generated with Dori, using the same parameters as in the main paper.
> - Applied our fine-tuning–based mitigation and evaluated its effectiveness against both memorized text prompts and adversarial embeddings. Again, using the same parameters as in the main paper.
>
> Setting|Adv.Steps|↓SSCD_Orig|↓SSCD_Gen|↓D_SSCD|↑A_CLIP|↓MR|↓FID|↓KID
> -|-|-|-|-|-|-|-|-
> No Mitigation|-|0.77±0.04|-|0.98±0.00|0.31±0.01|1.0|15.21|0.0076
> NeMo|-|0.23±0.07|0.20±0.04|0.71±0.07|0.32±0.01|0.06|15.35|0.0072
> Wanda|-|0.09±0.03|0.10±0.03|0.45±0.10|0.29±0.01|0.00|16.79|0.0075
> Our Mitigation|-|0.06±0.03|0.02±0.03|0.67±0.05|0.30±0.01|0.00|16.01|0.0071
> NeMo+Dori|50|0.87±0.02|0.74±0.12|0.99±0.00|0.31±0.01|1.00|15.35|0.0072
> Wanda+Dori|50|0.70±0.04|0.19±0.05|0.84±0.08|0.31±0.01|0.53|16.79|0.0075
> Our Mitigation+Dori|50|0.14±0.06|0.09±0.04|0.70±0.04|0.31±0.01|0.06|16.01|0.0071
>
> Our results are consistent with the findings reported in the main paper: while both NeMo and Wanda initially appear effective at mitigating memorization, Dori remains capable of inducing data replication. In contrast, our fine-tuning strategy provides a substantially more robust mitigation, reflected in a significantly lower memorization rate (MR), while preserving overall image quality (as measured by FID/KID). More details regarding the experiments can be found in Appendix H.8.
>
> **References:**
>
> [1] Webster, "A reproducible extraction of training images from diffusion models," Arxiv.
>
> [2] Ren et al., Unveiling and Mitigating Memorization in Text-to-image Diffusion Models through Cross Attention, ECCV 2024
>
> >**W2: Runtime and memory overhead.**
>
> We provided a detailed analysis of time and memory consumption in Appendix I.5 in the initial version of the paper. All experiments were conducted on a single A100 GPU, and the reported runtime accounts for the entire procedure, including multiple rounds of adversarial embedding optimization. The mitigation results presented in the main paper were obtained after approximately **2.5 hours** on this single GPU. We consider this computational cost reasonable, especially when compared to the initial training of diffusion models, which typically requires hundreds of GPUs (e.g., 256 A100s for Stable Diffusion v1.4) running for several weeks.
>
> >**W3: Theoretical explanation.**
>
> We appreciate the reviewer’s point regarding the theoretical analysis. In this work, we provide the **first empirical demonstration** that memorization in diffusion models is inherently non-local. Our results show that, although denoising trajectories may diverge substantially during generation, they can nonetheless converge to the same embedding, which explains why mitigation strategies that assume locality consistently fail. This insight establishes the core mechanism behind non-local memorization.
>
> While we agree that a deeper formal theory would be valuable, we believe our work lays essential groundwork for such future theoretical developments. Importantly, our claims are supported by an extensive analysis of model behavior, including both qualitative (generated image comparison, embedding visualizations) and quantitative evaluations (SSCD, weight agreement, CLIP alignment). Together, these analyses provide a comprehensive empirical foundation for understanding and further theorizing about non-local memorization in diffusion models.
>
> >**Q1: Non-locality in non-text-conditional (unconditional) diffusion models?**
>
> This is indeed an interesting question that is definitely worth investigating in future work. The main culprit for the non-locality is the ambiguity of the diffusion trajectories in the noise space. The way these trajectories are guided is by the (text-)conditioning. Therefore, our intuition would be that for non-text-conditional models memorization is also not local. Unconditional diffusion models, on the other hand, are probably more prone to have localized memorization because the trajectories are solely influenced by the initial random noise, leading to the same trajectories, given a specific initial random noise.

---

> > ### Author Response · Authors · 2025-12-01
> > **Rebuttal Summary**
> >
> > Since reviewers are not allowed to post any further comments, we would like to follow up with a summary of our rebuttal.
> >
> > In particular, we have:
> > - Expanded our experimental scope by conducting new evaluations on Stable Diffusion v2.0. These experiments include pruning-based mitigation (NeMo, Wanda), adversarial-embedding attacks via Dori, and our fine-tuning–based mitigation. The results (Appendix H.8) are aligned with the findings reported for SD v1.4 and support our claims regarding non-local memorization (W1).
> > - Provided a clear quantification of the computational overhead of our mitigation in Appendix I.5, detailing runtime and memory usage on a single A100 GPU. We confirm that the full procedure (including adversarial embedding optimization) completes in approximately 2.5 hours, which is reasonable relative to the cost of training diffusion models from scratch (W2).
> > - Clarified the empirical basis for the observed non-local memorization phenomenon and explained why diffusion trajectories can diverge yet converge to identical outputs, highlighting the underlying mechanism, even though a full formal theory remains an open direction for future research (W3).
> > - Addressed the question of unconditional diffusion models by discussing how the role of conditioning in guiding trajectories may influence the emergence of non-locality, and provided our intuition on how this behavior might manifest in unconditional settings (Q1).
> >
> > We hope this summary supports the assessment of our paper and provides a clear overview of how our rebuttal addressed the reviewers’ concerns. We sincerely thank everyone involved for their time and thoughtful consideration.

---

### Official Review · Reviewer_9wY6 · 2025-11-01

**Soundness:** 2
**Presentation:** 2
**Contribution:** 2
**Rating:** 2
**Confidence:** 4

**Summary:**

This paper investigates memorization in text-to-image diffusion models and argues that such memorization is non-local—that is, it cannot be mitigated by pruning a small set of neurons or weights. The authors introduce Dori, an adversarial embedding optimization method that can recover memorized images even after pruning defenses, and propose an adversarial finetuning strategy as a stronger mitigation. Experiments on Stable Diffusion 1.4 demonstrate that pruning-based defenses merely “hide” memorized content rather than removing it.

**Strengths:**

1. The paper addresses an important topic, memorization and privacy in diffusion models, which is highly relevant to model safety and responsible AI.
2. The paper is well-written and highlights an underexplored dimension of diffusion model memorization behavior.

**Weaknesses:**

1. **Unrealistic scenario.** The proposed setting is impractical in the real world. The proposed finetuning-based method seems to align with the model publisher’s side (those who wish to make their models trustworthy). However, the paper assumes a white-box adversary with full access to the source code, as explicitly mentioned by the authors. However, most real-world text-to-image (T2I) systems, such as Midjourney or ChatGPT, only expose text-level APIs, not model internals. Therefore, the claimed adversarial setting does not reflect realistic threat models or deployment scenarios.

2. **Method is both unrealistic and impractical.** Even if a white-box API model existed, the proposed finetuning approach is infeasible for a model publisher. Finetuning has no practical value unless the memorized images are already known. Existing training-time approaches can prevent memorization from scratch, and inference-time approaches can operate without knowing whether a given prompt corresponds to a memorized sample. In contrast, the proposed finetuning method requires prior knowledge of memorized images—which is itself extremely impractical to obtain. For instance, prior works [1,2] identify memorized images by exhaustively scanning and regenerating huge datasets such as LAION-2B. Thus, using the proposed method would first require massive precomputation to find memorized images, before even performing the fine-tuning itself. The authors themselves acknowledge that they can only test on Stable Diffusion 1.4, precisely because memorized images are public only for that model. Moreover, Stable Diffusion 3 already shows that simple image deduplication in the dataset effectively prevents such memorization during training. Altogether, the method is highly impractical for any real API provider or model publisher.

3. **Questionable performance evaluation.** The authors report only SSCD and MR metrics in the main paper, while omitting another critical metric for memorization—CLIP score. In the appendix, their method actually achieves the lowest CLIP score among the compared methods. This discrepancy raises concern: Table 1 (pruning methods result) includes CLIP scores, but Table 2 (finetuning methods including their own method) omits them. This selective reporting appears intentional and undermines confidence in the claimed performance.

4. **Limited scalability and incomplete experimental coverage.** The paper evaluates only on Stable Diffusion 1.4, claiming that no other models have public memorized-image datasets. However, the dataset provided by Webster [2] (which the authors used) also includes prompts for Stable Diffusion 2—though smaller in number, they are sufficient for at least limited experiments. As someone who has used that dataset myself, I can confirm that while SD 2 has fewer memorized samples, it is by no means impossible to test. The authors’ decision not to include it weakens the generality of their conclusions.

[1] Carlini et. al., "Extracting training data from diffusion models," USENIX'23.

[2] Webster, "A reproducible extraction of training images from diffusion models," Arxiv.

**Questions:**

The fundamental question is when and where this method would be used. Given Weakness #1 and #2, the authors must clearly describe a realistic scenario in which their approach could be practically deployed.  At present, the paper feels like it constructs a problem that does not exist in real-world T2I systems.

---

> ### Author Response · Authors · 2025-11-20
> **Practicality and performance evaluation (Comment 1 of 2)**
>
> >**W1: Unrealistic scenario.**
>
> We would like to clarify that the goal of our paper is *not* to propose a real-world attack against diffusion models. We instead aim to verify whether the assumption that memorization is localized in these models, which prior work has explored to design targeted mitigation strategies. To test this hypothesis, we rely on open models whose memorization has been studied by the prior work. In addition to the models we explored following related work, there are many other state-of-the-art text to image models that are open source, like SD3, Flux, DeepFloyd IF, SD-XL which are publicly released, sparking practical privacy and copyright concerns regarding memorized content.
>
> >**W2: Unrealistic and impractical method.**
>
> We would also like to clarify that the primary goal of the paper is *not* to propose a practical defense mechanism. Instead, the adversarial fine-tuning emerged from our analysis of the locality hypothesis and represents a first step toward improved memorization-mitigation techniques. Given that our insights from the analysis of memorization triggers, embedding space, and activations in Sections 4.1-4.3 all suggest that memorization is not localized, in 4.4, we explore the fine-tuning as a non-localized alternative. The increased success of memorization removal in comparison to localized approaches yields us another indicator that memorization is not localized. Our insights can inspire the design of future (practical) mitigation methods to remove memorization *post-training*, in cases when some images slip through the deduplication and become memorized, e.g. for Stable Diffusion 3.5 as shown in [1].
>
>
> >**W3: Performance evaluation.**
>
> We excluded some metrics from the main paper to maintain a clear focus on the most relevant aspect of our work: memorization. However, we report all metrics transparently in Table 3, which is also referenced in the main text, hence there is no obfuscation of results.
>
> Importantly, the CLIP scores shown in Table 3 are computed **only on memorized samples**. They therefore **do not** reflect the overall image quality of the models after mitigation, but rather the alignment between the memorized prompts and their respective memorized images. We would expect a drop in this metric when a mitigation successfully removes memorized images. Specifically, given that both ESD and Concept Ablation fail to mitigate memorization (as indicated by their high SSCD and memorization rates), it is unsurprising that their generated images from memorized prompts exhibit high CLIP alignment with those prompts. In contrast, our mitigation effectively prevents data replication. This is partly reflected in its reduced CLIP scores on the memorized subset.
>
> Crucially, the FID score demonstrates that our approach **does not degrade overall image quality**, especially when compared with Concept Ablation. We are happy to include all metrics directly in the main paper in the final version. When adversarial embeddings are used, a higher CLIP score corresponds to more data replication. So in that scenario, **a lower score is in fact desirable**.
>
> To further demonstrate that our mitigation does not harm image quality or perform worse than existing unlearning methods, we additionally computed the CLIP alignment between 1k **non-memorized** generated images and their corresponding COCO prompts. As the results below show, all three methods perform on par with the baseline, indicating that image–text alignment remains intact.
>
> | Method| Memorization Rate |
> |-|-|
> | Baseline (vanilla SDv1.4) | 0.31 ± 0.03    |
> | CA| 0.31 ± 0.05    |
> | ESD| 0.31 ± 0.05    |
> | Our Mitigation| 0.31 ± 0.03    |
>
>
> **References:**
>
> [1] Low resource reconstruction attacks through benign prompts. Yarkoni S., Livni Roi, 2025.

---

> ### Author Response · Authors · 2025-11-20
> **Scalability, real-world problem (Comment 2 of 2)**
>
> >**W4: Limited scalability and incomplete experimental coverage.**
>
> Thank you for pointing that out. We did not include experiments on Stable Diffusion 2 as neither NeMo nor Wanda were tested on these models. While these prompts are classified as “memorized”, all these prompts are very similar and sometimes even have the same original image, making it rather unsuitable for evaluating memorization mitigation techniques. However, as requested by the reviewer, we performed additional experiments on Stable Diffusion v2.0 using the Webster [1] prompts, consistent with the setup in Ren et al [2]. To stabilize the evaluation, we first collected all original images and filtered the generated samples using the SSCD score (threshold = 0.7), yielding 32 clearly memorized image–text pairs. We then conducted the following experiments and analyses:
>
> - Applied NeMo and Wanda to mitigate memorization via pruning.
> - Evaluated the robustness of these pruned models against adversarial embeddings generated with Dori, using the same parameters as in the main paper.
> - Applied our fine-tuning–based mitigation and evaluated its effectiveness against both memorized text prompts and adversarial embeddings. Again, using the same parameters as in the main paper.
>
> |Setting|Adv.Steps|↓SSCD_Orig|↓SSCD_Gen|↓D_SSCD|↑A_CLIP|↓MR|↓FID|↓KID|
> |-|-|-|-|-|-|-|-|-|
> |No Mitigation|-|0.77±0.04|-|0.98±0.00|0.31±0.01|1.0|15.21|0.0076|
> |NeMo|-|0.23±0.07|0.20±0.04|0.71±0.07|0.32±0.01|0.06|15.35|0.0072|
> |Wanda|-|0.09±0.03|0.10±0.03|0.45±0.10|0.29±0.01|0.00|16.79|0.0075|
> |Our Mitigation|-|0.06±0.03|0.02±0.03|0.67±0.05|0.30±0.01|0.00|16.01|0.0071|
> |NeMo+Dori|50|0.87±0.02|0.74±0.12|0.99±0.00|0.31±0.01|1.00|15.35|0.0072|
> |Wanda+Dori|50|0.70±0.04|0.19±0.05|0.84±0.08|0.31±0.01|0.53|16.79|0.0075|
> |Our Mitigation+Dori|50|0.14±0.06|0.09±0.04|0.70±0.04|0.31±0.01|0.06|16.01|0.0071|
>
> Our new results are consistent with the findings reported in the main paper: while both NeMo and Wanda initially appear effective at mitigating memorization, Dori remains capable of inducing data replication. In contrast, our fine-tuning strategy provides a substantially more robust mitigation, reflected in a significantly lower memorization rate (MR), while preserving overall image quality (as measured by FID/KID). More details regarding the experiments can be found in Appendix H.8.
>
> >**Q1: The paper feels like it constructs a problem that does not exist in real-world T2I systems.**
>
> We would like to kindly disagree with the reviewer. Post-training memorization removal **is not** a new concept, and it has been extensively studied for large language models [2, 3] , as well as text-to-image models like Stable Diffusion [4, 5]. While training- and inference-time mitigation methods provide layers of defense, some training data can still slip through them, and could be replicated. For example, deduplication of the training set of Stable Diffusion 3 was not enough to prevent memorization, as shown in the recent work [1], which successfully extracts some training data from Stable Diffusion 3.5. The same paper reconstructs images from Midjourney v4 and v6.1, which should employ some form of inference-time mitigation. Moreover, inference-time approaches do not provide **a lasting protection** of the potentially sensitive memorized training data for open-source models, since the users can just disable these mechanisms by altering their source code. Since these methods are imperfect, a **post-training mitigation** paradigm aims to remove memorized content from the models when other defenses fall flat. Our work analyzes the reliability of existing post-training approaches, and shows they rely on a faulty locality assumption. We do not construct a problem that does not exist. Instead, we investigate the behavior of current solutions to an **already established problem** , highlight potential shortcomings, and provide initial directions towards effective methods.
>
> **References:**
>
> [1] Low resource reconstruction attacks through benign prompts. Yarkoni S., Livni Roi, 2025.
>
> [2] To each (textual sequence) its own: improving memorized-data unlearning in large language models. Bărbulescu G-O, Triantafillou P., ICML 2024.
> [3] Knowledge unlearning for mitigating privacy risks in language models. Jang, Joel, et al., ACL 2023.
>
> [4] Finding NeMo: Localizing neurons responsible for memorization in diffusion models. Hintersdorf D., et al., NeurIPS 2024.
>
> [5] Memorization is localized within a small subspace in diffusion models. Chavhan R., et al., 2024.

---

> > ### Author Response · Authors · 2025-12-01
> > **Rebuttal Summary**
> >
> > Since reviewers are not allowed to post any further comments, we would like to follow up with a summary of our rebuttal.
> >
> > In particular, we have:
> > - Clarified the intended scope of our work and emphasized that our goal is to examine the locality assumption underlying existing mitigation methods, rather than to propose a directly deployable real-world attack (W1).
> > - Explained the conceptual role of adversarial fine-tuning in our analysis, positioning it as a non-localized alternative to pruning-based approaches and a first step toward more practical mitigations in the future (W2).
> > - Provided a detailed explanation of our evaluation choices, including the role of CLIP alignment on memorized samples, and clarified why lower CLIP scores are expected when memorization is successfully mitigated (W3).
> > - Conducted new experiments on Stable Diffusion 2.0 to broaden the empirical scope. The new results (Appendix H.8) confirm the trends observed in our main experiments (W4).
> > - Addressed the realism of the problem setting by summarizing recent work showing that memorization persists even in modern T2I systems, and by explaining why post-training mitigation remains relevant for both open-source and commercial models (Q1).
> >
> > We hope this summary supports the assessment of our paper and provides a clear overview of how our rebuttal addressed the reviewers’ concerns. We sincerely thank everyone involved for their time and thoughtful consideration.

---

### Author Response · Authors · 2025-12-03
**Summary of the rebuttal.**

We would like to thank the reviewers for acknowledging that our paper addresses an important topic (Reviewer 9wy6) of **high practical significance** (Reviewer oYWT), and that our method is **innovative** (Reviewer ocCi), rigorous and comprehensive (Reviewer oYWT). Our results are described as **impressive** (Reviewer oYWT), “have the potential to give awareness to later researchers studying memorization of [DMs]” (Reviewer ocCi), and “present compelling evidence that may **overturn the previously assumed locality of memorization** in [DMs]” (Reviewer kJxd).

Moreover, our work benefited greatly from addressing the reviewers’ questions, and our results now are even more sound and comprehensive. Specifically:

1. We conducted new experiments on Stable Diffusion 2.0 to broaden the empirical scope. The results are similar as for Stable Diffusion 1.4, *i.e.,* pruning-based mitigation **does not remove** memorized images, while our full fine-tuning method is successfully mitigating it. These results further confirm the **non-local** nature of memorization in DMs (Reviewer 9wY6, oYWT).
2. We clarified the realism of the problem setting by explaining why post-training mitigation remains relevant, since recent work shows that memorization persists **even in modern T2I systems**. Moreover, we addressed concerns about the computational overhead, showing that our method requires **only 2.5h of runtime on a single A100 GPU** (Reviewer 9wY6, ocCi).
3. We explained that adversarial embeddings **do not** introduce new information to the model, since they **do not** alter the model’s weights. Effectively, Dori **does not** cause the model to “relearn” removed images. Instead, Dori reliably recovers images **concealed** by faulty, pruning-based mitigation methods (Reviewer kJxd).
4. In the new Fig. 2a we showed that Dori exhibits **an exceptionally low** false-positive rate of 2% on yielding memorized images from non-memorized prompts. Moreover, the new Fig. 2b highlights that Dori leaves non-memorized images **visually distinct** even after 50 update steps. Based on these targeted analyses, we show that **no overfitting or unintended “relearning”** occurs during Dori optimization. Effectively, Dori is a **sound and precise** tool, with which we show that memorization in DMs **is not local** (Reviewer kJxd).

We hope this summary supports the assessment of our paper and provides a clear overview of how our rebuttal addressed the reviewers’ concerns. We sincerely thank everyone involved for their time and thoughtful consideration.

---

### Meta-Review · Area_Chair_EZS8 · 2026-01-02

**Summary:**

The reviewers were primarily concerned that the paper relies on an unrealistic white-box threat model and assumes prior knowledge of memorized images. This makes the proposed fine-tuning–based mitigation impractical for real-world text-to-image systems. The paper’s framing suggests a deployable mitigation, but the method does not align with realistic deployment scenarios or existing defenses.

**Reviewer Concerns:**

The concerns about realism and practicality remain. The paper still presents adversarial fine-tuning as a mitigation method, but it relies on an unrealistic white-box setting and prior knowledge of memorized images that do not apply to real-world T2I systems.

**Reviewer Scores:**

I believe the reviewer would not have increased their score. While the authors’ response provides additional clarification about their intent, the core concerns would remain. At most, the score might move slightly toward a borderline reject, but not to acceptance.

---

### Decision · Program_Chairs · 2026-01-26

Reject